# p21⁺TREM2⁺ senescent macrophages fuel inflammaging and metabolic dysfunction-associated steatotic liver disease

Cellular senescence drives chronic sterile inflammation during aging via the senescence-associated secretory phenotype, yet the senescent cell types responsible are poorly defined. Macrophages share multiple features of senescence, including inflammatory secretion, yet whether macrophages can adopt a senescent state remains unclear. Here we identify p21⁺Trem2⁺ senescent macrophages as a major source of inflammaging, using primary mouse and human macrophage models of DNA damage and cholesterol-induced senescence characterized by multi-omic profiling. We found that senescent macrophages exhibit a distinctive p21-TREM2 expression profile and senescence-associated secretory phenotype, driven in part by type I interferon signaling via cytosolic mitochondrial DNA. We also found that senescent macrophage accumulation occurs in aging, metabolic dysfunction-associated steatotic liver disease mouse livers, and is enriched in human cirrhotic liver tissue. Finally, senolytic treatment targeting senescent macrophages reduced liver inflammation and steatosis in both aged mice and mice with metabolic dysfunction-associated steatotic liver disease. These findings establish macrophage senescence as a central driver of chronic inflammation in aging and metabolic liver disease, and a tractable therapeutic target.

Aging is a multifactorial process that, beyond reproductive maturity, becomes increasingly maladaptive, driving progressive physiological decline, heightened susceptibility to various aging-related diseases and systemic chronic inflammation, known as inflammaging[1]. Cellular senescence has emerged as a major source of tissue inflammaging, fueling intense interest in senescence-targeted therapies[2,3]. Indeed, preclinical studies demonstrate that eliminating senescent cells can improve both healthspan and lifespan[4–6]. However, a major barrier to this approach is the limited understanding of which cell types undergo senescence during aging and the lack of specific biomarkers to distinguish senescent cells from other proinflammatory populations, particularly macrophages.

Macrophages are highly versatile innate immune cells that adopt diverse polarization states in response to their microenvironment. Notably, they share multiple hallmark features of senescent cells, including inflammatory cytokine secretion, metabolic reprogramming, immunomodulatory metabolite production, increased cell size and transient expression of senescence-associated markers such as *Cdkn1a* (p21), *Cdkn2a* (p16), senescence-associated secretory phenotype (SASP) factors and senescence-associated β-galactosidase (SA-β-gal) activity[7,8]. This phenotypic overlap raises a fundamental unresolved question: can macrophages undergo bona fide cellular senescence, and if so, how can senescent macrophages be distinguished from activated proinflammatory (M1) or anti-inflammatory (M2) states? Although accumulating evidence links macrophage senescence to age-related inflammation and disease, many studies rely on limited, reversible and context-dependent markers, thereby complicating interpretation[9–18]. As a result, defining macrophage senescence remains a formidable challenge and a major unmet need in the field.

✉e-mail: AJCovarrubias@mednet.ucla.edu

Unlike other immune cells, tissue-resident macrophages are radi-oresistant, metabolically flexible and often long-lived, particularly in their tissue niche[19–22]. These features enable survival in highly stressful microenvironments but may also predispose macrophages to with-stand cumulative DNA damage and metabolic stress, rendering them susceptible to senescence with age. Here we test the hypothesis that macrophages represent a major source of senescent cells in aging tis-sues and act as key drivers of inflammaging.

The rising prevalence of global aging in parallel with an obesity epidemic underscores the need to investigate the role of senescent macrophages in driving chronic inflammation and metabolic dysfunc-tion associated with aging[23]. Our study identifies DNA damage and excessive cholesterol accumulation as critical contributors to mac-rophage senescence, both in vitro and in vivo. Importantly, we identify key biomarkers to better detect and quantify macrophage senescence and demonstrate that targeting senescent macrophages may represent a promising therapeutic strategy to attenuate inflammation in the aged liver and in patients with metabolic dysfunction-associated steatotic liver disease (MASLD).

## Results

### Exogenous DNA damage drives *Cdkn1a* (p21)⁺ senescence in macrophages in vitro

To test whether macrophages can undergo an irreversible and stable form of cellular senescence, mouse bone marrow-derived macrophages (BMDMs) were subjected to ionizing irradiation (IR) or doxorubicin (Doxo) (Fig. 1a). Ten days after DNA damage, we observed a decrease in the proliferation marker Ki-67 and Lamin B1, a nuclear protein whose expression decreases with senescence (Fig. 1b). We also observed increased pNF-κB, which likely results from the acquisition of cellular senescence and not activation of the acute DNA damage response, because it occurs post gain and loss of pH2A.X[24]. In support, we found that both IR and Doxo increased cell size and elevated SA-β-gal activ-ity, leaving cells arrested in the G2–M phase, accompanied by >4 N genomes (Fig. 1c–e). DAPI imaging showed that control macrophages had a uniform nuclear size and shape, whereas IR- and Doxo-treated macrophages were larger and irregularly shaped (Fig. 1f). Machine learning and neural network tools have been deployed to identify senescent cells from nonsenescent cells using nuclear imaging derived from DAPI-stained cells[25]. Using this approach, we found that the prob-ability of IR-treated macrophages being senescent is greater than that for control cells (Fig. 1g). To further examine this, macrophages were stained with a CellTrace Violet (CTV) dye before IR. Interestingly, ~20% of labeled cells divided 1 day post IR, but all cells stopped dividing between 5 and 10 days, suggesting a stable cell-cycle arrest phenotype (Fig. 1h). Collectively, our data show senescent macrophages acquire multiple hallmarks of senescence, including permanent cell-cycle arrest, 10 days post IR or Doxo, and they are referred to here as Sen(IR) or Sen(Doxo), respectively.

Senescent cells are characterized by increased expression of cell-cycle inhibitor genes, such as *Cdkn1a* (p21) and *Cdkn2a* (p16), as well as SASP genes. Accordingly, we measured the expression lev-els of these genes in Sen(IR) and Sen(Doxo) macrophages. We found that *Gdf15*, *Mmp9* and *Cdkn1a*, but not *Cdkn2a*, were upregulated in senescent macrophages relative to control (Fig. 1i). Of note, most studies investigating senescent macrophages have used expression of *Cdkn2a* (p16) as a positive indicator of macrophage senescence or p16 reporter mice to identify and target these cells for clearance[13,14,17,18]. To investigate whether p16 expression can be used to target senescent macrophages, we obtained BMDMs from a transgenic mouse model that contains a tri-modality reporter (3MR) on the p16 locus (p16-3MR)[26]. One of the three reporters drives apoptosis via ganciclovir (GCV) treatment. Interestingly, we found that p16-3MR senescent macrophages were resistant to cell death and apoptosis, while control macrophages were more sensitive to GCV (Extended Data Fig. 1a–c). These data suggest that senescent macrophages downregulate p16, relative to control cells, sparing them from apoptosis. In fact, in a previ-ous study[27], we found that in Doxo-treated p16-3MR mice, GCV reduced *Cdkn2a* mitochondrial RNA expression in the liver but not *Cdkn1a*. Therefore, p16⁺ and p21⁺ senescent cells can be distinct populations in aging tissues, as supported by recent studies showing divergent SASP programs and trajectories for p16⁺ versus p21⁺ senescent cells across tissues[28,29]. In addition to p21, we also discovered that senescent mac-rophages express canonical markers of M1 and M2 macrophages, which highlights the challenge of distinguishing senescent and nonsenescent macrophages (Fig. 1i). However, using transcriptomics, proteomics and metabolomics, we found senescent macrophages formed unique clusters from M0 (a naive state), M1 and M2 conditions suggesting that senescent macrophages are distinct from the classic M1 and M2 paradigm (Fig. 1j–l).

Using this model, we interrogated proposed mechanisms reg-ulating senescence in macrophages. To start, interleukin-4 (IL-4), a type 2 (T helper 2) cytokine, is proposed to be protective against macrophage senescence through a DNA damage repair mechanism[17] similar to the effects of lipopolysaccharide (LPS), which can pro-tect against DNA damage[27]. To test whether LPS or IL-4 can protect against DNA damage-induced senescence, BMDMs were activated with LPS or IL-4 before IR (Extended Data Fig. 1d). Ten days post IR, IL-4 or LPS pretreatment did not significantly prevent *Cdkn1a* and *Cdkn2a* expression, prevent cell-cycle arrest or decrease SA-β-gal activity (Extended Data Fig. 1e–g). Next, CD38 has been reported to drive macrophage senescence through heightened NAD⁺ consump-tion[18]. Interestingly, we also observed heightened *Cd38* mRNA expres-sion and CD38-mediated NADase activity in senescent macrophages (Extended Data Fig. 1h,i). Therefore, we hypothesized that elevated CD38 activity may drive senescence as recently proposed[18]. To test this, *Cd38* knockout (KO) BMDMs were subjected to the Sen(IR) and Sen(Doxo) models. However, we found no difference in SA-β-gal activity

**Fig. 1 | Irradiation and Doxo drive p21⁺ macrophage senescence in vitro.**
**a**, The in vitro model system using exogenous DNA damage with irradiation and Doxo from male C57BL/6J mice. **b**, SDS–PAGE gels and immunoblotting (western blot) with ImageJ quantification relative to loading control. **c**, Representative phase contrast images as well as cell volume quantification via flow cytometry. Average ± s.e.m. cell volume (pl) of *n* = 4 biological replicates. *P* value derived from ANOVA. **d**, SA-β-Gal images and average ± s.e.m. percent positive cell quantification by microscopy at ×20 magnification. *n* = 3 biological replicates. *P* value from ANOVA. **e**, Click-iT EdU labeling with propidium iodide (PI) labeling of cells to assess cell-cycle dynamics. Axes represent the percentage of cells positive for the stain, compared to the parental gate, as detected by flow cytometry. **f**, Representative images of control and senescent macrophages, DAPI-stained nuclei at ×20 magnification. **g**, Probabilistic deep-learning algorithm to generate a predicted senescence score. Average score ± 95% confidence interval, of 100 biologically independent DAPI images. *P* value derived from a two-sided *t*-test. **h**, CTV staining 24 h post IR. Dilution of CTV was performed at the represented time points via flow cytometry. Percent cells that diluted CTV from the previous day were quantified and trendlines are shown (right). **i**, Mean ± s.e. mRNA transcript levels (reverse transcription-quantitative polymerase chain reaction (RT–qPCR)) normalized to control. *n* = 3 biological replicates. *P* value derived from Tukey's test post-ANOVA analysis. **j**, PCA on bulk RNA-seq samples from day 7 BMDMs either unactivated (M0), or treated with 12 h of 100 ng ml⁻¹ LPS (M1) or 18 h of 10 ng ml⁻¹ IL-4 (M2), compared to senescent macrophages (Sen(IR), 10 days post irradiation) and (control) unactivated macrophages cultured for 10 days alongside senescent macrophages. **k**, PCA projections from senescent macrophages (Sen(IR), 10 days post irradiation), passage control condition (control) and passage control plus LPS (M1) or IL-4 (M2). **l**, PCA analysis from senescent macrophages (Sen(IR), 10 days post irradiation), passage control condition (control) and passage control plus LPS (M1) or IL-4 (M2). BMDMø, bone marrow-derived macrophage; p(senecence), predicted senescence. Panel **a** created in BioRender; Salladay-Perez, I. https://biorender.com/6umuckd (2026).

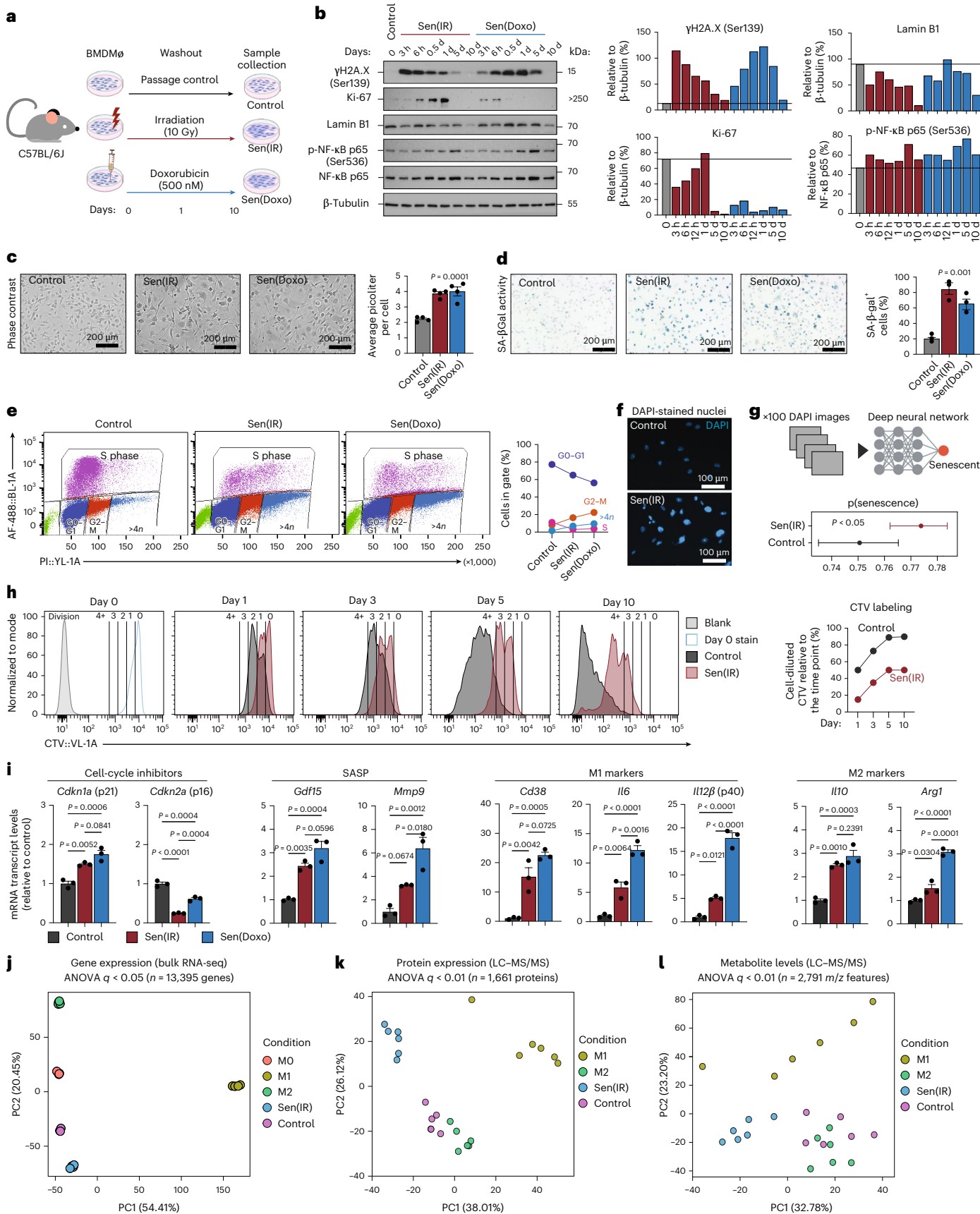

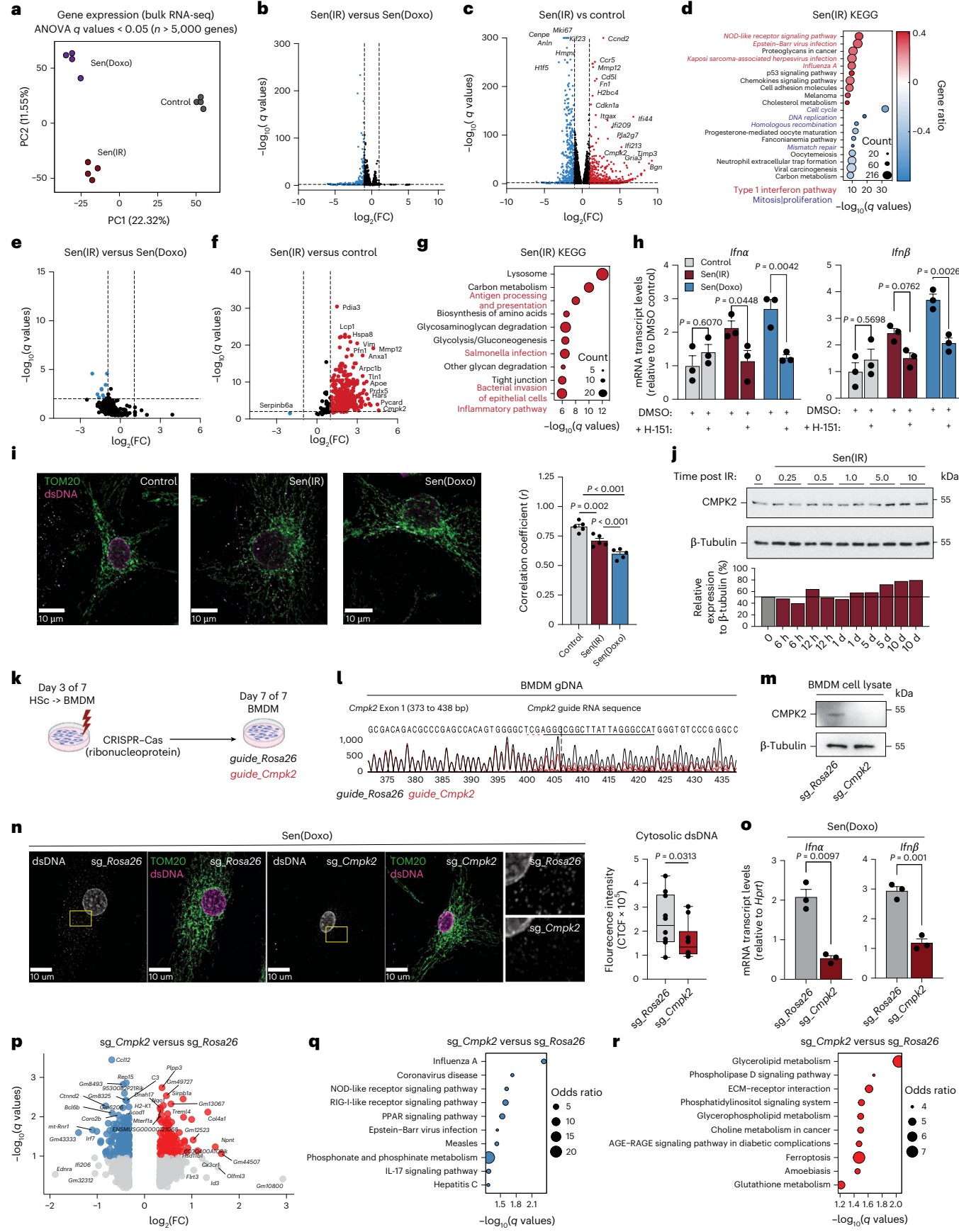

**Fig. 2 | Senescent macrophages are characterized by a SASP and a type I IFN response to mtDNA. a**, PCA on bulk RNA-seq samples. **b,c**, Volcano plots of differentially expressed genes ($\log_2$(FC) > 1.3 and FDR < 0.05) between Sen(IR) versus Sen(Doxo) (**b**) and Sen(IR) versus control (**c**). Adjusted $P$ values are derived from Fisher's exact test followed by FDR adjustment. **d**, KEGG terms associated with proteins upregulated or downregulated by both treatments. The red font highlights pathways related to a type I IFN signal. The blue font highlights terms related to the cell cycle. **e,f**, Volcano plot of upregulated (red) and downregulated (blue) proteins by FDR-adjusted $P$ value and FC from SASP-proteomics mass spectrometry between Sen(IR) versus Sen(Doxo) (**e**) and between control and senescent macrophages (**f**). **g**, KEGG terms associated with the proteins upregulated or downregulated by both treatments. The red font highlights terms related to an inflammatory pathway. Adjusted $P$ values are derived from Fisher's exact test followed by FDR adjustment. **h**, Mean ± s.e.m. mRNA transcript levels, relative to control, for type I IFN response genes in response to STING inhibitor (H-151). $P$ value of ANOVA test of $n$ = 3 biological replicates. **i**, Immunostaining for mitochondria (TOM20) and co-localized dsDNA (red). Average Pearson's correlation coefficients ± s.e.m. were calculated for $n$ = 5 biological replicates. **j**, SDS–PAGE gel immunoblotting (western blot) of macrophages in response to days post IR. ImageJ quantification relative to loading controls are shown to the right. **k**, Illustration of CRISPR–Cas9 gene editing of differentiating

hematopoietic stem cells into macrophages. **l**, Sanger sequencing base calling results from CRISPR–Cas9 editing of *CMPK2* locus exon 1. **m**, SDS–PAGE gels and immunostaining (western blot) for CMPK2 protein in response to edits sg_*Rosa*26 and sg_*Cmpk2* edits via CRISPR–Cas9. **n**, Immunostaining for mitochondria (TOM20) and co-localized dsDNA (red). To the right is the quantification of cytosolic dsDNA corrected total cell fluorescence (CTCF) intensity. Box-and-whisker plots show the median (center line), interquartile range (25th to 75th percentiles; box), and minimum and maximum values (whiskers). $n$ = 10 biological replicates, $P$ values were calculated using a one-tailed $t$-test. **o**, Mean mRNA transcript levels ± s.e.m. relative to control of *Ifnα* and *Ifnβ* in Sen(Doxo) macrophages with edits on the *Rosa* or *Cmpk2* locus. $n$ = 3 biological replicates were used to derive the $P$ value of the $t$-test. **p**, Volcano plots of differentially expressed genes ($\log_2$(FC) > 1.3 and FDR < 0.1) between sg_*Rosa*26 and sg_*Cmpk2* edited Sen(Doxo) macrophages. Adjusted $P$ values are derived from a two-sided $t$-test followed by FDR adjustment. **q**, KEGG pathway analysis on downregulated DEGs. Blue indicates downregulation. The size of the circles represents the odds ratio. Adjusted $P$ values are derived from Fisher's exact test followed by FDR adjustment. **r**, KEGG pathway analysis on upregulated DEGs. Red indicates upregulation. The size of the circles represents the odds ratio. Adjusted $P$ values are derived from Fisher's exact test followed by FDR adjustment.

or SASP gene expression in *Cd38* KO senescent macrophages compared to wild type (WT) (Extended Data Fig. 1j,k). Moreover, we did not observe any difference in SA-β-gal activity, senescent cell burden or inflammatory cytokine expression in visceral adipose tissue (VAT) from aged (24 months) WT versus *Cd38* KO mice, suggesting that targeting CD38 does not protect against the induction of senescence or the natural aging senescent cell burden (Extended Data Fig. 1l,m).

Aging bone marrow has been shown to acquire cell-intrinsic epigenetic modifications that affect macrophage polarization[30,31]. Therefore, we next examined whether bone marrow age influences BMDM senescence. To test this, BMDMs from young and aged male mice were used in our Sen(IR) model. It was observed that the age of the bone marrow had no major influence in modulating the hallmark features of senescence (Extended Data Fig. 2a–c). Collectively, these data suggest that our macrophage senescence model is stable and reproducible, and that macrophage senescence cannot be prevented or reversed by T helper 2 cell signaling, LPS activation, CD38 NADase activity and the age of the bone marrow.

### Senescent macrophages are characterized by a SASP and sterile type I interferon signaling

To identify the specific transcriptomic signature of senescent macrophages, we performed bulk RNA sequencing (RNA-seq). We found that both Sen(IR) and Sen(Doxo) macrophages share most of their transcriptomic phenotype (50%, 747 genes) (Fig. 2a,b and Extended Data Fig. 3a). Interestingly, the top upregulated genes are related to senescence, SASP and inflammation (*Cdkn1a*, *Ccnd2*, *Ccr5*, *Cmpk2*, *Fn1*, *Mmp9*, *Mmp12*,

*Mmp13*, *Pla2g7*, *Timp3*, as well as interferon (IFN)-induced (*Ifi*) genes like *Ifi44*, *Ifi209* and *Ifi213*) (Fig. 2c and Extended Data Fig. 3b). A set of downregulated genes was also identified that included *Mki-67* and *Cdkn2a* (Fig. 2c and Extended Data Fig. 3b). Kyoto Encyclopedia of Genes and Genomes (KEGG) analysis revealed that senescence-related genes are associated with inflammation, whereas genes downregulated in senescence are linked to the cell cycle (Fig. 2d and Extended Data Fig. 3c). The occurrence of inflammation and IFN-signaling pathways led us to hypothesize that senescent macrophages also acquire a SASP. To test this, conditioned media were subjected to proteomics. Few secreted proteins differed between Sen(IR) and Sen(Doxo); however, both senescent conditions secreted markedly more SASP proteins than controls, with MMP12, APOE, LGALS3 and CMPK2 among the most abundant (Fig. 2e,f and Extended Data Fig. 3d). KEGG pathway enrichment also highlighted a signature related to inflammation and metabolism (Fig. 2g and Extended Data Fig. 3e) with 80% (293 proteins) of the total SASP proteome in common (Extended Data Fig. 3f). Both the transcriptome and SASP proteome analysis suggest senescent macrophages upregulate inflammatory pathways related to a type I IFN (*Ifnα* or *Ifnβ*) response and undergo metabolic reprogramming suggesting an active response toward pathogens despite being in sterile conditions.

Sterile chronic inflammation related to macrophage-specific IFN responses has been observed in aging tissues and liver failure[32]. The mechanisms underlying age-related type I IFN activation involve the cytosolic DNA-sensing pathway, cyclic GMP–AMP synthase-stimulator of interferon genes (cGAS-STING)[33,34]. To assess cGAS-STING signaling in senescent macrophages, cells were treated with the STING

**Fig. 3 | Senescent p21⁺ macrophages accumulate in aged metabolic tissues. a**, Mean ± s.e. mRNA transcript levels (RT–qPCR) of *Cdkn1a* from $n$ = 8, 2–4-month-old (Y) versus $n$ = 8 21–24-month-old (O) male mice. $P$ value derived from a two-sided $t$-test. **b**, SA-β-gal histology of Y and O liver sections. $n$ = 2 male mice. Scale bar represents a 20 μm distance. **c**, Immunostaining for macrophages (F4/80) and p21 expression in Y and O male sections. Scale bar represents a 20 μm distance. **d**, Quantification of *Cdkn1a*⁺ cell accumulation with age in heart and liver datasets. $P$ values represent significance from Pearson's correlation test. **e**, Linear regression rates of tissue-specific *Cdkn1a*⁺ cell accumulation with age across six tissues in the TMS database. **f**, UMAP analysis of the Tabula Muris Senis dataset representing all cells in the liver annotated by cell type. **g**, UMAP analysis of the Tabula Muris Senis dataset representing all cells in the liver annotated by age. **h**, Percentage of cells positive for *Cdkn1a* (green), *Cdkn2a* (red) and double positive (blue) across liver Kupffer cells, hepatocytes and endothelial cells. **i**, Bubble map of genes grouped by hallmarks or macrophage senescence. Size represents the fraction of cells expressing each gene, and color reflects average

expression for endothelial cells, hepatocytes and Kupffer cells. **j**, Venn diagram analysis using Fisher's test with aged Kupffer cells and senescent IR macrophages. **k**, Bulk RNA-seq of 67 genes found enriched in senescence and aging. Conditions reflect different polarized states. Box outlines are manually placed, and highlighted clusters are generally downregulated compared to other conditions. The $z$-score reflects the FC of the condition over the control group. **l**, GSVA scoring of M1, M2 and Sen(IR) conditions using the SenMayo gene signature and the 67 genes identified in **j** as the MSen signature. Box-and-whisker plots display the median (center line), interquartile range (25th to 75th percentiles; box), and minimum and maximum values (whiskers). $n$ = 4 biological replicates were analyzed. $P$ values reflect post-hoc comparisons following one-way ANOVA. NS, not significant ($P ≥ 0.05$); *$P < 0.05$; **$P < 0.001$; ***$P < 0.001$. **m**, Distribution of cells expressing the MSen signature across all cell types in the Tabula Muris Senis dataset via a Seurat score. **n**, Distribution of MSen Seurat score across young and old Kupffer cells. $P$ value represents results of a Student's $t$-test. m, month; NK, natural killer.

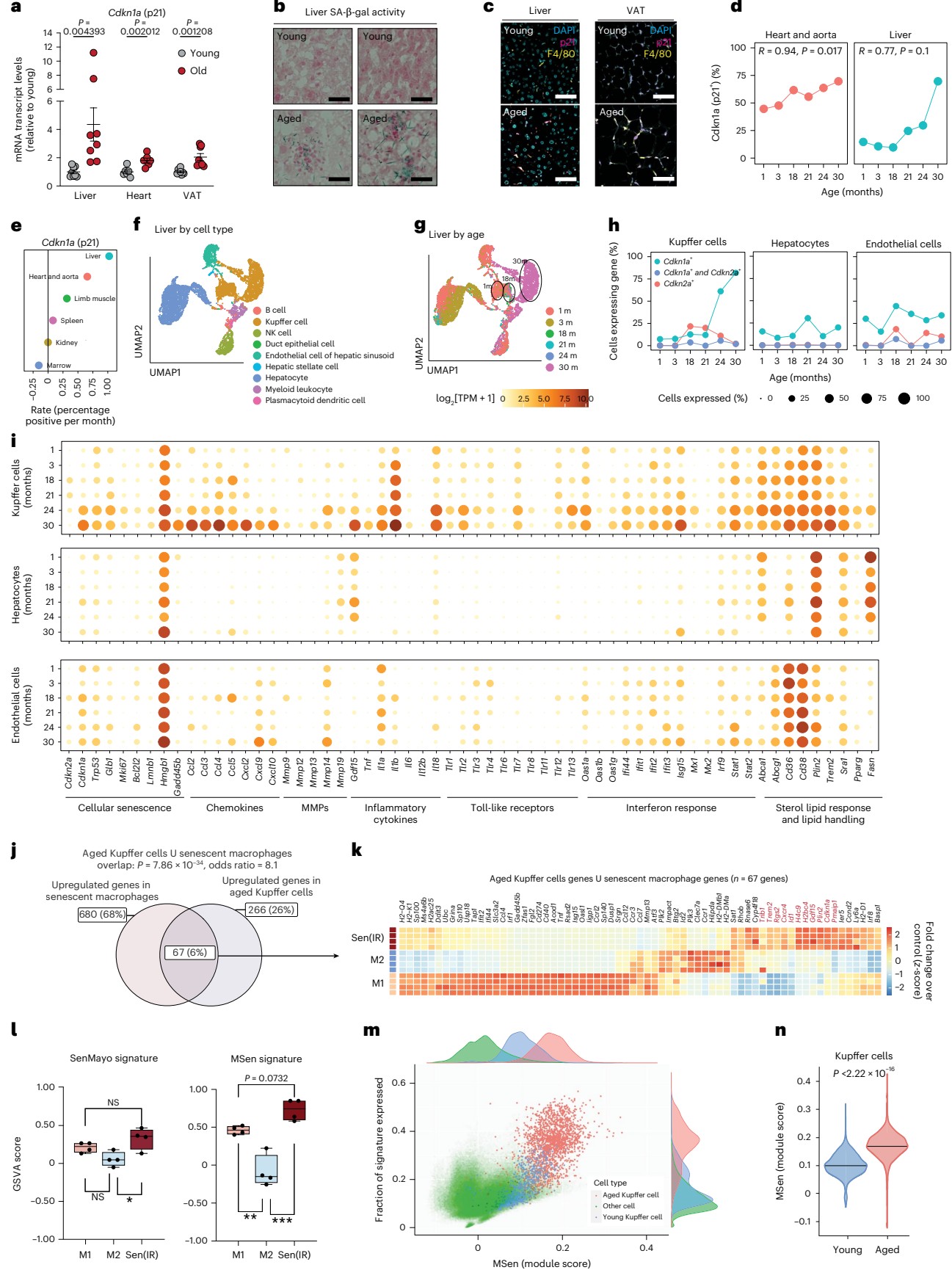

inhibitor (H-151). H-151 significantly reduced *Ifnα* and *Ifnβ* expression compared to DMSO controls (Fig. 2h). In addition, during senescence double-stranded DNA (dsDNA) re-localized from the mitochondria into the cytosol, suggesting mitochondrial DNA release as the source of cGAS-STING activation (Fig. 2i). To understand the mechanism, we interrogated the activity of CMPK2, a cytidine/uridine monophosphate kinase, that is upregulated in our omics datasets and a known regulator of inflammation via newly synthesized mtDNA (Fig. 2c,f)[35–37]. We found that *Cmpk2* gene and protein expression was associated with senescence and not an acute DNA damage response (Fig. 2j and Extended Data Fig. 3g). Therefore, we hypothesized that elevated CMPK2 activity in senescent macrophages may help drive activation of the type I IFN response. To test this, CRISPR–Cas9-based genome editing was used to disrupt the *Cmpk2* or *Rosa26* locus as a control (Fig. 2k). Sanger sequencing confirmed disruption leading to loss of CMPK2 protein expression, as validated by western blot (Fig. 2l,m). *Cmpk2*- and *Rosa26*-edited macrophages were subjected to senescence, which resulted in reduced total mitochondrial and cytosolic dsDNA as well as downregulated *Ifnα* and *Ifnβ* mRNA expression in the *Cmpk2*-edited senescent macrophages (Fig. 2n,o). Next, we performed RNA-seq on *Cmpk2*- and *Rosa26*-edited senescent macrophages. In senescent macrophages, *Cmpk2* loss reduced type I IFN-related genes (*Irf7*, *Acod1*, *Ifit206*) and decreased enrichment of KEGG pathways such as Influenza A, NOD-like receptor signaling and Epstein–Barr virus infection (Fig. 2p,q). By contrast, genes involved in lipid metabolism were upregulated (Fig. 2p,r).

Based on the elevated IFN signaling observed in senescent macrophages, we hypothesized that they are primed toward a proinflammatory phenotype. To test this, we exposed senescent macrophages to LPS and observed a heightened gene expression response as well as an increased capacity for phagocytosis (Extended Data Fig. 3h,i). Conversely, the ability to clear apoptotic cells via efferocytosis was significantly reduced compared to control macrophages (Extended Data Fig. 3j). Upon IL-4 treatment, senescent macrophages increased expression of *Cd36* and *Ym1*, but reduced expression of other hallmark M2 genes, along with diminished arginase-1 (*Arg1*) expression and enzymatic activity (Extended Data Fig. 3k,l). Therefore, our data demonstrated that senescent macrophages exhibit elevated IFN signaling, via activation of a CMPK2–mtDNA–cGAS-STING-dependent mechanism, and although they can still integrate both M1 and M2 polarization signals, they are biased toward a proinflammatory activation state.

### p21⁺ senescent macrophages accumulate in the aged liver

Next, we wanted to investigate whether macrophages naturally undergo senescence in vivo during the aging process. We focused on metabolic tissues, because these tissues acquired a high senescent cell burden during aging[27]. In fact, our analysis showed that among visceral tissues, the liver showed the largest increase in *Cdkn1a* (p21) expression, while the heart and VAT showed smaller but significant increases (Fig. 3a). The aged liver and VAT also displayed elevated SA-β-gal activity and an

increase in the amount of p21⁺ F4/80⁺ macrophages relative to young liver and VAT (Extended Data Fig. 1l and Fig. 3b,c). Notably, *Cdkn2a* (p16) mRNA expression also increased across tissues tested, potentially reflecting inflammation or senescence in non-macrophage cell types (Extended Data Fig. 4a). To further investigate the relationship between p21 expression and increased SA-β-gal activity, we analyzed the Tabula Muris Senis dataset, which includes single-cell RNA-seq data from 23 tissues across the murine lifespan[38]. Interestingly, the liver also exhibited the greatest increase and highest proportion of *Cdkn1a*⁺ cells with age (Fig. 3d,e). Based on these findings, we focused our analysis on the liver.

Kupffer cells, resident macrophages in the liver, are the most abundant hepatic immune cell population and our analysis revealed that the transcriptomic profile of Kupffer cells undergoes profound aging-dependent changes (Fig. 3f,g). Unlike other liver cell types, such as hepatocytes, endothelial cells and other immune cell populations, Kupffer cells exhibited marked age-dependent upregulation of *Cdkn1a* expression (Fig. 3h and Extended Data Fig. 4b). We next explored the expression of key senescence-associated genes, including chemokines, matrix metalloproteinases (MMPs), IFN and lipid-handling genes, across all major liver cell types. Notably, among these populations, only Kupffer cells were consistently enriched for hallmark senescence genes with age, including a pronounced inflammaging signature (Fig. 3i and Extended Data Fig. 4c). By contrast, in aged cardiac tissue a similar pattern was observed in the myeloid leukocyte population, although increased *Cdkn1a* expression was not limited exclusively to myeloid cells (Extended Data Fig. 4d,e).

Comparative transcriptomic analysis of aged Kupffer cells and in vitro Sen(IR) and Sen(Doxo) macrophages identified 67 commonly expressed genes, defining a macrophage senescence (MSen) signature (Fig. 3j). While containing M1 and M2 genes, the set also includes senescence-specific genes that may distinguish senescent macrophages from other subsets (Fig. 3k). Using Gene Set Variation Analysis (GSVA), the MSen transcriptomic signature identified senescent macrophages among M1 and M2 macrophages in a manner that outperformed the SenMayo[39], CellAge[40] and Senescence Gene Ontology signatures (Fig. 3l and Extended Data Fig. 5a). The same MSen signature was applied across all cell populations and tissues in the Tabula Muris Senis and was highly selective for identifying aged, but not young, Kupffer cells, or other cell types (Fig. 3m,n). Collectively, the data indicate that tissue-resident macrophages, Kupffer cells, represent a major pool of senescent cells in aged livers and possibly other metabolic tissues.

### Senescent macrophages form lipid droplets and express TREM2

To understand the physiological mechanisms driving macrophage senescence during aging, we analyzed transcriptomic and proteomic datasets from aged Kupffer cells and in vitro senescent macrophages. Both populations showed increased expression of genes involved in cholesterol metabolism, along with elevated levels of markers

---

**Fig. 4 | Senescent macrophages are TREM2⁺p21⁺ and form lipid droplets.**
**a**, Protein expression (LC–MS/MS) projected as a *z*-score of log₂(FC) over control conditions. Clustering represents the three major clades of a dendrogram. **b**, log₂-transformed raw protein expression for selected lipid-laden proteins. Box-and-whisker plots display the median (center line), interquartile range (25th to 75th percentiles; box), and minimum and maximum values (whiskers). *n* = 6 biological replicates were analyzed. Statistical *P* values reflect results from a one-way ANOVA identifying differences in at least one group. **c**, KEGG pathway enrichment for all proteins upregulated in Sen(IR) conditions over control macrophages. **d**, Immunostaining by microscopy for lipid-body labeling (green-LipoTOX) in Sen(IR) conditions. Green arrows represent LipoTOX+ cells. **e**, Bodipy (lipid droplet) staining for represented conditions. Total histograms are normalized to mode. **f**, SDS–PAGE gels and immunoblotting (western blot)

for TREM2 (both glycosylated and total), p21 and p16. ImageJ quantification for band intensity is shown to the right and represents relative intensity over the loading control. **g**, PCA on bulk RNA-seq samples from senescent macrophages (Sen(IR) (10 days post irradiation)) and passage control condition (Control) on a WT or *Trem2*⁻/⁻ background. **h**, Gene Ontology analysis for statistically significant downregulated DEGs in senescence in response to the *Trem2*⁻/⁻ background. The size of the data points represents the relative odds ratio. **i**, MSen scoring of bulk RNA-seq samples using GSVA methods. Box-and-whisker plots display the median (center line), interquartile range (25th to 75th percentiles; box), and minimum and maximum values (whiskers). *n* = 3 biological replicates were analyzed. *P* values represent results from Student's *t*-test. **j**, Scaled transcriptomic expression of the 67 MSen genes broken into the four major clades of a dendrogram.

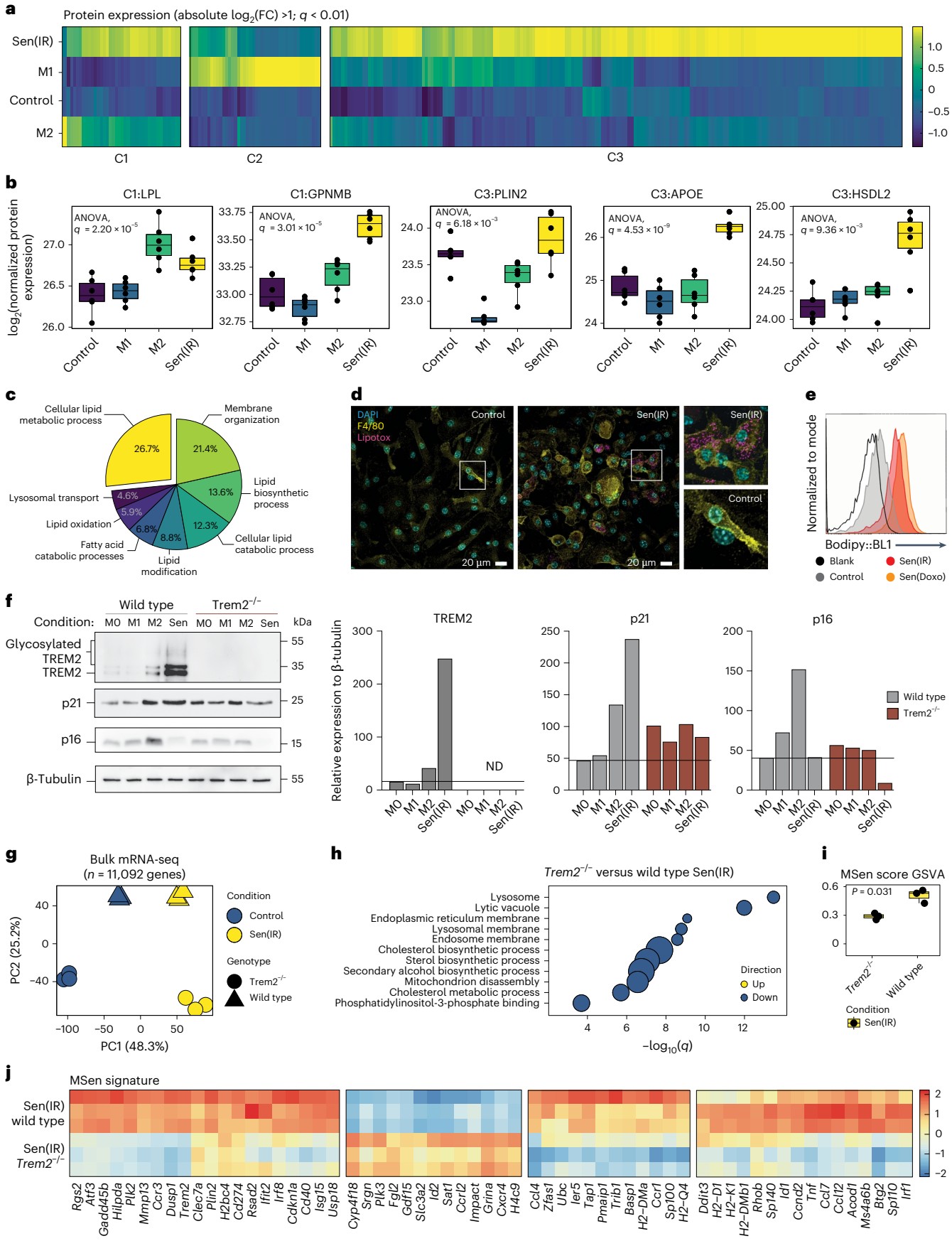

**a** Protein expression (absolute log₂(FC) >1; q < 0.01)

**b** C1:LPL, C1:GPNMB, C3:PLIN2, C3:APOE, C3:HSDL2

**c** Cellular lipid metabolic process 26.7%, Membrane organization 21.4%, Lipid biosynthetic process 13.6%, Cellular lipid catabolic process 12.3%, Lipid modification 8.8%, Fatty acid catabolic processes 6.8%, Lipid oxidation 5.9%, Lysosomal transport 4.6%

**d** DAPI, F4/80, Lipotox

**e** Bodipy::BL1; Blank, Control, Sen(IR), Sen(Doxo)

**f** TREM2, p21, p16

**g** Bulk mRNA-seq (n = 11,092 genes)

**h** Trem2⁻/⁻ versus wild type Sen(IR)

**i** MSen score GSVA

**j** MSen signature

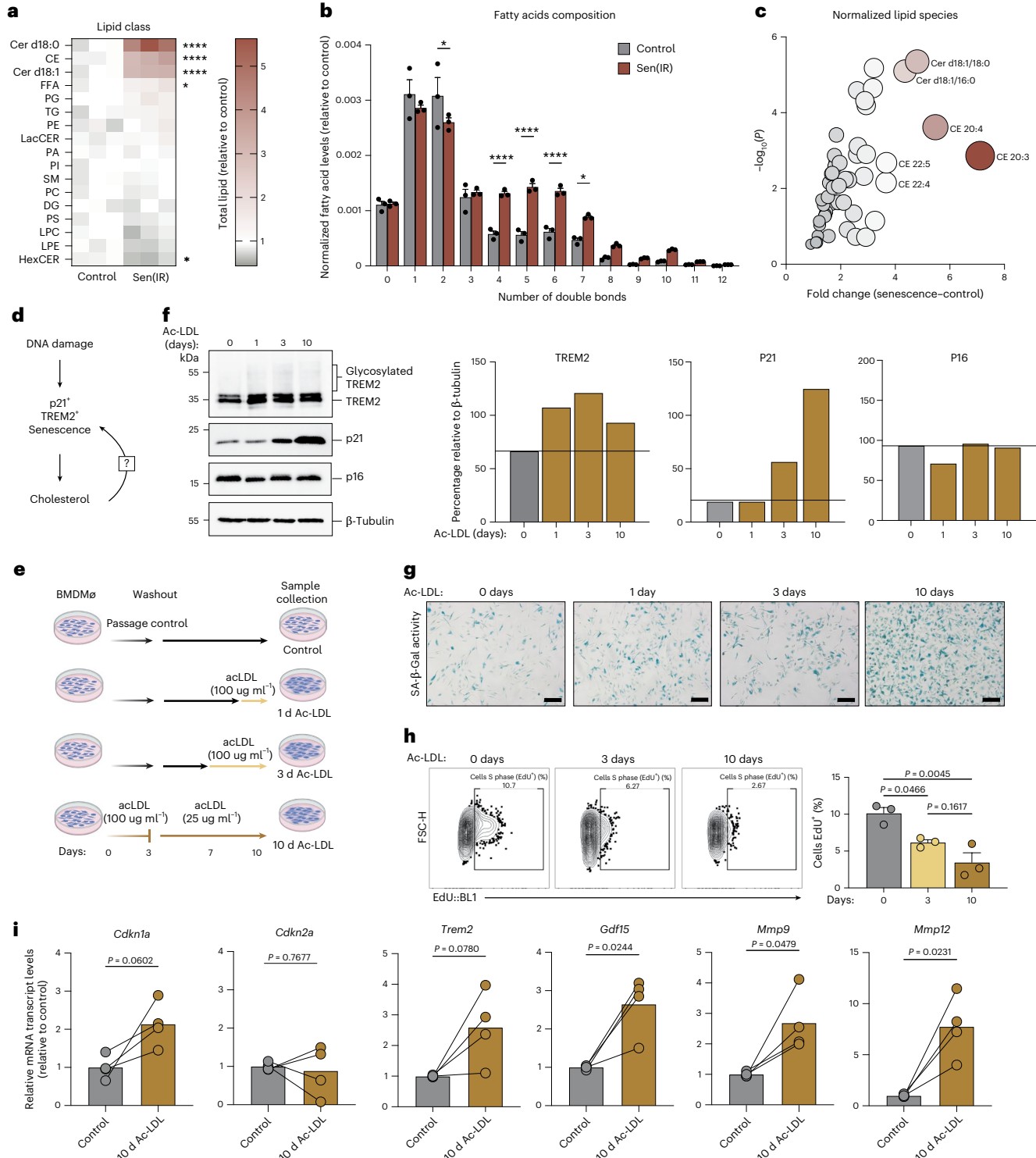

**Fig. 5 | Excess cholesterol ester loading via Ac-LDL is a driver of macrophage senescence in vitro. a**, Heatmap representation of fold change differences of normalized lipid abundance (nmol lipid per $1 \times 10^6$ cells per average cell size) for all lipid species via shotgun lipidomics (LC–MS/MS) in Sen(IR) macrophages relative to control macrophages. Asterisks represent the $P$ value of the ANOVA: ns, ≥0.05; *$P < 0.05$; **$P < 0.01$; ***$P < 0.001$; ****$P < 0.0001$. **b**, Normalized fatty acid levels (nmol fatty acid per $1 \times 10^6$ cells per average cell size) relative to control macrophages via shotgun lipidomics (LC–MS/MS). Bar: average ± s.e.m. of $n = 3$ biological replicates. Asterisks represent the $P$ value of the $t$-test: ns, ≥ 0.05; *$P < 0.05$; **$P < 0.01$; ***$P < 0.001$; ****$P < 0.0001$. **c**, Volcano plot of the most enriched lipid species in senescence relative to control macrophages. The color and size of the data points represent the FC gradient. **d**, Model illustration hypothesizing cholesterol as a driver of macrophage senescence.

**e**, Model illustration of an in vitro model system of Ac-LDL-induced macrophage senescence. **f**, SDS–PAGE gels and immunostaining (western blot) for TREM2 (both glycosylated and total), p21 and p16 in response to more time in Ac-LDL. ImageJ quantification for band intensity is shown to the right and represents the relative intensity over the loading control. **g**, SA-β-gal assay imaging at ×20 magnification. Scale bar: 10 μm. **h**, Click-iT EdU labeling of Ac-LDL-treated cells to assess cell-cycle dynamics. The average ± s.e.m. cell fraction in S phase (EdU⁺) is shown to the right. $P$ value derived from an ANOVA. $n = 3$ biological replicates were used. **i**, Mean ± s.e.m. mRNA transcript levels relative to control for hallmark senescence genes. $P$ value of paired $t$-tests from $n = 4$ biological independent experiments. Panel **e** created in BioRender; Salladay-Perez, I. https://biorender.com/6umuckd (2026). BMDMø, bone marrow-derived macrophage.

characteristic of lipid-laden, disease-associated macrophages, including *Trem2* and *Plin2* (Figs. 2d and 3i,k). These findings led us to hypothesize that senescent macrophages constitute a subset of lipid-laden macrophages implicated in metabolic diseases, including MASLD and atherosclerosis. To evaluate this, we analyzed five public datasets[41–45] and observed a significant upregulation of the MSen gene signature in Kupffer cells from MASLD livers and in atherosclerotic plaques (Extended Data Fig. 5b–f). Furthermore, deep proteomic analysis revealed that senescent macrophages upregulate several proteins related to M2 (cluster 1), M1 (cluster 2) and exclusively in the C3 cluster (Fig. 4a). Investigation into the clusters reveals the upregulation of lipid-related proteins compared to control and M1 conditions (Fig. 4b). Pathway enrichment into the C3 cluster highlights lipid remodeling terms with 26.7% of proteins perturbed (Fig. 4c). Consistent with this, we found that senescent macrophages exhibited greater lipid droplet accumulation compared to control macrophages (Fig. 4d,e).

TREM2 macrophages have been implicated in several diseases like metabolic dysfunction, chronic inflammation and tissue degeneration, including metabolic dysfunction-associated steatohepatitis (MASH) and neurodegenerative diseases such as Alzheimer disease[46–49]. Interestingly, TREM2 is highly upregulated in Sen(IR) macrophages compared to M0, M1 and M2 macrophages, with glycosylated TREM2 suggesting membrane localization (Fig. 4f). In addition, we observed that the simultaneous activation of Trem2, p16 and p21 is associated with M2 anti-inflammatory macrophages, whereas the upregulation of p21, TREM2, and downregulation of p16 is a unique signature of senescent macrophages (Fig. 4f). Furthermore, *Trem2*$^{-/-}$ senescent macrophages had reduced p21 expression, suggesting that TREM2 plays an active role in regulating senescence (Fig. 4f).

To determine the functional role of TREM2, *Trem2*$^{-/-}$ mice were used to evaluate hallmark features of senescence. SA-β-gal activity and cell-cycle arrest phenotypes were unchanged in response to the loss of TREM2 (Extended Data Fig. 6a,b), but a profound change in morphology was observed, and proliferation rates were also attenuated in the *Trem2*$^{-/-}$ background (Extended Data Fig. 6b). Efferocytosis was reduced in WT senescent macrophages, consistent with reports that TREM2 shedding impairs TREM2-dependent efferocytosis in chronically activated macrophages[48] (Extended Data Fig. 3j). To define the global impact of TREM2 on macrophage senescence, we performed unbiased transcriptomic analysis. Principal components analysis (PCA) revealed distinct clustering across all conditions, with the first principal component (PC1) separating control and senescent cells and the second principal component (PC2) reflecting genotype differences (Fig. 4g). Pathway analysis of downregulated genes in *Trem2*$^{-/-}$ senescent macrophages highlighted autophagy-related cellular compartments and cholesterol metabolism, while upregulated genes were enriched for focal adhesion, translation and actin cytoskeleton remodeling (Fig. 4h and Extended Data Fig. 6c). We next asked whether *Trem2* regulates the MSen transcriptomic signature developed

in Fig. 3k. Interestingly, the loss of TREM2 downregulates the MSen gene score and prevents the expression of key MSen-related genes, including *Cdkn1a*, p21 (Fig. 4i,j). Collectively, these data suggest that TREM2 plays a key role in regulating p21, SASP gene expression and lipid biology in senescent macrophages.

## Excess cholesterol ester loading via acetylated low-density lipoprotein is a driver of macrophage senescence in vitro

To characterize the lipid species in senescent macrophages, we performed shotgun lipidomics and identified ceramides (Cer d18:0 and Cer d18:1) and cholesterol esters to be enriched in senescence as well as a subtle downregulation of hexosylceramides (Fig. 5a). In addition, the composition of fatty acids represents polyunsaturated fatty acids in senescence (Fig. 5b). Cholesterol was found to be esterified to the bulk of the polyunsaturated fatty acid pool with arachidonic acid (20:4) and dihomo-γ-linolenic acid (20:3), bioactive fatty acids implicated in cellular senescence and the SASP, being the highest ranked compared to other lipid species[50] (Fig. 5c). The accumulation of cholesterol esters and lipid droplets suggests that senescent macrophages adopt a lipid-laden, foam cell-like phenotype that may contribute to chronic inflammation, metabolic dysfunction and tissue degeneration in aging and disease.

High serum levels of low-density lipoprotein (LDL) cholesterol is a major risk factor for metabolic disease and is known to increase intracellular cholesterol levels in macrophages. We hypothesized that excess cholesterol loading via LDL could serve as a driver of macrophage senescence in vitro (Fig. 5d). To test this, macrophages were treated with an acetylated form of LDL (Ac-LDL) which bypasses the LDL receptor, providing a pharmacological method to rapidly load macrophages with cholesterol (Fig. 5e). In response to Ac-LDL, we observed a time-dependent increase in TREM2 and p21 expression, while p16 was modestly decreased (Fig. 5f). Increased SA-β-gal activity and reduced cell proliferation were also observed during prolonged Ac-LDL exposure (Fig. 5g,h). To quantify changes in gene expression in response to Ac-LDL, macrophages were treated with four independent batches of Ac-LDL to embrace the natural variability of commercialized LDL. Under high-cholesterol conditions, both *Cdkn1a* (p21) and *Trem2* were upregulated, whereas *Cdkn2a* (p16) did not vary significantly (Fig. 5i). In addition, we observed selective upregulation of SASP genes such as *Gdf15*, *Mmp9* and *Mmp12* (Fig. 5i). Collectively, these results demonstrate that cholesterol ester loading promotes a senescent, foam cell-like macrophage phenotype characterized by lipid accumulation and SASP activation.

It has recently been proposed that cholesterol accumulation, resulting from genetic deletion of the cholesterol efflux ATP-binding cassette, ABCA1, drives senescence through a mechanism involving CD38 activation and NAD$^+$ decline[18]. CD38 expression was also increased during DNA damage-induced senescence (Fig. 1i), and a senescence-associated decline in NAD$^+$ levels is a well-known feature of

**Fig. 6 | ABT-263 efficiently and selectively targets senescent macrophages for apoptosis. a**, Average ± s.e.m. mRNA transcript levels (RT–qPCR) for a collection of pro- and anti-apoptotic genes relative to control. *P* value derived from an ANOVA test. *n* = 3 biological replicates. **b**, Model illustration hypothesizing that ABT-263 drives apoptosis in senescent cells via BCL-2. **c**, Dose response curve (±s.e.m.) identifying the IC50 for specific senescent macrophage killing. The $R^2$ value represents the coefficient of correlation from *n* = 3 biological replicates. **d**, Flow cytometry quantification of dead cell gates via microscopy in response to ABT-263. *P* value of Tukey's test from two-way ANOVA. Data points represent percentage propidium iodide positive (PI+) cells (±s.e.m.) for *n* = 384 biological replicates. **e**, Flow cytometry quantification of apoptotic cell gates (Annexin V (AV)+ and PI+), ± s.e.m., in response to ABT-263 across a spectrum of polarized macrophage phenotypes. M0 represents naive macrophages, PC represents passage control macrophages. *P* value of Tukey's test from two-way ANOVA of *n* = 3 biological replicates. **f**, Model illustration of young (4 month) and aged (24 month) male mice treated with ABT-263 via oral gavage every day

for 7 days on weeks 1 and 4. **g**, Immunofluorescence microscopy staining for macrophages (F4/80) and p21 in treated liver tissues. Images taken at ×40 magnification. Scale bar: 20 μm. **h**, Relative p21$^+$ F4/80$^+$ (senescent macrophages) abundance relative to total macrophages (±s.e.m.). *n* = 6 mice were used in each young condition, *n* = 5 aged male mice in the placebo control and *n* = 7 aged male mice on ABT-263. *P* value of Tukey's test post ANOVA. **i**, Spleen to total mouse weight ratio (±s.e.m.) in *n* = 6 male mice used in each young condition, *n* = 5 aged male mice in the placebo control and *n* = 7 aged mice on ABT-263. *P* value of Tukey's test post ANOVA. **j**, Average ± s.e.m. mRNA transcript levels (RT–qPCR) for a collection of proinflammatory genes relative to young control. *P* value of Tukey's test post ANOVA. *n* = 6 male mice were used in each young condition, *n* = 5 aged mice in the placebo control and *n* = 7 aged mice on ABT-263. **k**, Oil Red O stain representative images of *n* = 6 male mice used in each young condition, *n* = 5 aged mice in the placebo control and *n* = 7 aged mice on ABT-263. Average ± s.e.m. lipid droplet area and volume were used in the statistics. *P* value of Tukey's test post two-way ANOVA.

aging[27]. Furthermore, *Cd38* mRNA was elevated during acute exposure to Ac-LDL and remained modestly elevated during later stages of senescence (Extended Data Fig. 7a). These findings suggest that cholesterol may contribute to macrophage senescence, at least in part, through a CD38-dependent mechanism. However, as we discussed earlier, our data suggest CD38 is not required for DNA damage and natural aging-induced senescence (Fig. 1i and Extended Data Fig. 1k–m). To test the role of CD38 in cholesterol-driven macrophage senescence, *Cd38* KO mice were subjected to Ac-LDL loading. Across multiple assays, no significant differences in senescence gene expression (*Cdkn1a*, *Mmp9*,

*Gdf15*), cholesterol sensing (*Abca1*) or SA-β-gal activity were observed compared to WT macrophages, suggesting CD38 is not necessary for cholesterol-induced senescence (Extended Data Fig. 7b,c), consistent with our earlier results.

## ABT-263 efficiently and selectively targets senescent macrophages for apoptosis

We next hypothesized that Ac-LDL-induced senescent macrophages can contribute to diseases associated with hypercholesterolemia. To target these cells for apoptosis, we investigated a panel of pro- and

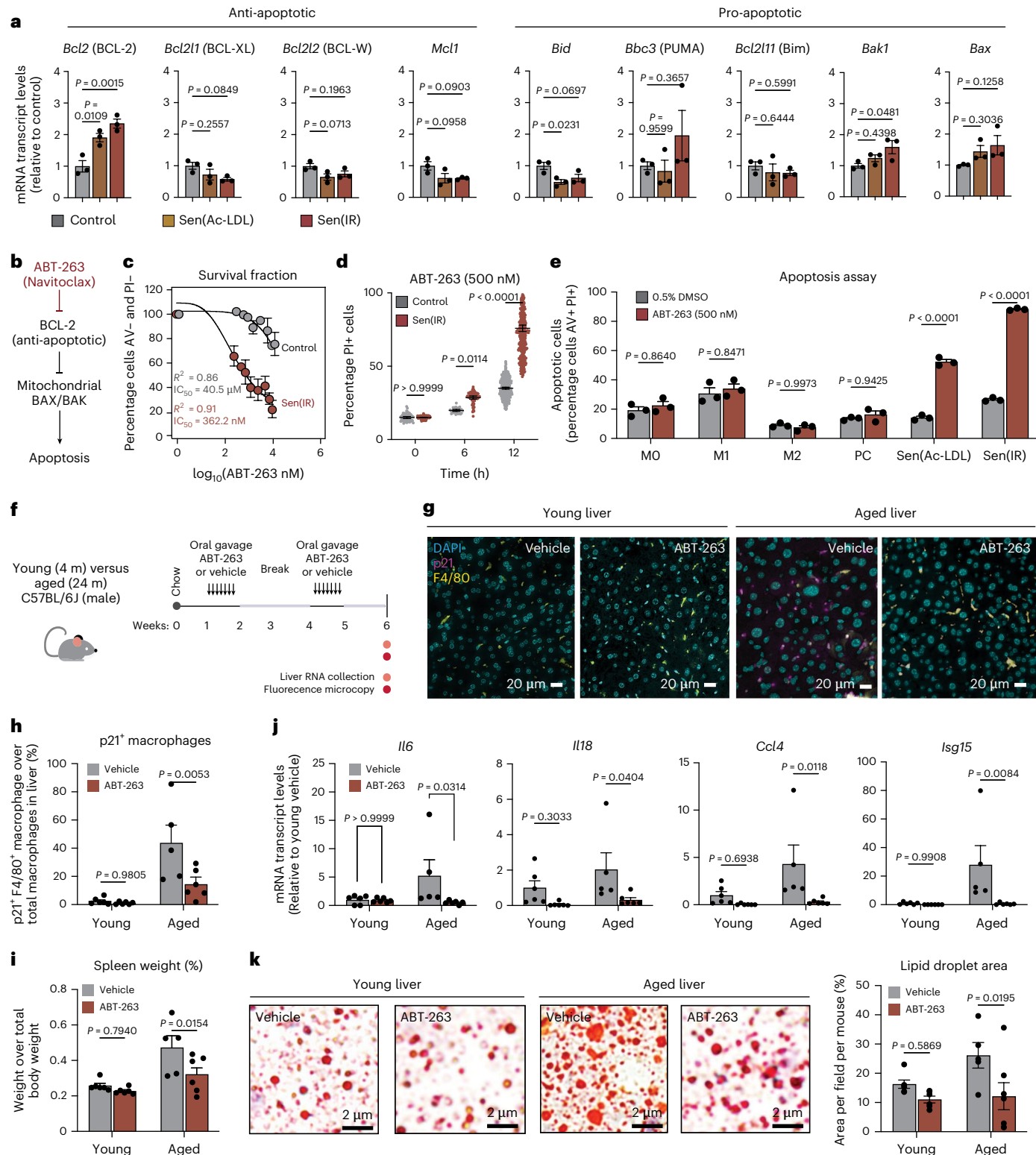

anti-apoptotic genes and identified *Bcl-2*, a key gene involved in anti-apoptotic pathways, as well as pro-apoptotic mitochondrial genes such as *Bak1* and *Bax* (Fig. 6a). BCL-2 is a direct inhibitor of mitochondrial BAX and BAK proteins and given this pattern in transcription, it was hypothesized that pharmacological inhibition of BCL-2 should allow mitochondrial BAX or BAK to drive apoptosis (Fig. 6b). To test this hypothesis, we used the known senolytics ABT-263 (Navitoclax), a BCL family inhibitor, and Dasatinib and Quercetin (D + Q). Selective elimination of senescent macrophages was observed in response to ABT-263, unlike D + Q, which induced cytotoxicity in both senescent and nonsenescent cells (Extended Data Fig. 8a,b). A dose- and time-dependent apoptotic effect was also observed with ABT-263, with a half maximal inhibitory concentration (IC50) value in the nanomolar range (Fig. 6c,d and Extended Data Fig. 8c). Next, we asked how effective ABT-263 is at targeting macrophage senescence across various polarized states. To test this, we treated M1 and M2 macrophages with ABT-263, along with Ac-LDL-induced senescent macrophages. After a 24-h exposure to ABT-263, apoptosis was only observed in senescent conditions, suggesting that ABT-263 is selective and effective at killing senescent macrophages, and not generally activated pro- or anti-inflammatory macrophages (Fig. 6e).

To assess the effects of ABT-263 on p21⁺ senescent macrophages in vivo, young and aged mice were treated by oral gavage using an intermittent 'hit-and-run' dosing strategy[5] (Fig. 6f). The liver, VAT and spleen were collected to determine the effect of senescent macrophage clearance on systemic inflammation and MASLD. Using immunofluorescence microscopy, the number of p21⁺ and F4/80⁺ macrophages in the liver increased from ~5% to ~50% during aging and was significantly reduced to ~10% in response to ABT-263 (Fig. 6g,h). Interestingly macrophages represent 60%–80% of p21⁺ cells in young and aged livers (Extended Data Fig. 8d). Systemically, it was observed that the relative spleen weight was decreased, suggesting repression of splenomegaly and systemic inflammation (Fig. 6i). Lower SA-β-gal activity in the VAT was observed and ABT-263 had no effect on total mouse weight (Extended Data Fig. 8e,f). The effect of p21⁺ macrophage clearance on liver inflammation resulted in significant downregulation of multiple cytokines and chemokines (Fig. 6j). In addition, we observed reduced lipid droplet area in both the young and aged livers in response to ABT-263 (Fig. 6k). Overall, these results identify ABT-263 as a potent

senolytic capable of selectively targeting macrophage senescence in vivo, and reducing inflammation and liver steatosis associated with aging.

## ABT-263 targets a senescent macrophage signature in MASLD

To assess the pathological relevance of macrophage senescence, *CETP-APOE*\*3-Leiden transgenic mice on a B6 background were placed on a high-fat high-cholesterol diet (HFHCD) to model MASLD and MASH. The mice were crossed with three genetic backgrounds (129/SvJ, C57BL/6J and BXD19/Tyj) that exhibit varying degrees of liver fibrosis (Fig. 7a,b). Of note, these mice only develop MASH when fed on HFHCD for 16 weeks, and not just a high-fat diet alone[51]. Liver samples were collected throughout the diet and pathology validated the genetic susceptibility to fibrosis, with the BXD19/Tyj strain being sensitive and the 129/SvJ strain resistant to fibrosis (Fig. 7b,c). Liver tissue samples were subjected to RNA-seq to identify MSen genes associated with macrophage senescence, such as *Cdkn1a* and *Trem2*. A strain- and time-dependent interaction was observed across all genotypes, with the highest expression of senescence markers in the BXD19/Tyj strain and the lowest in the 129/SvJ strain (Fig. 7d). Quantification of the MSen score revealed that the relationship between diet, time and macrophage senescence was strongest in the BXD19/Tyj strain compared to the 129/SvJ strain (Fig. 7e). These data suggest that strain-specific differences in fibrosis and senescent macrophage accumulation may influence the progression of liver MASLD. Based on these findings, we hypothesized that macrophages from these strains may exhibit varying intrinsic susceptibility to senescence. To test this, BMDMs were isolated from each strain and gene expression analysis revealed strain-specific upregulation of *Cdkn1a* (p21), *Trem2* and SASP genes in Sen(IR) macrophages, with stronger effects in strains more prone to fibrosis, such as BXD19/Tyj and C57BL/6J, compared to 129/SvJ mice (Fig. 7f). Expression of *Cdkn2a* (p16) was also assessed; however, no significant differences were observed in each mouse strain (Extended Data Fig. 9a). These findings suggest that hypercholesterolemia drives macrophage senescence in vivo, and that the extent of fibrosis and senescent macrophage accumulation may be positively correlated and influenced by genetic determinants through cell-intrinsic mechanisms.

To test the contribution of macrophage senescence to fatty liver disease progression, we treated HFHCD-fed C57BL/6J *APOE*\**CETP*

**Fig. 7 | ABT-263 targets a senescent macrophage signature in MASLD. a**, Model illustration of *CETP-APOE*\* Leiden transgenic mice maintained on C57BL/6J then crossed to three additional genetic backgrounds susceptible (BXD19/TyJ and C57BL/6J) and resistant (129/SvJ) to developing to MASLD. **b**, Representative picrosirius red stain for liver fibrosis after 16 weeks on HFHCD. Scale bars represent a 20 µm distance. **c**, Pathology grades of fibrosis of n = 4–6 male liver sections stained with picrosirius red stain. Data represent median ± s.e.m. **d**, Bulk RNA-seq of transcript levels for MSen gene signature across weeks 0–16 on HFHCD. **e**, Mean ± s.e.m. MSen scores across weeks 0–16 on HFHCD for each genetic background. Best-fit line and P value derived from a simple linear regression model of n = 3–6 male mice per time point. **f**, Mean ± s.e.m. mRNA transcript levels (RT–qPCR) relative to control of BMDMs from *CETP-APOE*\*Leiden mice crossed on different three backgrounds. P value from a two-way ANOVA. n = 3 biological replicates per strain and condition were used. **g**, Model illustration of *CETP-APOE*\*Leiden transgenic mice crossed on the C57BL/6J genetic background. Mice were treated with ABT-263 via oral gavage every day for 7 days on weeks 12 and 15. **h**, Mean ± s.e.m. weight of n = 9 placebo mice and n = 10 ABT-263-treated mice at 17 weeks. P value derived from a two-sided t-test. **i**, Mean ± s.e. change in food intake for n = 6 placebo cages and n = 5 ABT-263-treated cages at 17 weeks. **j**, Mean ± s.e. spleen weight (in grams) relative to total weight of n = 9 placebo mice and n = 10 ABT-263-treated mice at 17 weeks. P value derived from a two-sided t-test. **k**, Enzyme-linked immunosorbent assay quantification for picograms of tumor necrosis factor found in every microliter of serum collected. P value derived from a one-sided t-test. Mean ± s.e.m. of n = 9 placebo mice and n = 10 ABT-263-treated mice at 17 weeks. **l**, Mean liver weight (in grams) relative to total weight ± s.e. of n = 9 placebo mice and n = 10 ABT-263-

treated mice at 17 weeks. P value derived from a two-sided t-test. **m**, Representative liver photographs at 17 weeks immediately after tissue harvest. Scale rule: inches. **n**, Mean mRNA transcript levels (RT–qPCR) for SASP, senescent and macrophage genes in bulk liver samples relative to placebo control. P value derived from a two-sided t-test. Bar plots: average ± s.e.m. of n = 9 placebo mice and n = 10 ABT-263-treated mice at 17 weeks. **o**, Bulk RNA-seq of the MSen transcriptomic signature (genes derived from aged macrophages in vivo and senescent macrophages in vitro) in response to the senolytic drug, ABT-263. Scale bar: log₂[TPM + 1] expression on a z-score axis. n = 6 vehicle and n = 5 ABT-263-treated mice used in this experiment. **p**, GSVA scoring of MSen genes in the livers of vehicle- or drug-treated mice. Box-and-whisker plots display the median (center line), interquartile range (25th to 75th percentiles; box) and minimum and maximum values (whiskers). P values were calculated using a nonparametric Wilcoxon test. n = 6 vehicle-treated and n = 5 ABT-263-treated mice were analyzed. **q**, Total NAD (pmol per mg of liver analyzed) quantified by LC–MS/MS. Mean ± s.e.m. of NAD distribution in response to ABT-263. n = 9 placebo mice and n = 10 ABT-263-treated mice at 17 weeks. P value derived from a two-sided t-test. **r**, Nonalcoholic fatty liver disease (NAFLD) activity score based on pathology grades for each condition. Bar: mean ± s.e. P value derived from Student's t-test. **s**, Oil Red O stains are shown for three representative mice treated with vehicle or ABT-263. Scale bars represent a 20 µm distance. Mean ± s.e.m. lipid droplet size and area analyzed in eight images per liver section, and the average size (µm²) from eight slides for each mouse was used for statistical analysis. n = 6 vehicle and n = 5 ABT-263-treated mice used in this analysis. P value derived from Student's t-test.

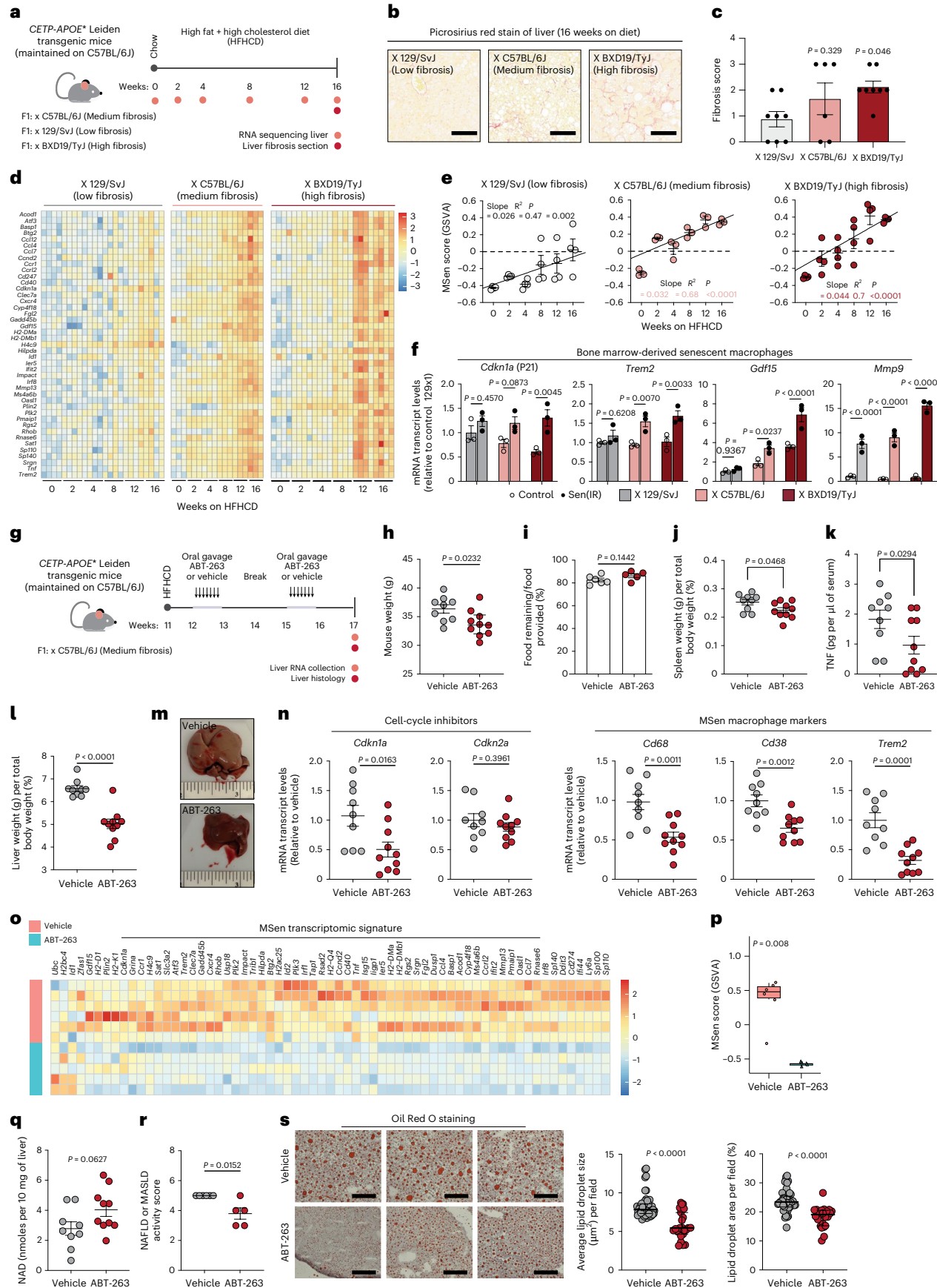

transgenic mice with ABT-263 or vehicle control. Treatment commenced at week 12 of the diet, because RNA-seq analysis indicated a strong increase in expression of our MSen genes in the liver at that time point (Fig. 7d–g). Interestingly, ABT-263-treated mice exhibited a significant reduction in body weight without changes in food intake (Fig. 7h,i). In the progression to MASH, splenomegaly is common because of systemic inflammation, and is reduced in ABT-263-treated mice, along with a decrease in serum tumor necrosis factor levels (Fig. 7j,k). Serum metabolites remained largely unchanged, aside from a modest effect on blood glucose, and circulating liver enzymes (alanine transaminase (ALT) and aspartate transaminase (AST)) were unaffected (Extended Data Fig. 9b–d). Notably, the liver improved dramatically, with treated livers appearing smaller and darker red compared to the enlarged, yellow-steatotic appearance typically observed in HFHCD-fed controls (Fig. 7l,m). Total liver gene expression revealed no effect on *Cdkn2a* (p16) and a reduction in *Cdkn1a* (p21), *Trem2* and inflammatory macrophage markers like *Cd68* and *Cd38*. (Fig. 7n). By contrast, we observed only modest decreases in *Cdkn1a* or *Cd38* expression in other tissues such as the kidney or VAT (Extended Data Fig. 9e), suggesting that cholesterol-driven macrophage senescence primarily affects the liver. Furthermore, using unbiased transcriptomics, we observed reduced expression of M1 and M2 genes and a significant decrease in the MSen transcriptomic signature (Fig. 7o,p and Extended Data Fig. 9f).

High CD38 expression, linked to senescent cell burden, can lead to NAD$^+$ degradation in the liver[27]. Given the reduction in liver *Cd38* expression and reduction in the MSen signature following senolytic treatment, we hypothesized that liver NAD$^+$ levels might be restored after targeting senescent macrophages. When measured, liver NAD$^+$ levels in ABT-263-treated mice increased by 30%, suggesting that senescent macrophages regulate tissue NAD$^+$ metabolism likely via a *Cd38*-dependent mechanism (Fig. 7q). Histological analysis of liver sections stained with picrosirius red revealed no significant change in fibrosis grade; however, a reduction in the nonalcoholic fatty liver disease activity score, driven by improvements in steatosis, was observed (Fig. 7r and Extended Data Fig. 9g). Oil Red O staining showed a significant decrease in lipid droplet size and area in ABT-263-treated mice (Fig. 7s). In addition, ABT-263 treatment reshaped the liver lipidome, reducing cholesterol esters and triglycerides while increasing lysophosphatidylethanolamine (Extended Data Fig. 9h). Together, these findings suggest that macrophage senescence occurs in cholesterol-driven liver disease and that targeting senescent macrophages reduces systemic inflammation and improves liver steatosis in MASLD, therefore preventing CD38-dependent NAD$^+$ decline. These results complement a recent study showing that elimination of p21$^+$ senescent cells during aging improves metabolic health and extends lifespan in mice[52], and suggest this may be dependent in part on targeting p21$^+$Trem2$^+$ senescent macrophages.

## *TREM2*$^+$ scar-associated macrophages express a human senescent macrophage gene signature

To determine whether human macrophages can undergo senescence, macrophages derived from peripheral blood mononuclear cells (PBMCs) were subjected to the Sen(IR) and Sen(Doxo) models, similar to the mouse BMDM protocols (Fig. 8a). We observed a significant increase in *TREM2*, *CDKN1A* (p21), *GDF15* and *CD38* expression in both Sen(IR) and Sen(Doxo) macrophages (Fig. 8b). The expression of *CDKN2A*, *IFNα*, *IFNβ*, *MMP9* and *MMP12* did not change significantly (Fig. 8b and Extended Data Fig. 10). Interestingly, the increased protein expression of p21 and reduced p16 expression were similar to that seen in mouse senescent macrophages (Fig. 8c). In addition, SA-β-gal activity and reduced proliferative capacity were observed in human Sen(IR) macrophages (Fig. 8d,e). Together, these findings suggest that the Sen(IR) model in human macrophages serves as a reliable system for investigating the mechanisms and biomarkers of human macrophage senescence.

To further explore human senescent macrophage biology, an unbiased bulk RNA-seq approach was used to capture global transcriptomic changes. In this experiment, we observed a donor-specific response to IR and Doxo, suggesting that genetics has an influence on senescence susceptibility, similar to that we observed in senescent macrophages from varying strains of mice. Among donors, PCA demonstrated clear separation between senescent and control macrophages, with 1,200 DEGs identified (analysis of variance (ANOVA), with a modest *q* value <0.1) (Fig. 8f). This clustering indicated a distinct transcriptional phenotype in senescent macrophages, with 163 genes being differentially expressed (ANOVA, *q* < 0.05) across all donor conditions during senescence (Fig. 8g). Upregulated genes included redox and detoxification factors (*MGST1*, *QSOX1*, *ALDH3B1*), mitochondrial regulators (*ISCU*, *NDUFA4*, *COX7A1*) and lysosomal or vesicular components (*ATP6V1D*, *ATP6V0E1*, *MAN2B1*, *SIDT1*), reflecting increased lysosomal activity characteristic of senescence. Metabolic and lipid-handling genes (*FABP4*, *GRAMD1B*, *APMAP*) and canonical senescence markers (*CDKN1A* and *CDKN1C*) were elevated, alongside antigen presentation and immune-modulatory transcripts (*HLA-DRA*, *HLA-DMA*, *FCN1*, *MS4A7*, *ADGRE1*), indicating an inflammatory phenotype. By contrast, genes governing cell-cycle progression and DNA replication were markedly repressed, including *CDK1*, *CDC20*, *CHEK1*, *MCM7* and *TOP2A*, consistent with irreversible proliferative arrest. DNA repair factors (*EXO1*, *MSH6*, *BLM*, *RECQL4*) and nuclear lamina components (*LMNB1* and *LMNA*) were also reduced, reflecting deep senescence and impaired genome maintenance (Fig. 8g). KEGG analysis revealed that downregulated genes were primarily involved in cell proliferation pathways, while the upregulated genes were associated with inflammatory responses (Fig. 8h,i). Together, these transcriptional changes define a stable senescent macrophage phenotype characterized by heightened stress and lysosomal programs coupled with altered proliferative, metabolic and immune functions.

**Fig. 8 | *TREM2*$^+$ scar-associated macrophages express a human MSen gene signature. a**, Illustration of the in vitro model system using exogenous DNA damage with irradiation and Doxo on PBMC-derived macrophages from male and female human donors. **b**, Mean ± s.e. gene expression (RT–qPCR) normalized to control. *n* = 9 donors (mixed male and female) for control and Sen(IR) conditions. *n* = 8 for Sen(Doxo) conditions. *P* value from Tukey's test post repeated measures ANOVA analysis. **c**, SDS–PAGE gels and immunostaining (western blot) for p21 and p16 in response to 10 days post DNA damage. ImageJ quantification for band intensity is shown to the right and represents relative intensity over the loading control. **d**, SA-β-gal images of human senescent macrophages. **e**, Click-iT EdU labeling of cells to assess proliferation dynamics. Axes represent the percentage of cells positive for the stain relative to the parental gate, as detected by flow cytometry. The bar plot represents the mean ± s.e. of *n* = 3 independent donors. *P* value derived from unpaired one-sided *t*-test. **f**, PCA on bulk RNA-seq samples from 10 days post DNA damage for three blood donors. **g**, Heatmap projection of all statistically significant DEGs compared to control conditions. **h**, KEGG pathway enrichment analysis for the top ten pathways downregulated in human senescent macrophages. **i**, KEGG pathway enrichment analysis for the top ten pathways upregulated in human senescent macrophages. **j**, Model illustration of running MSen score on publicly accessible single-cell dataset[49] of human liver biopsy samples from five people with liver cirrhosis and five with a healthy liver. **k**, UMAP projection of all annotated CD45$^+$ cell types and colored by human MSen Seurat score. **l**, UMAP projection of *CD68*$^+$ cells (macrophages) in healthy and cirrhotic tissue samples. Color denotes human MSen Seurat score. **m**, Distribution of MSen Seurat score across resident macrophage (Kupffer cells) and non-resident monocyte and macrophage populations in healthy and cirrhotic conditions. *P* value represents the results of a two-sided *t*-test. **n**, Distribution of human MSen score across all CD45$^+$ cell types in the dataset. ALD, alcohol-related liver disease; F, female; KC, Kupffer cell; IgA, immunoglobulin A; IL-C, innate lymphoid cell; M, male; MoMac, monocyte-derived macrophage; Mø, macrophage; PBC, primary biliary cholangitis; pDC, plasmacytoid dendritic cell. Panels **a** and **j** created in BioRender; Salladay-Perez, I. https://biorender.com/6umuckd (2026).

To evaluate whether senescent macrophages represent *TREM2*+ macrophages in liver disease, we analyzed a publicly available single-cell RNA-seq dataset of human liver cirrhosis that identified a *TREM2*+ scar-associated macrophage (SAM) population[49] (Fig. 8j). Using the genes identified in our bulk human MSen transcriptomic analysis as a scoring method, we observed significant enrichment for senescence-associated genes in Cd45+ immune cells in both healthy and

diseased livers, with macrophages being the most enriched (Fig. 8k,l). Further analysis into CD45+CD68+ macrophages revealed a pre-existing SAM population enriched for senescence in the human control samples (Fig. 8m). In the context of cirrhosis, Kupffer cells and monocytes and/or macrophages appear to transition into SAMs that score high for the human MSen signature and outrank all other immune cells in the liver (Fig. 8m,n). These results indicate that macrophage senescence

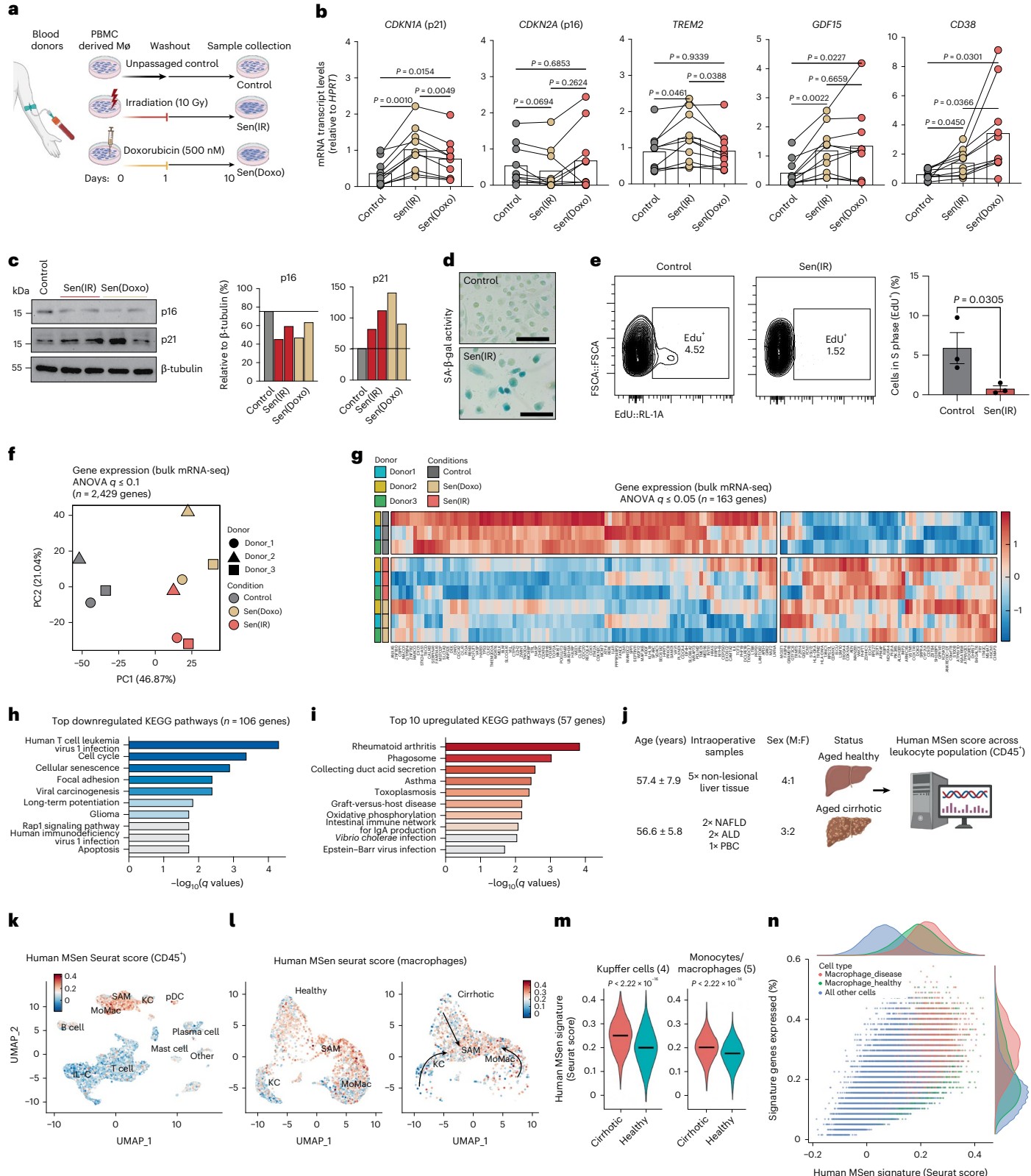

is present in human chronic liver diseases, including MASLD and cirrhosis, and may represent a therapeutic target.

## Discussion

In this study, we characterized DNA damage and cholesterol-induced macrophage senescence, uncovering stable senescent phenotypes defined by irreversible cell-cycle arrest and a robust SASP phenotype. A key finding was the upregulation of *Cdkn1a* (p21) expression during senescence, supporting p21 as a more reliable marker of macrophage senescence compared to *Cdkn2a* (p16), which was paradoxically downregulated or unchanged in both mouse and human macrophage models. Our data are also consistent with recent publications describing distinct p21⁺ or p16⁺ senescent cells, referred to as senotypes[28,29]. Nevertheless, our data suggest that neither p21 nor p16 alone is sufficient to definitively identify senescent macrophages, reinforcing the need for multiple markers and cell-specific approaches. By comparing senescent macrophages with classical M1 and M2 polarized states, we identified additional distinguishing features, notably high expression of *Trem2*, altered nuclear morphology, lipid metabolism and type I IFN hyperactivation.

Lipidomic analyses revealed that cholesterol storage, particularly storage of cholesterol esters and ceramides, not only marked senescent macrophages, but also acted as a physiological driver of the senescent state. This finding links senescence to metabolic dysfunction and highlights an important, previously underexplored lipid axis that drives the senescent cell burden in age-related diseases. Furthermore, we demonstrated that the senolytic ABT-263 selectively eliminated p21⁺ senescent macrophages both in vitro and in vivo, resulting in reduced hepatic steatosis and systemic inflammation. Moreover, prior studies using ABT-263 in aged mice reported improvements in organ function and metabolic health[53]. Our findings suggest that clearance of senescent macrophages may account, at least in part, for these benefits. This therapeutic effect is particularly timely given the rising global burden of metabolic diseases driven by dietary excess and aging.

Our findings raise the intriguing possibility that diets rich in excess cholesterol accelerate aging by driving senescence in tissue-resident macrophages, an aging hallmark traditionally associated with advanced age. Our data also suggest that senolytic strategies could potentially treat not only MASLD, but also other cholesterol-driven diseases such as atherosclerosis, where senescent-like macrophages have been observed[54]. Future studies will need to determine whether targeting senescent macrophages can halt or reverse the broader spectrum of age-associated inflammatory disorders where these cells have been recently implicated, including cancer[13,14].

Another critical question centers on how cholesterol acts as a stressor to induce senescence. Cholesterol has essential roles in membrane structure and cell signaling, but excess intracellular cholesterol can be cytotoxic and drive inflammation[55]. Although cholesterol itself has not been directly linked to genomic instability, its contribution to mitochondrial dysfunction, oxidative stress and endoplasmic reticulum stress may promote a stress response that drives senescence. Elucidating the molecular mechanisms connecting lipid overload to macrophage senescence will be crucial for understanding the intersection of metabolism and inflammaging at a deeper mechanistic level.

In conclusion, this study establishes p21⁺TREM2⁺ senescent macrophages as a distinct, pathogenic population that drives sterile inflammation in aging and MASLD. However, several important limitations of our study remain. First, our study uses male mice as the primary source of BMDMs and for in vivo studies. However, given the strong genetic influence on senescent macrophage phenotypes, it is possible that sex chromosomes may influence key senescent hallmarks, which is currently under investigation in our laboratory. Second, our in vivo analysis primarily focuses on characterizing and identifying senescent Kupffer cells in the liver. We hypothesize that other tissue-resident macrophage populations, particularly in organs with large tissue-resident

macrophage reservoirs such as the brain, skin, VAT and heart, are also susceptible to senescence. Importantly, these macrophages may undergo senescence via distinct mechanisms, resulting in diverse phenotypic states and biomarker profiles that will require tailored identification strategies. Thus, future work should prioritize the development of macrophage-specific senescence reporter mice that could facilitate the selective targeting and tracking of senescent macrophages in complex tissues and conditions. Establishing these tools would enable the field to determine not only the burden and dynamics of macrophage senescence during aging and disease, but also causal relationships with pathology and macrophage-targeted senolytic strategies as promising therapeutic interventions to combat chronic inflammation, metabolic dysfunction and age-related tissue decline.

## Methods

### Ethical compliance for mouse experiments

All mouse experiments and procedures were approved and performed in accordance with the guidelines set forth by the University of California, Los Angeles (UCLA) Animal Resource Committee and the Institutional Animal Care and Use Committees (IACUC). In addition, PBMCs were isolated at UCLA's Virology Core; because these samples do not contain any identifying information, these experiments are IRB-exempt.

### Mice

Male C57BL/6J mice 8–12 weeks old (Jackson Laboratory) served as a source of bone marrow for BMDMs. Aged C57BL/6J mice from the Jackson Laboratory and from the National Institute of Aging were used to harvest aged (21–24 months) tissues. WT and *Cd38* KO mice on a C57BL/6J background were obtained from the Jackson Laboratory and were bred and housed at UCLA's animal facility in the Center for Health Sciences. Bone marrow from these mice were used in experiments comparing WT and *Cd38* KO. To eliminate p16-positive cells, p16-3MR mice (provided by the Campisi Lab) were housed at the Buck Institute, and bone marrow was shipped to UCLA for in vitro BMDM experiments with GCV. The breeding and characterization of *APOE*\*Leiden-CETP transgenic mice have been described previously[51,56,57]. To generate the F1 mice for the fibrosis studies, homozygous male mice carrying both *CETP* and *APOE*\*-Leiden transgenes were maintained on C57BL/6J background and crossed to C57BL/6J, 129/SvJ and BXD19/TyJ mice, respectively. All mice were housed at UCLA's animal facility in the Center for Health Sciences. Unless used for MASLD studies described below, all mice were maintained at UCLA on a standard chow diet.

### Mouse macrophage culture

BMDMs used in experiments were derived from bone marrow extracted from the femurs of euthanized mice (6–12 weeks old, male C57BL/6J) by mortar and pestle. Femurs were placed in the mortar and were washed with 70% ethanol to sterilize followed by two washes with complete RPMI (cRPMI; standard RPMI (Corning) supplemented with 10% fetal calf serum, penicillin–streptomycin solution (Corning), 1 mM sodium pyruvate solution (Corning), 2 mM L-glutamine solution (Corning), 10 nM HEPES buffer (Corning) and 50 μM 2-mercaptoethanol). After washing, 10 ml of cRPMI was added to the mortar and the femur bones were gently crushed. The resulting media was collected, filtered through a 70-μm filter and placed in a conical tube. The filtered supernatant was centrifuged at 100 *g* (150 relative centrifugal force) for 5 min. Cells were resuspended, counted and plated at a density of $3 \times 10^6$ cells per 10-cm dish in 10 ml of macrophage growth media (cRPMI containing 25% macrophage-colony stimulating factor (M-CSF) containing L929 conditioned media (made in-house)). Cells were left to grow for 7 days to differentiate and were supplemented with 5 ml of macrophage growth media on day 5. On day 7, BMDMs (yielding $10–12 \times 10^6$ cells per 10-cm dish) were lifted off the plate using cold phosphate-buffered saline (PBS) containing 5 mM EDTA. BMDMs were

counted and replated in macrophage growth media overnight before experiments. On the day of experiments, macrophage growth media was replaced with cRPMI 6 h before stimulation to remove M-CSF. M2 polarization was performed by stimulating macrophages with 10 ng ml$^{-1}$ recombinant mouse IL-4 (PeproTech). For M1 polarization macrophages were stimulated with 100 ng ml$^{-1}$ LPS (LPS EK-Ultrapure, InvivoGen). To induce senescence, day 7 BMDMs were either irradiated (10 Gy) or treated with Doxo (Fisher) for 24 h at 500 nM in 60% cRPMI/40% macrophage growth media. DNA-damaged cells were left in culture for 10 days and media components were replaced every 2–3 days. As a negative control, a mock irradiated or drug-treated condition was cultured in parallel and required passing every 2–3 days. All senescent macrophage experiments were performed 10–13 days post DNA damage.

### Reverse transcription-quantitative PCR and primers

For all human PBMC and mouse BMDM cell-culture experiments, RNA was isolated using RNA STAT 60 (Amsbio) as per the manufacturer's protocol. Approximately 2,000 ng of RNA was converted to complementary DNA using a High-Capacity cDNA Reverse Transcription Kit (Applied Biosystems). Gene expression was measured using a CFX384 Real Time System (Bio-Rad) and Thermo Scientific Maxima 2× SYBR Green. Data were analyzed using Bio-Rad CFX Maestro 3.1 software and normalized using the $2^{-\Delta\Delta Ct}$ method. Mouse BMDM genes were normalized to hypoxanthine phosphoribosyltransferase (*Hprt*). For in vivo experiments, RNA was extracted from flash-frozen liver, VAT, kidney and heart using RNA STAT 60 in a TissueLyser system (Qiagen). cDNA was generated as described above, and gene expression was measured using iCFX384 Real Time System (Bio-Rad) and Thermo Scientific Maxima 2× SYBR Green. Expression was normalized to *Hprt* by the $2^{-\Delta\Delta Ct}$ method automatically using Bio-Rad CFX Maestro (v.0.3.1) with the gene-specific primer pairs listed below. All primers are written in the 5′ → 3′ direction.

For mouse genes, the following primer pairs were used: *Cdkn1a* (forward: TTGCCAGCAGAATAAAAGGTG; reverse: TTTGCTCCTGTGCGGAAC), *Cdkn2a* (forward: AACTCTTTCGGTCGTACCCC; reverse: TCCTCGCAGTTCGAATCTG), *Cd38* (forward: GTATGGCCTTGCTGGAATAG; reverse: TTGAAGGCTGTTAGTGGAATAG), *Gdf15* (forward: CCGAGAGGAACTCGAACTCAG; reverse: GGTTGACGCGGAGTAG), *Hprt* (forward: TTTCCCTGGTTAAGCAGTACAGCCC; reverse: TGGCTGTATCCAACACTTCGAGA), *Cd38* (Exon 2; forward: GGAGCATTTGTTTCCAAGAAC; reverse: GAGAGTCTTGTTACATGGTATGG), *Ifnβ* (forward: GCACTGGGTGGAATGAGACTATTG; reverse: CTCCCACGTCAATCTTTCCTCTTG), *Ifnα* (forward: GGATGTGACCTTCCTCAGACTCATA; reverse: GAATCCAAAGTCCTTCCTGTCCTTC), *Abca1* (forward: GCCAAGCATCTTCAGTCCATCAG; reverse: CAGTGTAGCAGGGACCACATAA), *Plin2* (forward: TCTGAGGTCAAAGCTCAGTAAAC; reverse: AGCCATCTACCAAGTTAATTTCAATAC), *Il6* (forward: ACAAAGCCAGAGTCCTTCAGAGAG; reverse: AGAACTGATGAGAGGGGAGGCCATT), *Il10* (forward: GCTCTTACTGACTGGCATGAG; reverse: CGCAGCTCTAGGAGCATGTG), *Fizz1* (forward: TCCAGCTGATGGTCCCAGTGAATA; reverse: ACAAGCACACCCAGTAGCAGTCAT), *Mgl1* (forward: TGCAACAGCTGAGGAAGGACTTGA; reverse: AACCAATAGCAGCTGCCTTCATGC), *Mrc1* (forward: TGGGCTACAGGAGAACCCAACTTT; reverse: GCAGTGGCATTGATGCTGCTGTTA), *Arg1* (forward: ACCTGGCCTTTGTTGATGTCCCTA; reverse: AGAGATGCTTCCAACTGCCAGACT), *Ym1* (forward: AGAAGGGAGTTTCAAACCT; reverse: GTCTTGCTCATGTGTGTAAGTGA), *Cd36* (forward: TCATGCCAGTCGGAGACATGCTTA; reverse: AACTGTCTGTACACAGTGGTGCCT), *Mgl2* (forward: GCATGAAGGCAGCTGCTATTGGTT; reverse: TAGGCCCATCCAGCTAAGCACATT), *Il12b* (forward: TGCCCTCCTAAACCACCTCAGTTT; reverse: TTTCTCTGGCCGTCTTCACCATGT), *Mmp12* (forward: CTGGACAACTCAACTCTGGCAATAA; reverse: TGAGGTACCGCTTCATCCATCT), *Bcl2l11* (forward: GACCACCCTCAAATGGTTATCT; reverse: GTTCTCCAGTCTGAACAAGAG), *Bbc3* (forward: CCTGGAGGGTCATGTACAATCT;

reverse: CACCTAGTTGGGCTCCATTTCT), *Bid* (forward: GCTCCTTCAACCAAGGAAGAATA; reverse: GGTCCATCTCATCGCCTATTTG), *Bcl2l1* (forward: TGGTCGACTTTCTCTCCTACAA; reverse: CCCTCTCTGCTTCAGTTTCTTC), *Mcl1* (forward: CTAGAAGGCGGCATCAGAAATG; reverse: CCTATTGCACTCACAAGGCTATC), *Bcl2l2* (forward: GTGGGTAGAAGCTTTGGTAGTT; reverse: AGACTCAGCTGGATAGAGAGAC), *Ccl4* (forward: AGCTCTGTGCAAACCTAACC; reverse: GGTGTAAGAGAAACAGCAGGAA), *Il18* (forward: GCCAGAAGCAGACTCCTTAAT; reverse: ACACCAGGAAATCGTTACCC), *Oas1a* (forward: GACGCTGGACAAGTTCATAGAG; reverse: CTTGGAAGCATCTCTCCTTCAG), and *Cmpk2* (forward: GTTCCTCATTACACCGGAGAAG; reverse: GCTCAGTCAGTCAGAAGGAAAG).

For human genes, the following primer pairs were used: *TREM2* (forward: GAGCCTCTTGGAAGGAGAAATC; reverse: TGGCTGCTAGAATCTTGATGAG), *IFNA* (forward: CCAGCAGATCTTCAACCTCTTT; reverse: GGTAGAGTTCGGTGCAGAATTT), *IFNB1* (forward: GCTTCTCCACTACAGCTCTTTC; reverse: CAGTATTCAAGCCTCCCATTCA), *GDF15* (forward: GGAATCTGGAGTCTTCGGAGTG; reverse: CACGTATGTCATCAGCAGAGAG), *MMP9* (forward: TCACTTTCCTGGGTAAGGAGTA; reverse: CTGTCAAAGTTCGAGGTGGTAG), *MMP12* (forward: CTAGTGATCCAAAGGCCGTAAT; reverse: CACGTATGTCATCAGCAGAGAG), *P16* (forward: GAGCAGCATGGAGCCTTC; reverse: CGTAACTATTCGGTGCGTTG), and *P21* (forward: TCACTGTCTTGTACCCTTGTGC; reverse: GGCGTTTGGAGTGGTAG).

### Transcriptomics

Mouse BMDM-derived and human PBMC-derived macrophage cultures were lysed with RNA STAT 60 (Amsbio) and ~50 ng per μl of RNA was submitted to the UCLA's Technology Center for Genomics and Bioinformatics (TCGB) core for library preparation and 2 × 100 paired-end sequencing. Roughly 100 million high-quality reads were sequenced per sample using NovaSeq X Plus (Illumina). Raw data in FASTAq format were filtered for adapter sequences or low-quality sequences using SOAPnuke. Reads were aligned to the mm10 reference genome using STAR. Aligned reads were processed using TopHat and a counts matrix was generated using FeatureCounts. Raw reads were processed using R (v.4.5.0) and DESeq2 (v.1.48.2) where the raw reads were normalized to gene length then transformed into transcripts per million (TPM). Genes with a TPM value of 10 or greater were kept for normalization using log$_2$ and quantification. Differentially expressed genes (DEGs) were determined using parametric tests (ANOVA) and *P* values were corrected for false positives using false discovery rate (FDR) tests for *q* values. Gene interactions with a *q* value <0.05 were assessed for net effect size and genes with log(fold change) (log(FC)) >0.32 were considered a DEG. All heatmaps were made using the latest version of pheatmap (v.1.0.12) and all statistics were performed using base R commands.

### Immunofluorescence

Cells were transferred to six-well plates including coverslips. After 24 h, attached cells were fixed in 4% formaldehyde, then permeabilized in 0.25% Triton X-100. Subsequently, cells were blocked in 0.1% PBS with 3% bovine serum albumin (BSA) and 0.1% Tween-20, followed by incubation with the primary antibodies overnight. Cells were washed and incubated with the appropriate secondary antibodies for 1 h. The cells were washed and transferred to mounting media (Antifade GOLD reagent with DAPI; Thermo Fisher Scientific). A 3i Spinning Disk Confocal microscope (Marianas) consisting of a CSU-X1 A1 Spinning Disk (Yokogawa) attached to a Zeiss Axio Observer 7 microscope was used. Standard filter sets were used for imaging. Pearson's correlation coefficients were calculated using the JACoP plugin on ImageJ. Mouse livers were collected and either fixed in formalin or mounted into Optimal Cutting Temperature Compound Freezing Medium (OCT) and frozen at −80 °C. Tissue samples were sectioned at 5-μm thick and mounted onto slides for immunofluorescence. For in vivo tests, paraffin slides were dewaxed in SafeClear (Fisher Scientific) and ethanol, then antigens

were unmasked at 95 °C in 10 mM sodium citrate with 0.05% Tween-20. OCT slides were allowed to acclimate to room temperature, then fixed in cold acetone and permeabilized with PBS with 0.1% Triton X-100 and 1% BSA. All slides were blocked in PBS with 5% BSA and 0.1% Triton X-100 and incubated overnight with primary antibodies at a 1:200 dilution. Slides were washed and incubated with secondary antibodies at a 1:500 dilution for 1 h. After washing, slides were mounted in VECTASHIELD antifade mounting medium with DAPI (Vector Laboratories) and imaged with an Axioplan 2 fluorescent microscope (Zeiss) equipped with ×10–20 differential interference contrast objectives and Axiovision Rel 3.0 software (Zeiss). Cell counts from tiled whole-tissue images of slides were generated on QuPath (v.0.6.0) cell counter. F4/80+ macrophage counts and p21+ cells were quantified using QuPath's object classification tools. For immunofluorescence staining, the following primary antibodies were used: anti-TOM20 (rabbit; Proteintech, cat. no. 11802-1-AP; 1:1,000), anti-dsDNA (Abcam, cat. no. ab27156-1002; 1:1,000), anti-F4/80 (rat; Abcam, cat. no. ab6640; 1:200) and anti-p21 (rabbit; Abcam, cat. no. ab188224; 1:200).

### SA-β-Galactosidase
Cells and tissues were fixed and processed using a senescence detection kit as per the manufacturer's instructions (Cell Signaling Technologies, cat. no. 9860S or Biovision Senescence Detection Kit, cat. no. K320-250). Shortly, cultured macrophages were fixed, and liver tissues were frozen in OCT. Liver sections were cut (~5–10 μm) and processed for SA-β-gal and nuclear Fast Red (Fisher Scientific, cat. no. R5463200-500A) staining. Cultured murine cells and tissues were incubated in staining solution at pH 6 and 37 °C for 8–10 h. Cultured human macrophages were incubated in staining solution at pH 6 and 37 °C for 24 h. Cells were then stored in glycerol for long-term preservation and handling.

### Western blot analysis and antibodies
Cell-culture lysates from both mouse and human macrophages were prepared by lysing cells in RIPA Buffer containing Halt protease and phosphatase inhibitor cocktail 1:100 (Thermo Fisher). Protein concentrations were determined using the BCA Assay kit (Thermo Fisher). Approximately 10–20 μg of delineated protein in sodium dodecyl sulfate (SDS) was loaded onto 4%–20% polyacrylamide gels to separate by molecular weight via gel electrophoresis. Proteins were then transferred to polyvinylidene difluoride membranes. Polyvinylidene difluoride blots were placed in 5% milk for unspecific binding for 1 h, then exposed to primary and secondary antibodies. Western quantification was performed using ImageJ (v.150i). For immunoblotting, primary antibodies were used at a dilution of 1:1,000 unless otherwise specified. The following antibodies were used: anti-β-Tubulin (rabbit; Cell Signaling Technology, cat. no. 2146S), anti-Lamin B1 (rabbit; Cell Signaling Technology, cat. no. 13435 T), anti-phospho-Histone H2AX (γH2AX) (rabbit; Cell Signaling Technology, cat. no. 9718T), anti-phospho-NF-κB p65 (rabbit; Cell Signaling Technology, cat. no. 3033T), anti-NF-κB p65 (rabbit; Cell Signaling Technology, cat. no. 8242T), anti-p21 (rabbit; Abcam, cat. no. ab188224), anti-TREM2 (rat; Invitrogen, cat. no. MA5-28223), anti-CMPK2 (rabbit; Abcam, cat. no. ab139720) and anti-p16 (Abcam, cat. no. 211542).

### Shotgun lipidomics
Senescent macrophages and passage control macrophages were subjected to shotgun lipidomics using publicly accessible methods[58]. Briefly, media was removed and replaced with 1 ml of ice-cold PBS, then cells were scraped with cell lifters and spun down in glass tubes at 365g for 5 min at 4 °C. A modified Bligh and Dyer extraction was carried out on lifted cells, then 13-lipid class Lipidyzer Internal Standard Mix (AB Sciex, cat. no. 5040156) was added to each sample, before biphasic extraction. Following two successive extractions, pooled organic layers were dried down in a Genevac EZ-2 Elite. Lipid samples were resuspended in 1:1 methanol/dichloromethane with 10 mM ammonium acetate then transferred to robovials (Thermo Scientific, cat. no. 10800107) for analysis. Samples were analyzed for targeted quantitative measurement of 1,100 lipid species across 13 classes. Quantitative values were normalized to cell counts and average cell volume.

### Generation of MASLD mouse model
*CETP-APOE*\*Leiden mice (homozygous for transgene) were maintained on a C57BL/6J background then crossed with three distinct strains of mice with known response to liver fibrosis[51]. These strains were: 129/SvJ (fibrosis-resistant), C57BL/6J (medium-fibrosis) and BXD19/TyJ (fibrosis-prone), and male F1 progeny were used for experiments. Animals were maintained on a 12-h light–dark cycle with ad libitum access to water. Twenty-five male mice per strain (8–10 weeks old) were fed HFHCD (33% kcal from cocoa butter and 1% cholesterol; Research Diets, cat. no. D10042101) for 16 weeks. At the end of the study, mice were fasted for 4 h, beginning at 10:00 am and were killed at 2:00 pm.

### Preparation and administration of senolytic
*APOE*\*Leiden-CETP mice maintained on a C57BL/6J background and crossed to C57BL/6J mice, the male F1 progeny were placed on HFHCD at 12 weeks of age. Starting at week 12 of the HFHCD, mice were either treated with vehicle or ABT-263 diluted in 10% ethanol, 30% polyethylene glycol 400 (Sigma, cat. no. 807485) and 60% Phosal 50 PG (Medchem Express, cat. no. HY-Y1903). ABT-263 was administered by oral gavage at 50 mg per kg body weight per day for 7 consecutive days with a 2-week break or interval between another 7-day cycle. Tissues were harvested in week 17 of the HFHCD, 1 week after the last dose of ABT-263.

### Quantitative assessment of serum lipids and fibrosis in the liver
Liver lipids were extracted as previously described[59] and 100 mg of liver was used for lipid extraction; the dried organic extract was dissolved in 1.8% (w/v) Triton X-100. The amount of lipids in each extract was determined using a colorimetric assay from Sigma (triglyceride, total cholesterol and unesterified cholesterol) and Wako (phospholipids) according to the manufacturers' instructions. For histological examination and fibrosis quantitation, livers were fixed in 10% formalin, embedded in paraffin, sectioned at 5 μm and stained with picrosirius red. The fibrosis score was assessed by a pathologist blinded to the study, according to the MASLD Clinical Research Network[60].

### Oil Red O staining
Oil Red O staining was performed as described previously[61]. Sections were imaged using a Zeiss Axiolab microscope and Axiocam 305 color camera (Carl Zeiss Microscopy) with a ×10 objective. An average of eight images was taken across the liver sections for each experimental condition as described[61]. Similarly, for lipid droplet size, eight images per field per liver section were analyzed using the Analyze Particle function in ImageJ with Feret's diameter to determine the average size (μm$^2$).

### Human macrophage culture
Although sex and gender information was not recorded, blood from biological male or female human donors was collected and PBMCs were isolated at UCLA's Virology Core. Some 100 × 10$^6$ blood cells were collected from three to five donors with no clinical information. PBMCs were centrifuged at 100g for 5 min then 25 × 10$^7$ single cells were seeded onto 10-cm polystyrene-treated culture plates (VWR) with 10% adherence and allowed to differentiate into macrophages in the presence of 50 ng ml$^{-1}$ human M-CSF recombinant protein (PeproTech, cat. no. 300-25-100) in the same RPMI composition above. Cells were provided with an additional 5 ml of M-CSF-containing medium 5 days into differentiation. Seven days after seeding, confluent macrophage cultures were placed under similar senescence-inducing conditions using irradiation and Doxo (Thermo Fisher).

## Metabolomics

Metabolome analysis was performed on an Orbitrap Ascend Tribrid Mass Spectrometer using a direct injection method. Thermo (.RAW) files were converted to mzML files using ms convert. The produced mzML files were analyzed with MZmine3 software for mass detection, feature detection, alignment, gap filling and feature filter. The output files contain $m/z$ and quantification by field asymmetric ion mobility spectrometry compensation voltage peak area. For metabolite annotation, we applied a direct infusion-based data-dependent acquisition tandem mass spectrometry analysis with the same sample and the same compensation voltages applied in direct injection metabolomic analysis. The MS2 spectrum of metabolites was then compared with the library and analyzed through the GNPS website. Python (v.3.9.7) was used for metabolomic data analysis. $Z$-score normalization was applied to the dataset before one-way ANOVA. PCA analysis was performed with the Python package 'sklearn'.

## Proteomics

Liquid chromatography–mass spectrometry (LC–MS) proteome analysis was performed on an Orbitrap Ascend Tribrid Mass Spectrometer using data-independent acquisition (DIA). The tuning method details were as follows: positive ionization (2,500 V), ion injection time (100 ms), DIA window (12 Da), MS2 range (400–1,000), AGC (100%) and HCD energy (30%). For liquid chromatography, peptides were trapped on an EXP2 Stem Trap column and separated using a 200-cm µPAC column from PharmaFluidics. The entire setup was connected with 20 µm internal diameter Viper capillaries (Thermo Fisher Scientific) and maintained at 55 °C in a column oven. The reversed-phase analytical gradient was delivered as follows. Mobile phase A: 0.1% formic acid (FA) in water; mobile phase B: 0.1% FA in acetonitrile (ACN). Start at 8% B, linear ramp to 25% B at 70 min, linear ramp to 37% B at 95 min, jump to 98% B at 96 min and hold for 9 min, drop back to 8% B at 105 min and hold for 5 min (110 min total). The loading pump was connected to three solvents: 0.1% FA in water (loading buffer A), 0.2% FA in 70% ACN and 30% water with 5 mM ammonium formate (loading buffer B) and 0.2% FA in 50% isopropanol, 30% ACN, 20% water and 5 mM ammonium formate (loading buffer C). At the start of the method, the loading pump delivered 50% B and 50% C at 50 µl min$^{-1}$ to 5 min. The solvent was switched to 100% A at 6 min and ended at 62 min. The loading pump flow rate was reduced to 10 µl min$^{-1}$ at 70 min and held to the end of the gradient. Proteomic raw files were analyzed with DIA-NN v.1.9 and Python v.3.9.7 was used for proteomic data analysis. Proteins with missing values were removed and $z$-score normalization was applied to the dataset before one-way ANOVA analysis. Uniform Manifold Approximation and Projection (UMAP) analysis was performed with the Python packages 'UMAP'.

## Synthetic guide RNA design

Synthego Knockout Guide design was utilized to generate synthetic guide sequences targeting exon 1 of the *Cmpk2* gene. The top two synthetic guide sequences generated by the software were purchased from Synthego and targeting efficiency was assessed via Synthego's Inference of CRISPR Edits analysis platform and immunoblotting. Guides that produced indel scores of 50 or above were used for experimentation. Negative control guides targeting the Rosa26 allele were also purchased from Synthego. The guide sequences were as follows: guide 1, ATGGCCCTAATAAGCCGCCC; guide 2, CCCTCGAGCCCCACTGTGGC.

## CRISPR

CRISPR-Cas9 ribonucleoprotein editing of macrophage precursors isolated from murine bone marrow was performed as previously reported[62]. CRISPR-Cas9 ribonucleoprotein complexes were created by combining nuclease-free water, synthetic single guide RNA (Synthego, 1.5 nmol) and Alt-R Cas9 electroporation enhancer (IDT, 10 nmol) in a 1.5-ml tube. In a separate 1.5-ml tube, SpCas9-NLS (Macrolab, 40 pmol)

was diluted 1:5 in nuclease-free water. The diluted SpCas9-NLS was then added to the sgRNA/IDT enhancer mixture in a 1:1 ratio and incubated at room temperature for 10 min. Macrophage precursors were then electroporated using a Neon NxT Electroporation System (Invitrogen) with the following parameters: 1,900 V, 20 ms, 1 pulse. Electroporated cells were then transferred to 1.5-ml tubes containing pre-warmed macrophage growth medium, and incubated for 1 h at 37 °C, 5% CO$_2$ before collection.

## Phagocytosis and efferocytosis

Phagocytosis assays were performed on control or day 10 Sen(IR) BMDMs using heat-killed green labeled *Escherichia coli* particles (Abcam, cat. no. 235900). As per the manufacturer's protocol, $7.5 \times 10^5$ BMDMs were seeded onto six-well plates the day before the assay. Macrophages were exposed to *E. coli* particles at 5 µl per well, and were left to incubate at 37 °C for 45 min. After 45 min of incubation, macrophages were quenched with the provided quenching solution. Microscopy images were taken immediately after quenching at excitation and emission wavelengths of 490 and 520 nm respectively. For the efferocytosis assay, apoptotic bodies were generated using a murine colon cancer cell line (MC38). MC38 cells were exposed to 5-ethynyl-2′-deoxyuridine (EdU) at 10 µM for 24 h for labeling then subjected to 1% DMSO for 24 h. Apoptotic MC38 cells were collected, pelted, then frozen at −80 °C for 24 h. Apoptotic labeled MC38 cells were exposed to BMDM macrophages at a 1:5 MC38-to-BMDM ratio then left to incubate at 37 °C for 90 min. Macrophages were lifted with PBS EDTA, fixed in 4% paraformaldehyde for 15 min and permeabilized with saponin. Intracellular detection of EdU was performed using the Click-iT EdU AF647 Cytometry Assay Kit (Thermo Fisher) according to the manufacturer's directions.

## Arginase activity

BMDMs were stimulated with IL-4 10 ng ml$^{-1}$ for 24 h, then lysed with 75 µl of 0.1% Triton X-100 buffer with Halt protease inhibitor cocktail (no phosphatase inhibitor) (Thermo Fisher). Then 50 µl of 25 mM Tris–HCl and 10 µl of 2 mM MnCl$_2$ were added to each sample and heated at 56 °C for 10 min; 100 µl of 500 mM L-arginine (pH 9.7) was added then incubated for 45 min at 37 °C; and 800 µl of strong acid (H$_2$SO$_4$/H$_3$PO$_4$/H$_2$O 1:3:7) was used to stop each reaction. Urea production was measured by adding 40 µl of 9% α-isonitrosopropiophenone (in 100% ethanol) and heating to 100 °C for 15 min. A standard curve was run in parallel, and both the standards and samples were measured at 540 nm using a plate reader.

## SASP sample preparation and proteomic analysis of SASP

**SASP sample preparation for LC–MS/MS.** The conditioned media from the cell cultures under different tensions and senescence induction were concentrated to ~30 µl using 3 kDa molecular cut-off filters (Millipore Sigma) and protein content was quantified using the bicinchoninic acid assay. Aliquots of concentrated secretome (50 µg) from each sample were reduced using 20 mM dithiothreitol in 50 mM triethylammonium bicarbonate buffer (TEAB) at 50 °C for 10 min, cooled to room temperature, held at room temperature for 10 min, and alkylated using 40 mM iodoacetamide in 50 mM TEAB at room temperature in the dark for 30 min. Samples were acidified with 12% phosphoric acid to obtain a final concentration of 1.2% phosphoric acid. S-Trap buffer consisting of 90% methanol in 100 mM TEAB at pH ~7.1, was added and samples were loaded onto S-Trap micro spin columns. The entire sample volume was spun through S-Trap micro spin columns at 4,000g and room temperature, binding the proteins to the micro spin columns. Subsequently, S-Trap micro spin columns were washed twice with S-Trap buffer at 4,000g at room temperature and placed in clean elution tubes. Samples were incubated for 1 h at 47 °C with 2 µg of sequencing grade trypsin (Promega) dissolved in 50 mM TEAB. Afterwards, trypsin solution was added again in the same

amount, and proteins were digested overnight at 37 °C. Peptides were sequentially eluted from S-Trap micro spin columns with 50 mM TEAB, 0.5% FA in water and 50% ACN in 0.5% FA. After centrifugal evaporation, samples were resuspended in 0.2% FA in water and desalted with Oasis 10 mg Sorbent Cartridges (Waters). The desalted elutions were then subjected to an additional round of centrifugal evaporation and resuspended in 30 µl of 0.2% FA in water. Finally, indexed Retention Time standard peptides (iRT; Biognosys) were spiked in the samples according to manufacturer's instructions.

**Proteomic analysis of SASP (LC–MS/MS).** LC-MS/MS was performed on an Eksigent Ultra Plus nano-LC two-dimensional HPLC system (Dublin) combined with a cHiPLC system directly connected to an orthogonal quadrupole time-of-flight Sciex TripleTOF 6600 mass spectrometer). The solvent system consisted of 2% ACN, 0.1% FA in $H_2O$ (solvent A) and 98% ACN, 0.1% FA in $H_2O$ (solvent B). Proteolytic peptides were loaded onto a C18 pre-column chip (200 µm × 6 mm ChromXP C18-CL chip, 3 µm, 300 Å; Sciex) and washed at 2 µl min⁻¹ for 10 min with the loading solvent (0.1% FA in $H_2O$) for desalting. Peptides were transferred to the 75 µm × 15 cm ChromXP C18-CL chip, 3 µm, 300 Å (Sciex) and eluted at 300 nl min⁻¹ with the following gradient of solvent B: 5% for 5 min, linear from 5% to 8% in 15 min, linear from 8% to 35% in 97 min, and up to 80% in 20 min, with a total gradient length of 180 min. Samples were analyzed by DIA using 64 variable-sized windows covering the $m/z$ 400–1,250 range[63–65]. Masss spectrometry scans were collected with 250-ms accumulation time, and tandem mass spectrometry (MS/MS) scans with 45-ms accumulation time in 'high-sensitivity' mode. The collision energy for each segment was based on the $z = 2+$ precursor ion centered in the window with a collision energy spread of 10 or 15 eV.

**DIA data processing.** DIA–MS data were processed using Spectronaut, in-house customized spectral libraries (Buck Institute) and queries were performed against human databases (UniProt–SwissProt). Data extraction parameters were selected as dynamic using nonlinear iRT calibration. Identification was performed using 1% precursor and protein $q$ values. Quantification was based on the MS/MS peak areas of three to six best fragment ions per precursor ion, iRT profiling was selected, interference correction was selected and no normalization was applied. Differential protein abundance analysis was performed using an unpaired $t$-test, and $P$ values were corrected for multiple testing, using the Storey method[66,67]. Protein groups with at least two unique peptides, $q \leq 0.001$ and absolute $\log_2(FC) \geq 0.58$ were considered to be significantly altered.

### NAD metabolomics analysis
Pulverized liver extracts (10.0 mg) were processed using a boiling buffered ethanol (75% EtOH/25% 10 mM HEPES, pH 7.1) extraction protocol. Two solutions of internal standard mixes were prepared for independent processing. Mix A contained nicotinamide mononucleotide, nicotinamide riboside, nicotinic acid riboside, nicotinic acid, nicotinamide and 1-methylnicotinamide. Mix B contained nicotinic acid mononucleotide, nicotinamide mononucleotide and an aliquot of [¹³C]6-glucose-grown yeast extracts (12.5 µl per sample). Frozen liver extracts were removed from −80 °C storage and placed on dry ice until they were processed. Samples were processed in sets of five. Each sample received an aliquot of either mix A or mix B. Boiling buffered ethanol (400 µl) was added to each sample and samples were vortexed until the solution was thoroughly mixed, then placed on ice. After all samples had undergone this step, they were transferred to a water bath sonicator and sonicated for 1 min, then placed on ice again. Samples were further homogenized on a thermomixer at 55 °C for 3 min with constant shaking at 100 $g$. Samples were clarified by centrifugation at 16,200$g$ for 10 min in a prechilled centrifuge set to 4 °C. The clear supernatants were transferred to fresh tubes and dried

overnight in a refrigerated vacuum centrifuge set to 4 °C. The samples were resuspended in a 97% 10 mM ammonium acetate/3% ACN solution on the intended day of analysis. All samples were analyzed in positive ion mode on a Thermo Scientific TSQ Altis Plus triple quadrupole mass spectrometer coupled with a Thermo Scientific Vanquish Flex UHPLC system. Samples processed with mix A were analyzed using an acidic liquid chromatography separation and samples processed with mix B were analyzed using an alkaline separation. Samples were run on independent Thermo Scientific Hypercarb 100 × 2.1 mm 3 µm columns. Mix A samples were analyzed with mobile phases: A, 7.5 mM ammonium acetate in 0.1% FA in LC–MS grade water; and B, 0.1% FA in LC–MS grade ACN. Acidic separation was carried out at a flow rate of 0.5 ml min⁻¹ with the gradient: 0–2.25 min, 2% B; 2.25–12 min, 2%–25% B; 12–13.1 min, 25%–92% B; 13.1–14.3 min, 92% B; 14.3–14.4 min 92%–2% B; 14.4–18 min, 2% B. Mix B samples were analyzed with mobile phases: A, 10 mM ammonium in 0.1% ammonium hydroxide in water; and B, 0.05% ammonium hydroxide in ACN. Alkaline separation was carried out at a flow rate of 0.353 ml min⁻¹ with gradient: 0–1.8 min, 3% B; 1.8–10 min, 3%–34.5% B; 10–11 min, 34.5%–90% B; 11–13.2 min; 90% B; 13.2–13.3 min, 90%–3% B; 13.3–17.5 min, 3% B. Mass spectrometer source parameters were set to: 1,500 V ion spray voltage, sheath gas of 50, aux gas of 15, sweep gas of 1, collision gas pressure of 1.0 mTorr, ion transfer tube temperature 300 °C, 325 °C vaporizer temp, 0.7 first quadrupole full width at half maximum and 1.2 third quadrupole full width at half maximum. Instrument parameters were identical for both sets of analytes. Mass transitions and fragmentation parameters were optimized on this instrument before the analysis.

### Plasma lipids and metabolite measurements
Mice were fasted for 4 h. Plasma glucose and lipids were measured by colorimetric analysis as described previously[68]. Plasma insulin was measured using the mouse insulin ELISA kit (80-INSMS-E01) from Alpco[56].

### ALT and AST activity assays
Plasma ALT and AST activities were assayed using a kinetic colorimetric assay kit from Pointe Scientific according to the manufacturer's protocol. Transaminase activity was determined by the rate of decrease in NADH as measured by the change in absorbance at 340 nm.

### Statistics and reproducibility
Sample size and statistical analyses were determined before data collection. Sample sizes were chosen based on mouse availability and experimental feasibility. No statistical methods were used to pre-determine sample sizes for in vitro and in vivo studies. No data were excluded from the analyses. All in vitro experiments used at least three biological replicates to meet the minimum requirements for parametric statistical testing. The number of mice for each in vivo experiment was determined by mouse availability. Aged mice or and MASLD mice were pre-determined for in vivo experiments and removed from studies if mice contained tumors or became ill. All in vitro experiments used at least three biological replicates to meet the minimum requirement for parametric statistical testing. Data distribution was assumed to be normal, but this was not formally tested. In vitro experiments for transcriptomics, proteomics, metabolomics and SASP-proteomics required at least four to six biological replicates, which were determined by sample availability and sequencing efficiency. The number of mice for each in vivo experiment was determined by mouse availability and by prior similar studies with expected similar effect sizes. Power analysis was not performed to determine cohort size, as discussed above, instead the size was determined by animal availability and mouse number used in prior similar studies with expected similar effect size. In Fig. 3a, eight aged C57BL/6J male mice (22–24 months) and eight young mice were used to match the aged mice cohort size. In Fig. 3b, two aged (19 months) male livers were used based on liver

tissue availability and two young (3 months) male livers were used to match the aged mice cohort. In Fig. 3c, representative images of one mouse per age, aged male (24 months) and young male (3 months) was used based on tissue availability for this qualitative study. For in vivo experiments in Fig. 6f–k a sample of n = 12 C57BL/6J male mice (24 months) was determined by animal availability. Six aged mice were allocated per experimental condition (vehicle versus ABT). One vehicle control mouse died, therefore the final vehicle control in n = 5. A sample of n = 6 young male mice was determined to match the aged mice cohort size. In Fig. 7a–e in vivo experiments that involved HFHCD with the CTEP APOE mutant mice, 25 male mice per strain (8–10 weeks old) were used. In Fig. 7g–s 20 CTEP APOE mutant male mice crossed to C57BL/6J mice were generated across three cohorts. An even number of mice was allocated per condition as a vehicle control or treated with ABT-263. In Extended Data Fig. 1i–m, between six and ten male young (3 months) and aged (26 months) CD38 KO mice were used for in vivo gene expression experiments. The experiments were not randomized. Investigators were not blinded to allocation during experiments and outcome assessment. Statistical analyses were performed using the latest Prism (v.10.6.1) and R (v.4.5.2). The specific statistical tests used for each experiment are described in the figure legends. All tests were two-sided unless otherwise stated, and $P < 0.05$ was considered statistically significant. Individual data points are shown where appropriate to illustrate the distribution and variability of the data. All experiments were repeated independently three times with similar results. Detailed protocols and analysis methods are provided to ensure reproducibility.

### Reporting summary

Further information on research design is available in the Nature Portfolio Reporting Summary linked to this article.

## Data availability

Raw SASP-omics data and complete MS datasets were uploaded to the Center for Computational Mass Spectrometry, to the MassIVE repository at UCSD under the dataset identifier MSV000098058. The proteomics and metabolomics datasets generated and analyzed during this study have been deposited in the MassIVE repository under the dataset identifier MSV000098413. All transcriptomic data, both Fastq and counts data, are publicly available for download in the Gene Expression Omnibus (GEO) database under the following accession numbers: Fig. 1 (GSE318801), Fig. 2 (GSE318651, GSE318804), Fig. 4 (GSE318802), Fig. 7 (GSE319035, GSE318652) and Fig. 8 (GSE318806). Raw and normalized lipidomics data are included as supplementary data, source data are provided, and all other data supporting the findings are available from the corresponding author upon request. Source data are provided with this paper.

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

## Acknowledgements

Research reported in this publication was supported by the National Institute of General Medical Sciences of the National Institutes of Health Maximizing Investigators' Research Award (MIRA) award no. 1R35GM156893-01. This project was supported by funds provided by the Glenn Foundation for Medical Research and American Federation of Aging Research, Junior Faculty Award (20225528), and the UCLA-UCSD Diabetes Research Center Award funded by the National Institute of Diabetes and Digestive and Kidney Diseases (NIDDK) grant no. P30DK063491, and also supported by the National Institute of Diabetes and Digestive and Kidney Diseases (NIDDK), the Office of Disease Prevention (ODP), the Office of Nutrition Research (ONR), the Chief Officer for Scientific Workforce Diversity (COSWD), and the Office of Behavioral and Social Sciences Research (OBSSR) of the National Institutes of Health under award no. U24DK132746-01, UCLA LIFT-UP (Leveraging Institutional Support for Talented,

Underrepresented Physicians and/or Scientists). A.J.L. and S.H. were supported by NIH grant DK117850. G.T. is supported by the NIH Ruth L. Kirschstein National Research Service Award AI007323, T32 training grant 2T32AI007323-31. I.A.S.-P. was supported by the National Institute of General Medical Sciences of the National Institutes of Health under Award Number T32GM145388. J.G. is supported by the Ruth L. Kirschstein National Research Service Award F30 (NIA F30AG086001), the UCLA Tumor Cell Biology Training Program (NCI T32CA009056) and the UCLA-Caltech Medical Scientist Training Program (NIGMS T32GM008042 and NIGMS T32GM152342). C.M.K. and M.R.K. were supported by the National Institute of General Medical Sciences of the National Institutes of Health (GM037981 and GM61721). J.J.M. was supported by the American Heart Association Career Development Award (19CDA34760007). J.G.M. and Y.J. were supported by the NIGMS (R35GM142502). We acknowledge the support of instrumentation from the NCRR shared instrumentation grant 1S10 OD016281 (Buck Institute). We thank C.D. King for assistance with repository data uploads. The content is solely the responsibility of the authors and does not necessarily represent the official views of the National Institutes of Health. We thank Z. Robinson and I. Conboy for their guidance on the in vivo administration of ABT-263. While preparing this work, the authors used AI tools Grammarly and ChatGPT to edit the text for grammar and clarity. After using this tool/service, the authors reviewed and edited the content as needed and are fully responsible for the publication's content.

## Author contributions

Conceptualization of the study was undertaken by I.A.S.-P. and A.J.C. The methodology was developed by I.A.S.-P. and A.J.C. The investigation was undertaken by I.A.S.-P. (all wet lab experiments and bioinformatic analysis), I.A. (senolytic testing in vitro and in vivo), L.E. (in vitro senescence experiments), A.C.A. (in vitro senescence experiments), C.P. (CRISPR in vitro senescence experiments), A.D. (in vitro senescence experiments), G.T. (in vitro senescence experiments), C.Y.D. (tissue immunofluorescence experiments), R.H. (bioinformatics), J.G. (handled and provided aged mice), A.K. (in vitro senescence experiments), I.H. (machine learning imaging analysis), S.H. (provided MASLD mice), C.E. (provided MASLD mice), J.A.S. (in vitro senescence experiments), A.J.N. (in vitro senescence experiments), I.L. (in vitro senescence experiments), M.L. (in vitro senescence experiments), J.R. (in vitro senescence experiments), L.F. (in vitro senescence experiments), E.D.J.L.G. (NAD mass spectrometry), M.R.K. (immunofluorescence of in vitro senescence experiments), K.C. (liver microscopy), M.S. (liver microscopy), Y.J. (metabolomics), K.W. (lipidomics), M.S.-K. (in vitro senescence experiments), C.M.K. (immunofluorescence of in vitro senescence experiments), J.G.M. (proteomics), J.J.M. (immunofluorescence), C.B.

(NAD mass spectrometry), S.J.B. (lipidomics), C.L. (RNA-seq analysis), J.P.d.M. (RNA-seq analysis), B.S. (SASP-omics), R.S. (liver microscopy) E.V. (in vitro senescence experiments) and A.J.L. (provided MASLD mice). The original draft was written by I.A.S.-P. and A.J.C. All authors reviewed and edited the manuscript. The study was supervised by A.J.C. Funding was acquired by A.J.C.

## Competing interests

The authors declare no competing interests.

## Additional information

**Extended data** is available for this paper at https://doi.org/10.1038/s43587-026-01101-6.

**Correspondence and requests for materials** should be addressed to Anthony J. Covarrubias.

Ivan A. Salladay-Perez[1,2], Itzetl Avila[1], Lizeth Estrada [1], Andreea C. Alexandru[3], Cristian Ponce [1], Anika Dhingra[1], Grasiela Torres[1,2], Christina Y. Deng [1,2], Ronak Hegde[1], Julia Gensheimer [2,4], Abhijit Kale[3], Indra Heckenbach [5], Simon Hui[6], Chantle Edillor[6], Jose A. Soto [2], Alexander J. Napior[2], Isaiah Little [2], Mark Larsen[2], Jacob Rose [3], Lia Farahi[6], Edwin D. J. Lopez Gonzalez[7], Matthew R. Krieger [8], Kushan Chowdhury[6,9], Mridul Sharma[6,9], Yuming Jiang[10,11], Kevin Williams[12], Morten Scheibye-Knudsen [5], Carla M. Koehler[2,8,13], Jesse G. Meyer [10,11], Julia J. Mack [2,6], Charles Brenner [7], Steven J. Bensinger[1], Cyril Lagger[14], João Pedro de Magalhães [15], Birgit Schilling [3], Rajat Singh [9,16], Eric Verdin [3], Aldons J. Lusis [1,6,17] & Anthony J. Covarrubias [1,2,13,16,18]✉

[1]Department of Microbiology, Immunology, and Molecular Genetics, David Geffen School of Medicine at UCLA, Los Angeles, CA, USA. [2]Molecular Biology Institute at UCLA, Los Angeles, CA, USA. [3]Buck Institute for Research on Aging, Novato, CA, USA. [4]Department of Pathology & Laboratory Medicine, David Geffen School of Medicine at UCLA, Los Angeles, CA, USA. [5]Center for Healthy Aging, Department of Cellular and Molecular Medicine, University of Copenhagen, Copenhagen, Denmark. [6]Department of Medicine, Division of Cardiology, David Geffen School of Medicine at UCLA, Los Angeles, CA, USA. [7]Department of Diabetes & Cancer Metabolism, Beckman Research Institute of City of Hope, Duarte, CA, USA. [8]Department of Chemistry and Biochemistry at UCLA, Los Angeles, CA, USA. [9]Department of Medicine, Vatche and Tamar Manoukian Division of Digestive Diseases,

David Geffen School of Medicine at UCLA, Los Angeles, CA, USA. [10]Department of Computational Biomedicine, Cedars Sinai Medical Center, Los Angeles, CA, USA. [11]Advanced Clinical Biosystems Research Institute, Cedars Sinai Medical Center, Los Angeles, CA, USA. [12]Department of Biological Chemistry, David Geffen School of Medicine at UCLA, Los Angeles, CA, USA. [13]Jonsson Comprehensive Cancer Center at UCLA, Los Angeles, CA, USA. [14]Laboratory for Mechanistic Learning of Ageing Biology, Computer Science Department, Kyiv School of Economics, Kyiv, Ukraine. [15]Genomics of Ageing and Rejuvenation Lab, Department of Inflammation and Ageing, College of Medicine and Health, University of Birmingham, Birmingham, UK. [16]Comprehensive Liver Research Center at UCLA, Los Angeles, CA, USA. [17]Department of Human Genetics, David Geffen School of Medicine at UCLA, Los Angeles, CA, USA. [18]Broad Stem Cell Research Center, University of California, Los Angeles, USA. ✉e-mail: AJCovarrubias@mednet.ucla.edu

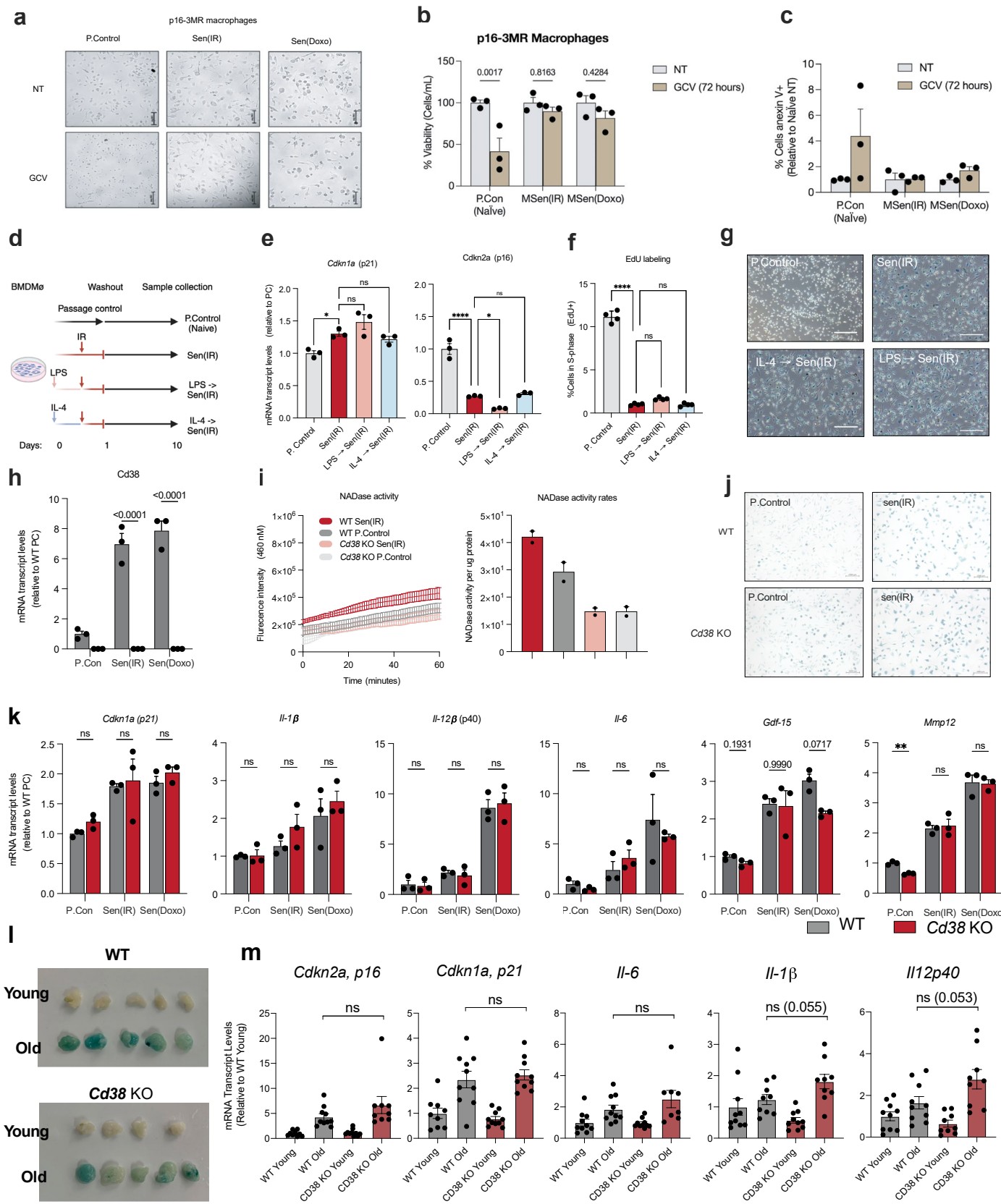

**Extended Data Fig. 1 | See next page for caption.**

**Extended Data Fig. 1 | P16, TLR4, IL-4 receptor signaling, and CD38 are not protective or required in DNA damage-induced macrophage senescence.**
(**a**) Phase contrast images from BMDM macrophages from p16-3MR mouse models. Control and senescent macrophages treated with ganciclovir (GCV) for 72 hours. P values represent post-hoc tests from a 2-way ANOVA. (**b**) Percent cells (live cells / dead cell gate) in liquid suspension after removal from tissue culture plates. (**c**) Percent cells stained for Annexin V (a known marker of apoptosis) relative to non-treated control macrophages. (**d**) Model of pre-polarization with LPS (100 ng/mL) and IL-4 (10 ng/mL) followed by Irradiation on macrophages. Analysis was performed 10 days post 10 Gy irradiation. (**e**) Mean, +/− SE, gene expression (RT-qPCR) normalized to control. N = 3 biological replicates, p-value from Tukey-test post-ANOVA analysis. (**f**) Click-it EdU labeling with propidium iodide (PI) labeling of cells to assess cell-cycle dynamics. Axes represent percent cells positive for stain compared to the parental gate detected by flow cytometry. (**g**) Senescence-associated beta-galactosidase images at 20x magnification.

(**h**) Cd38 mRNA transcript levels in response to senescent macrophage conditions in wild-type and Cd38 KO macrophages. (**i**) NADase assay from whole cell lysates. NADase activity was measured by epsilon-NAD cleavage, with fluorescence intensity monitored over 1 hour. Traces reflect the average of N = 2 biological replicates. Average NADase activity was determined from the slope of the trace from 15 to 30 min. (**j**) Senescence-associated beta-galactosidase images at 20x magnification for WT and Cd38 ko senescent macrophages. (**k**) Mean, +/− SE, gene expression (RT-qPCR) normalized to control. N = 3 biological replicates, p-value from Tukey-test post-ANOVA analysis. Control means passage control. (**l**) Whole tissue staining of Senescence-associated beta-galactosidase images for young (3 m) and old (24 m) VAT tissue against wild-type and Cd38 KO backgrounds. (**m**) Mean, +/− SE, gene expression (RT-qPCR) normalized to control. N = 10 biological replicates, p-value from Tukey-test post-ANOVA analysis. Control means passage control. Panel **d** created in BioRender; Salladay-Perez, I. https://biorender.com/6umuckd (2026).

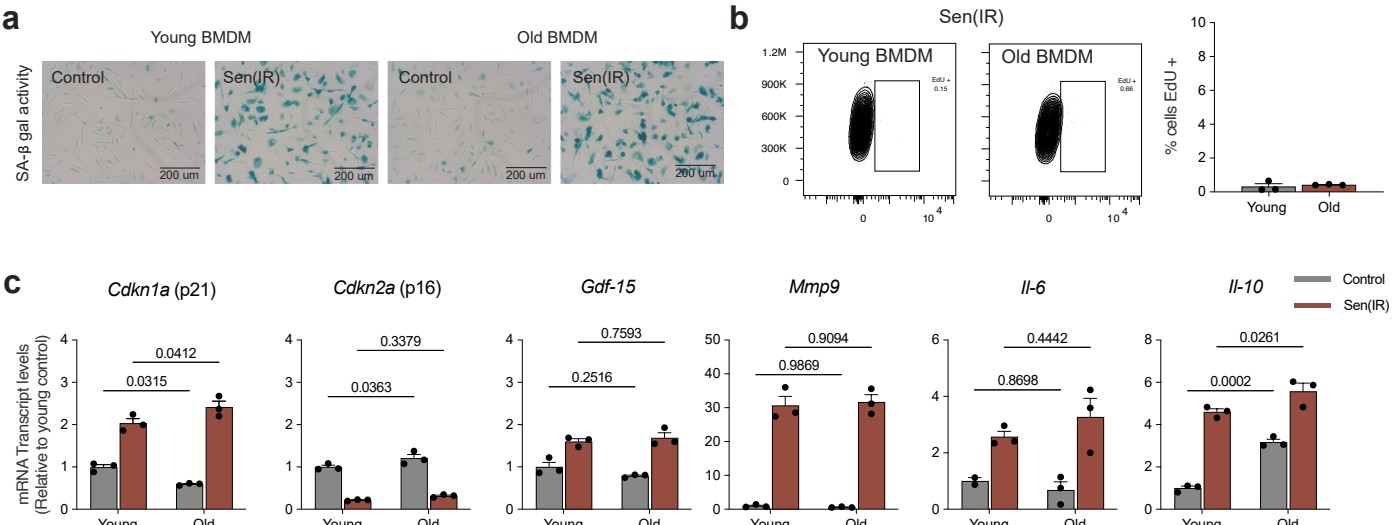

**Extended Data Fig. 2 | The effects of age on BMDM-derived senescent macrophages.** (**a**) Senescence-associated β-galactosidase assay images of young and old BMDM senescent macrophages at 20x magnification. (**b**) Click-it EdU labeling of cells to assess the percentage of cells in S-phase. Bar graph quantification is an average of three independent biological replicates. (**c**) Mean, +/− SE, gene expression (RT-qPCR) normalized to young control. N = 3 biological replicates, p-value from Tukey-test post-ANOVA analysis. P.Control means passage control.

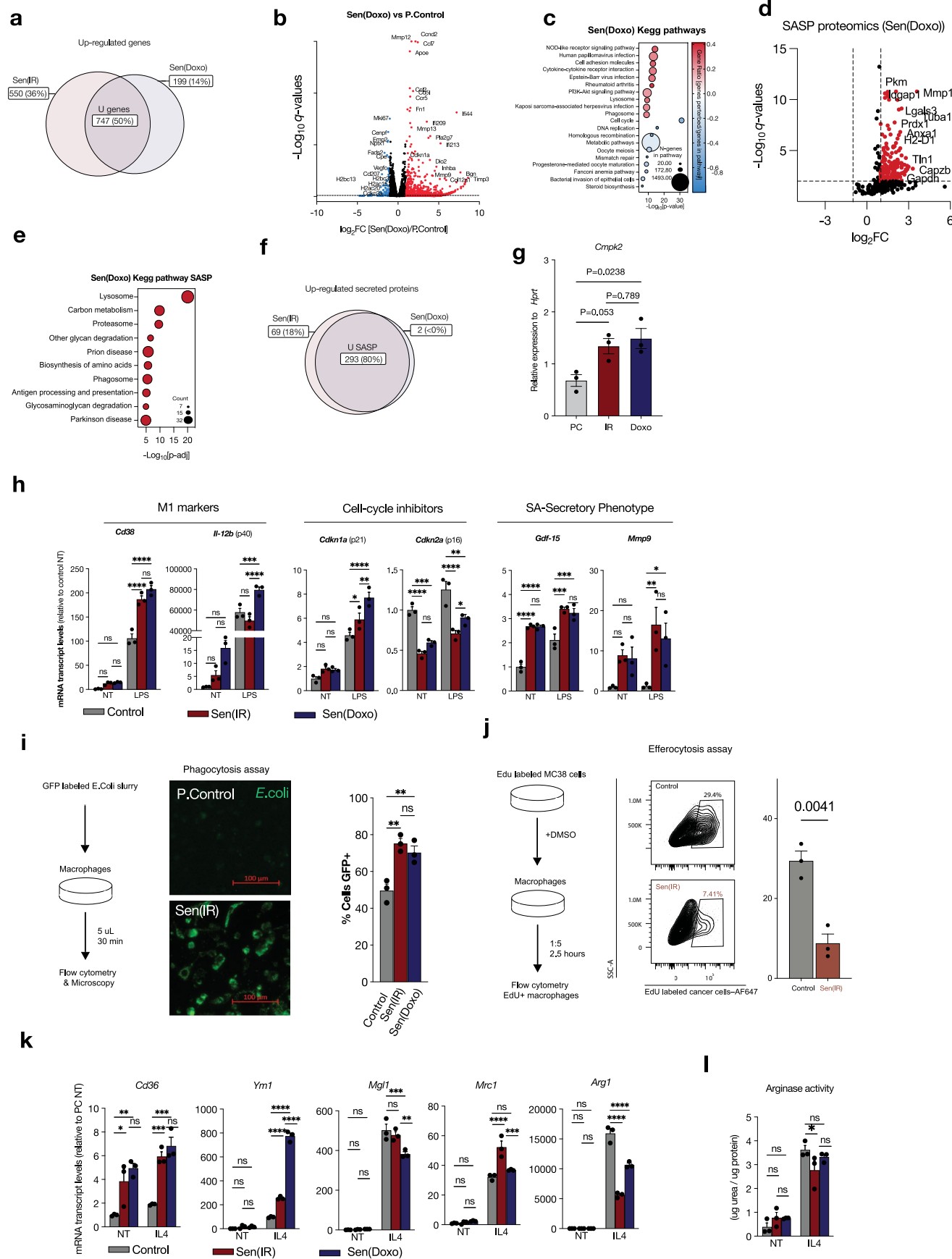

**Extended Data Fig. 3 | See next page for caption.**

**Extended Data Fig. 3 | Senescent macrophages are characterized by elevated interferon signaling and have augmented responses towards LPS. (a)** Venn-diagram analysis of all upregulated genes across irradiation and doxorubicin-induced senescence models. (**b**) Volcano plots are differentially expressed (log2FC > 1.3 and FDR < 0.05) between control and senescent Sen(Doxo) macrophages. (**c**) Kegg terms associated with the genes up/down-regulated by both Sen(Doxo) macrophages compared to control macrophages. (**d**) Volcano plot of upregulated (red) and downregulated (blue) proteins by FDR adj p-value and fold change from SASP-proteomics (SASPomics) mass-spectrometry between control and Sen(Doxo) macrophages. (**e**) Kegg terms associated with the proteins up-regulated by Sen(Doxo) over control macrophages. (**f**) Venn-diagram analysis of all upregulated SASP proteins across irradiation and doxorubicin-induced senescence models. (**g**) Relative mRNA transcript levels of Cmpk2 across passage control, Sen(IR) and Sen(Doxo) macrophages. Asterisks represent the p-value of ANOVA ≥ 0.05 (ns), <0.05 (✳), <0.01 (✳✳), <0.001 (✳✳✳),

<0.0001 (✳✳✳✳). (**h**) Relative mRNA transcript levels of M1 and senescent genes in response to LPS (24 hr 100 ng per mL) across P.Control, Sen(IR) and Sen(Doxo) macrophages. Asterisks represent the p-value of ANOVA ≥ 0.05 (ns), <0.05 (✳), <0.01 (✳✳), <0.001 (✳✳✳), <0.0001 (✳✳✳✳). (**i**) Phagocytosis assay using GFP-labeled E.Coli slurry. Images were taken at 40x magnification. Quantification of percent cells was performed using Image J software using 16-bit images. (**j**) Efferocytosis assay using apoptotic bodies derived from EdU-labeled MC38 cancer cells. Mean, +/− SE, EdU+ macrophages. N = 3 biological replicates, p-value from students t-test. (**k**) Relative mRNA transcript levels of M2 genes in response to IL-4 (24 hr 10 ng per mL) markers across P.Control, Sen(IR) and Sen(Doxo) macrophages. Asterisks represent the p-value of ANOVA ≥ 0.05 (ns), <0.05 (✳), <0.01 (✳✳), <0.001 (✳✳✳), <0.0001 (✳✳✳✳). (**l**) Arginase activity in cellular lysates as assessed by urea production. Asterisks represent the p-value of post-hoc 2-way ANOVA ≥ 0.05 (ns), <0.05 (✳), <0.01 (✳✳), <0.001 (✳✳✳), <0.0001 (✳✳✳✳).

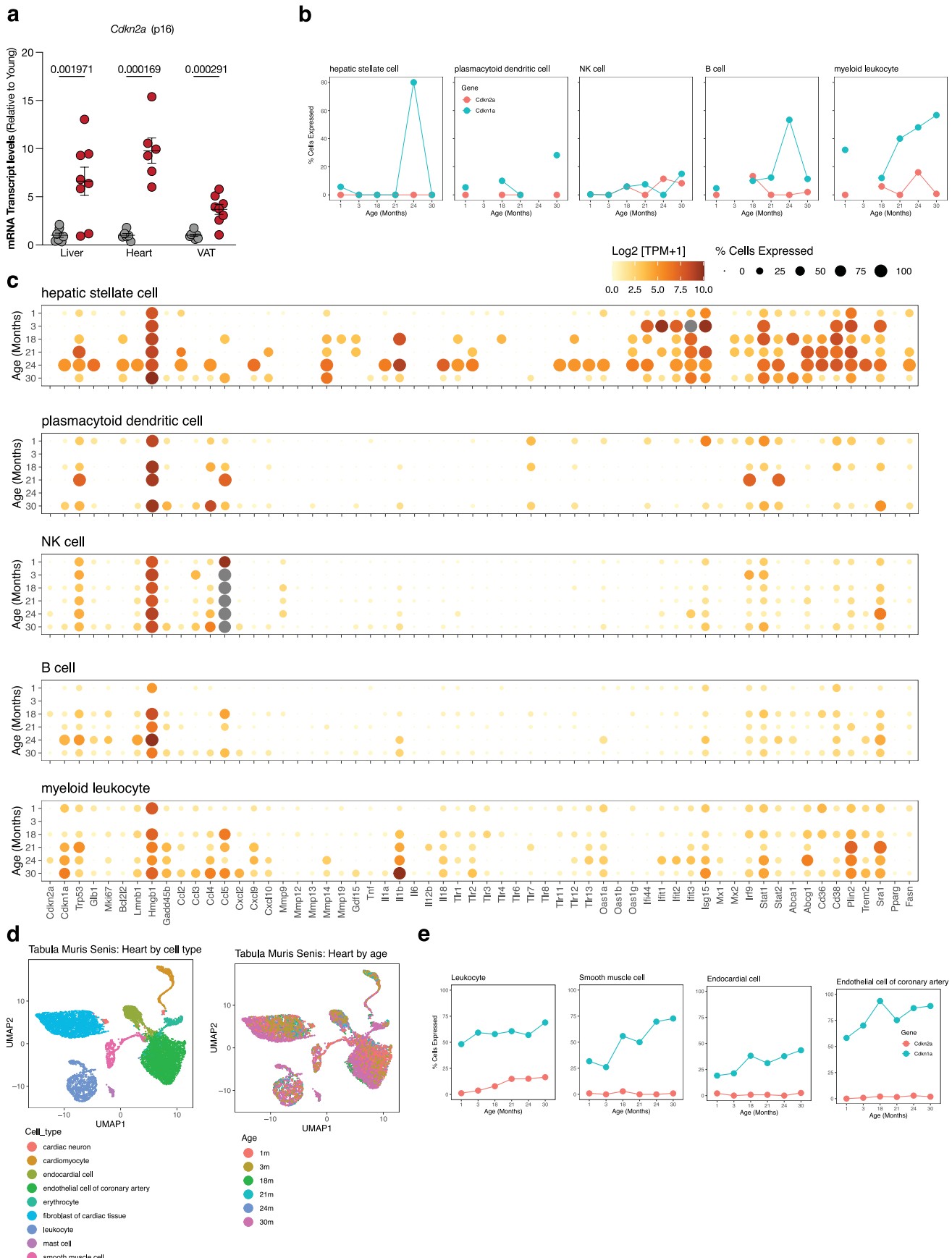

**Extended Data Fig. 4 | See next page for caption.**

**Extended Data Fig. 4 | Evaluation of senescence in aging liver cells and aging heart cells.** (**a**) Mean, and +/− SE, gene expression (RT-qPCR) of Cdkn2a from N = 5, 2–4 month young (Y) vs. 21–24-month-old (O) mice. Asterisks represent the p-value of t-test ≥0.05 (ns), <0.05 (✱), <0.01 (✱✱), <0.001 (✱✱✱), <0.0001 (✱✱✱✱). (**b**) The percent cell is positive for Cdkn1a (blue) and Cdkn2a (red) across the remainder of liver cells across age. (**c**) Bubble map of genes grouped by hallmarks or macrophage senescence. Size represents the fraction of cells expressing each gene, and color reflects average expression for the remainder of liver cells. (**d**) UMAP analysis of the Tabula Muris Senis data set representing all cells in the liver annotated by cell type. UMAP analysis of the Tabula Muris Senis data set representing all cells in the liver annotated by age. (**e**) The percent cell is positive for Cdkn1a (blue) and Cdkn2a (red) across all cells in the heart and aorta data set.

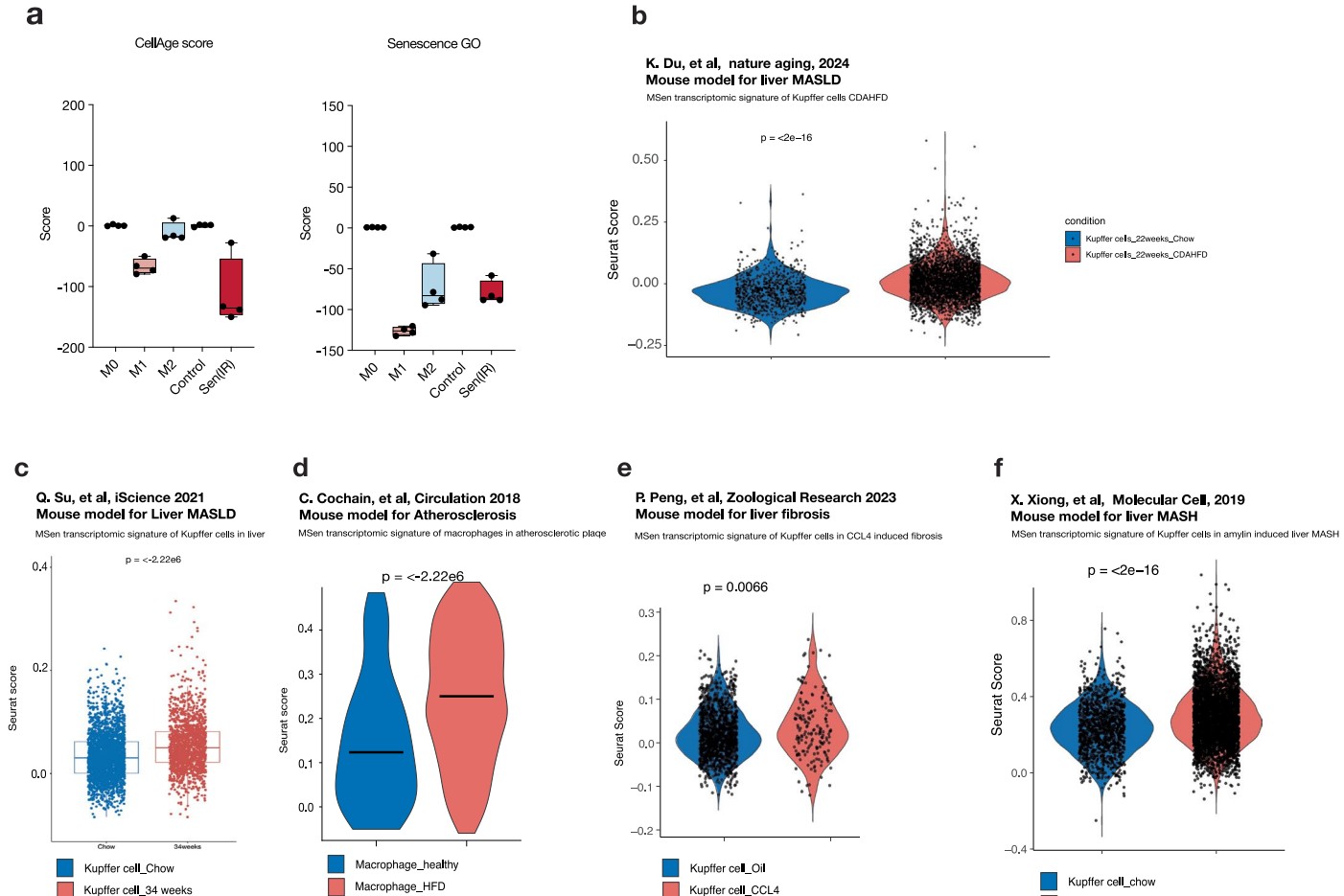

**Extended Data Fig. 5 | MSen transcriptomic signature increases in metabolic disease.** (**a**) GSVA scoring on M1, M2, Sen(IR), and control conditions of CellAge, gene ontology (GO) for senescence from the gene ontology database. (**b-f**) Median difference in the distribution of MSen Seurat score across all macrophages in MASLD liver disease and atherosclerosis. P-value represents results of a t-test. Included is the distribution of scores (x-axis) across all cells in data sets compared to the proportion of genes expressed in MSen signature (y-axis).

**a**

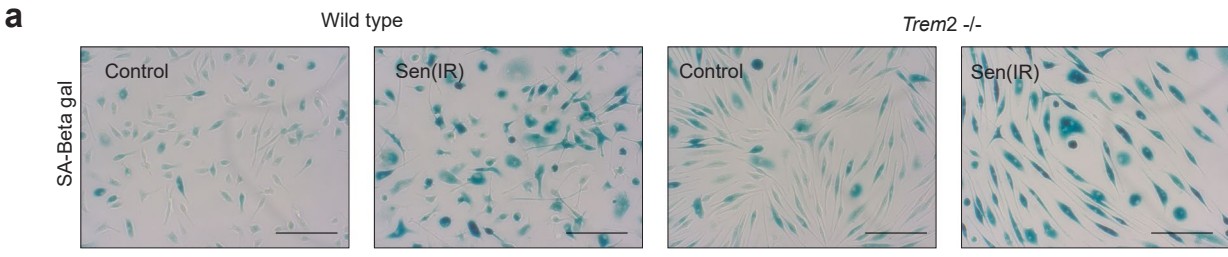

**b**

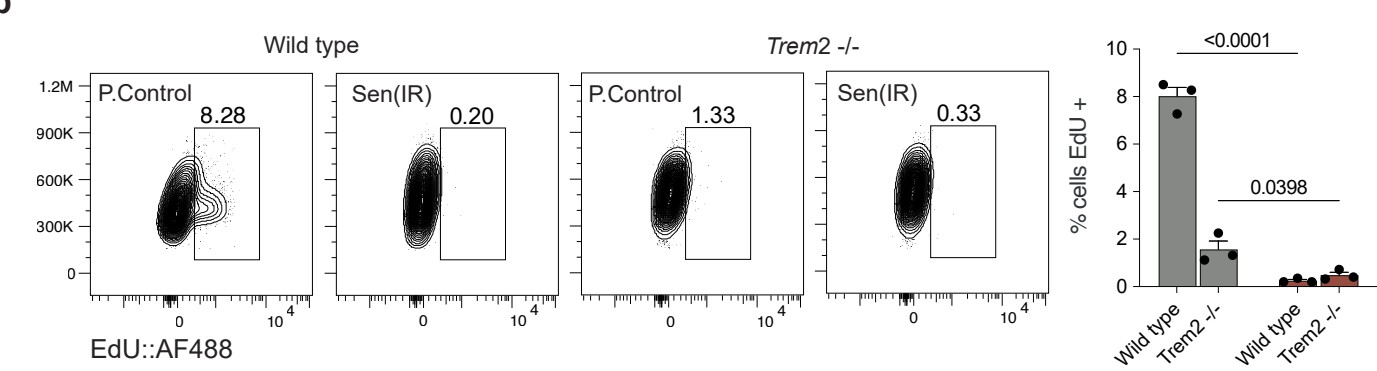

**c**

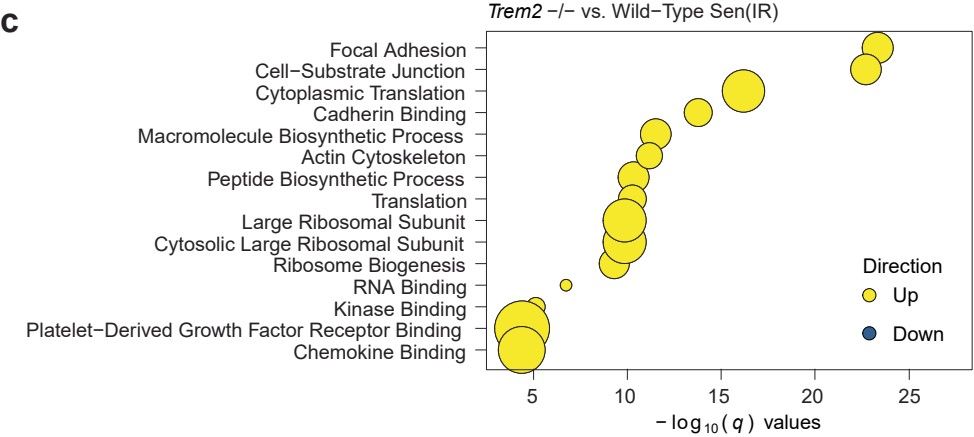

**Extended Data Fig. 6 | The effects of Trem2 −/− in senescent macrophages.**
(**a**) Senescence-associated β-galactosidase assay imaging for wild-type and
Trem2 −/− senescent macrophages at 20x magnification. Scale bars represent
20-micron distances. (**b**) Click-it EdU labeling of cells to assess % cells in S-phase.
Bar graph quantification is an average of three independent biological replicates.

N = 3 biological replicates, p-value from Tukey-test post-ANOVA analysis.
(**c**) Gene ontology analysis for statistically significant upregulated DEGs within
senescence in response to the Trem2 −/− background. The size of the data points
represents the relative odds ratio.

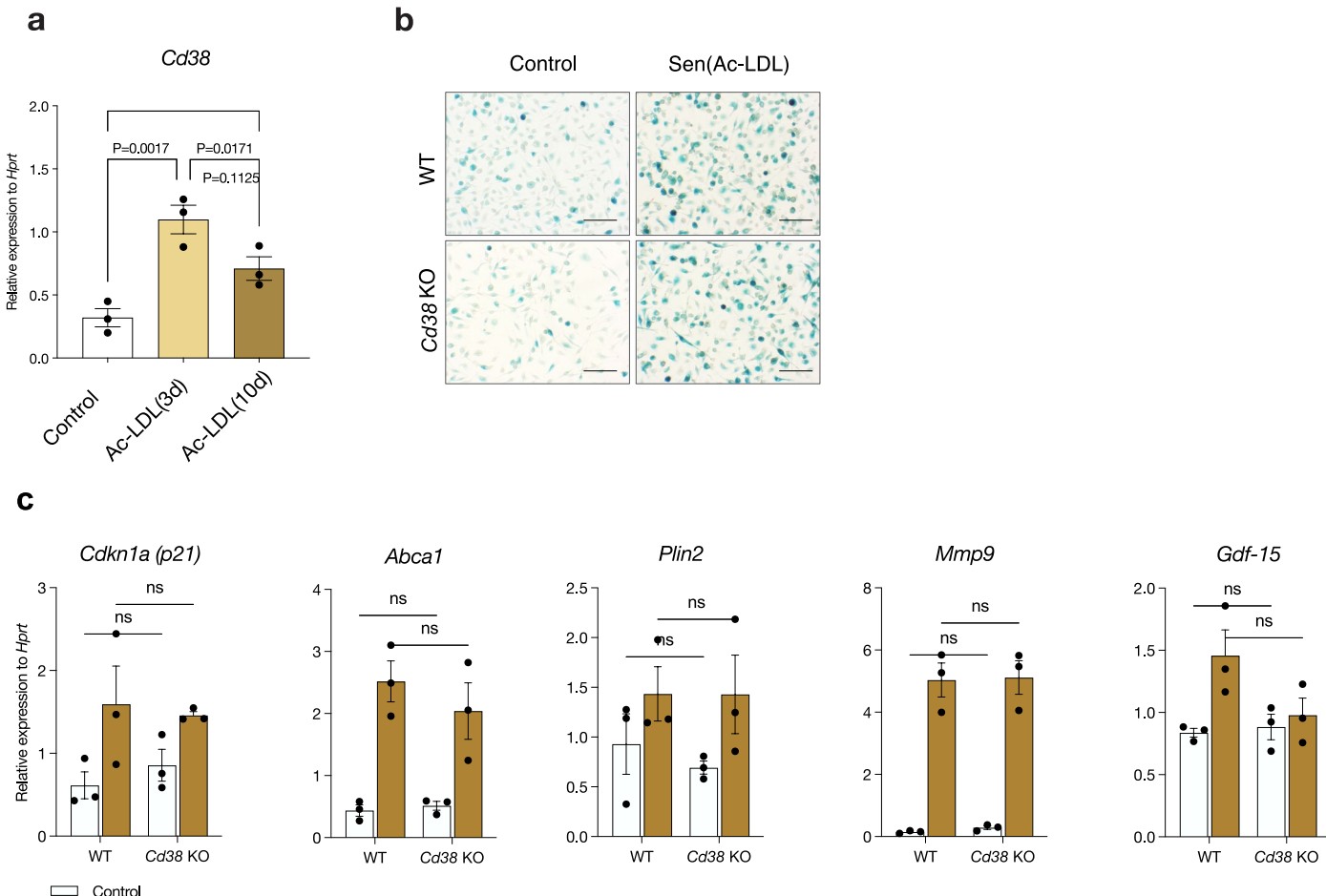

**Extended Data Fig. 7 | Effect of CD38 on Ac-LDL induced senescent macrophages.** (**a**) Relative Cd38 mRNA transcript levels across control, 3 days acLDL (foamy) and 10 days acLDL (sen). Asterisks represent the p-value of ANOVA ≥ 0.05 (ns), <0.05 (✱), <0.01 (✱✱), <0.001 (✱✱✱), <0.0001 (✱✱✱✱). (**b**) Senescence associated beta galactosidase assay imaging for Cd38 −/− macrophages compared to WT control at 20 x magnification. Scale bars represent 10 um. (**c**) Relative mRNA transcript levels for senescence and SASP genes across control and 10 days Ac-LDL (sen) for WT and Cd38 −/− genotypes. Asterisks represent the p-value of ANOVA ≥ 0.05 (ns), <0.05 (✱), <0.01 (✱✱), <0.001 (✱✱✱), <0.0001 (✱✱✱✱).

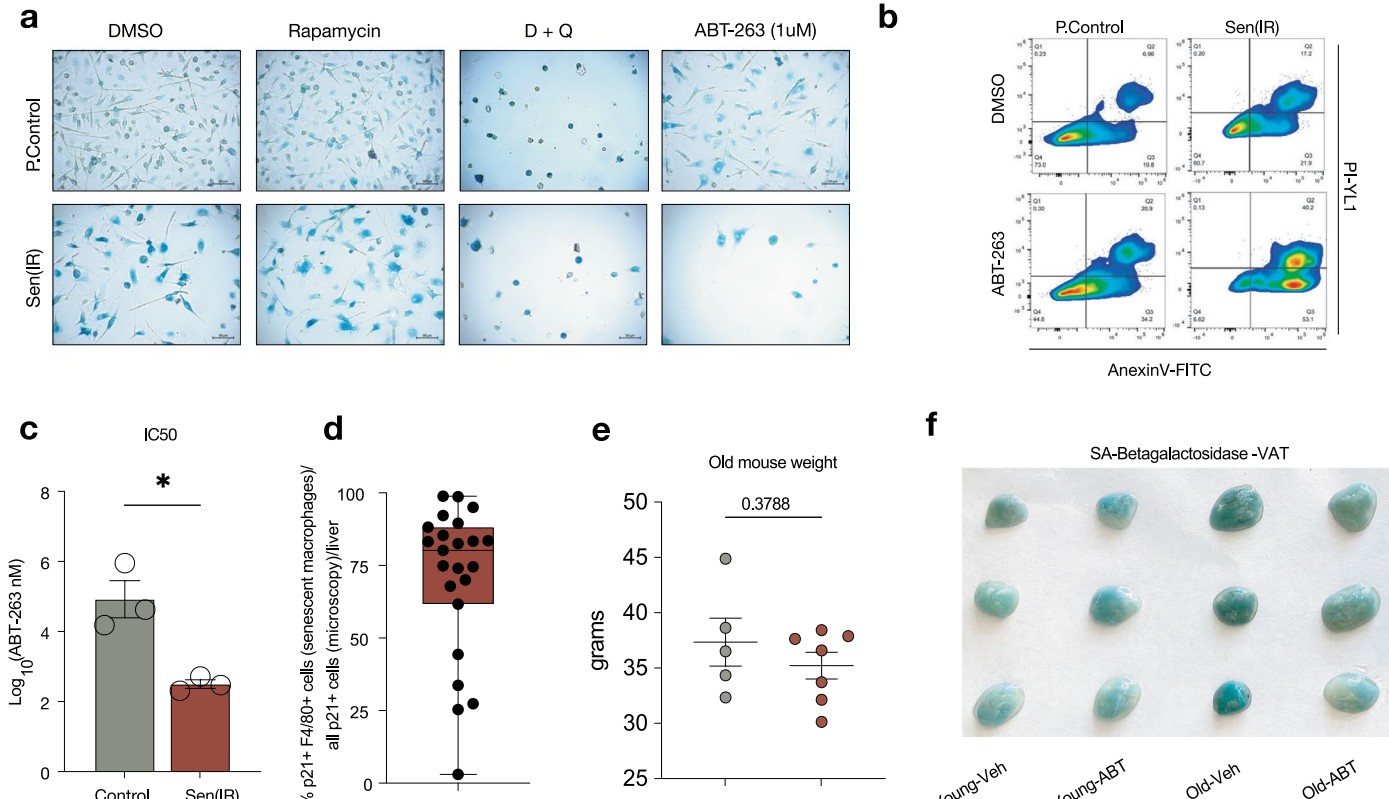

**Extended Data Fig. 8 | Comparative and systemic effects of ABT-263 as a senolytic. (a)** SA beta galactosidase assay imaging for BMDM macrophages at 20x magnification treated with DMSO control, Rapamycin (50 nM), D (15 uM) + Q (20 uM), and ABT-263 (1 uM) for 24 hours. Scale bars represent 20 uM distance. **(b)** Representative flow cytometry gates of BMDM macrophages treated with ABT-263 1 uM for 24 hours. Gates represent AV+ and PI+ cell populations. **(c)** Average IC50 quantified on nm scale. N = 3 biological replicates. Asterisks represent the p-value of t-test ≥0.05 (ns), <0.05 (✱), <0.01 (✱✱), <0.001 (✱✱✱), <0.0001 (✱✱✱✱). **(d)** P21 + F4/80+ (senescent macrophages) abundance relative to p21+ cells in n = 5-6 liver tissues per condition. P-value of Tukey test post ANOVA. **(e)** Average mouse weight +/− SE of N = 5–7 aged mice treated with Vehicle or ABT-263. **(f)** Whole tissue staining of SA beta-galactosidase images for young (4 m) and old (24 m) VAT tissue in response to ABT-263.

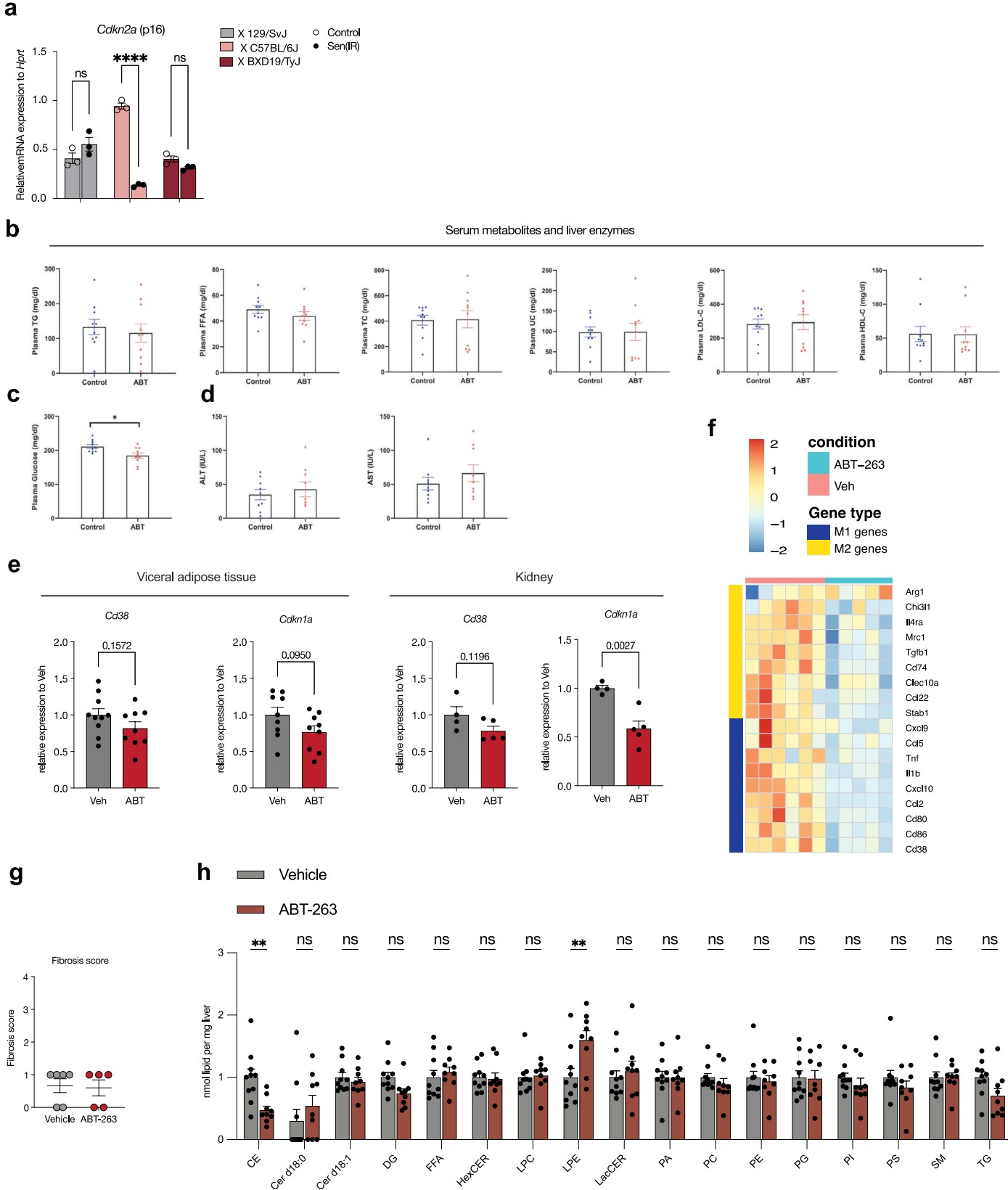

**Extended Data Fig. 9 | See next page for caption.**

**Extended Data Fig. 9 | Effects of ABT-263 on gene expression, blood tests, liver enzymes, metabolites, and lipids.** (**a**) Relative Cdkn2a mRNA transcript levels of BMDMs from CETP-APOE* Leiden mice crossed on different three backgrounds. Asterisks represent the p-value of ANOVA ≥ 0.05 (ns), <0.05 (✱), <0.01 (✱✱), <0.001 (✱✱✱), <0.0001 (✱✱✱✱). (**b-d**) Serum metabolite, serum lipid levels, and serum liver enzymes related to pathology MASH mice. ND was observed outside blood glucose. Asterisks represent the p-value of t-test ≥0.05 (ns), <0.05 (✱), <0.01 (✱✱), <0.001 (✱✱✱), <0.0001 (✱✱✱✱). (**e**) Relative mRNA transcript levels for inflammatory genes in bulk visceral adipose (VAT) and kidney samples. P-value represents the results of the t-test. (**f**) Bulk-RNAseq of the M1 and M2 genes in response to the senolytic drug, ABT-263. Scale bar represents Log2[TPM + 1] expression on a z-score axis. N = 6 Vehicle, and N = 5 ABT-263 treated mice used in this experiment. (**g**) Pathology graded histology reports for fibrosis on 5-6 MASH mice with and without ABT-263. (**h**) Fold change differences of normalized lipid abundance (nmol lipid per mg liver) for all lipid species via shotgun lipidomics (LC-MS/MS) relative to vehicle. Asterisks represent the p-value of t-test ≥0.05 (ns), <0.05 (✱), <0.01 (✱✱), <0.001 (✱✱✱), <0.0001 (✱✱✱✱).

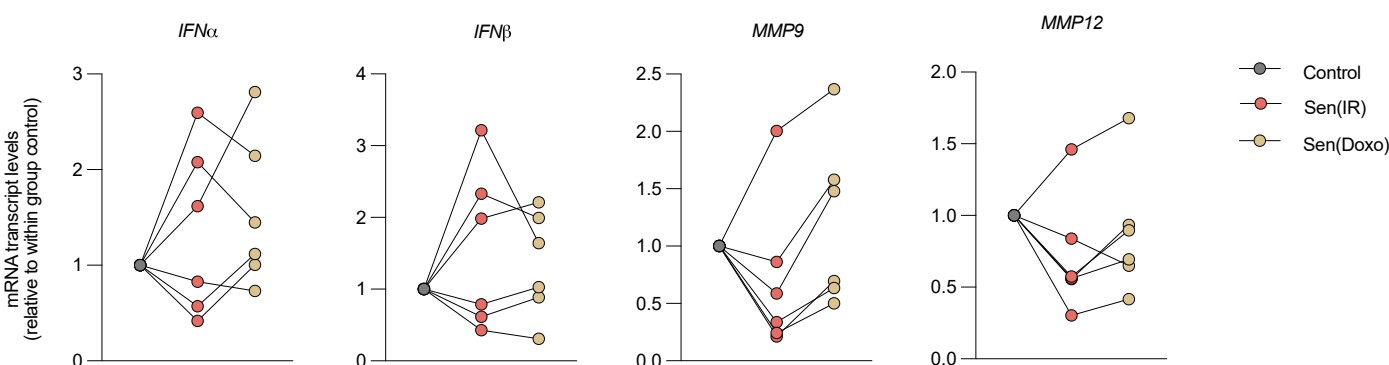

**Extended Data Fig. 10 | Senescent human macrophage SASP characterization by gene expression.** Relative gene expression (RT-qPCR) normalized to within donor control. N = 9 donors, no gene was significant by p-value statistics. Each line represents paired data points.

# Reporting Summary

## Statistics

For all statistical analyses, confirm that the following items are present in the figure legend, table legend, main text, or Methods section.

| n/a | Confirmed | |
|---|---|---|
| ☐ | ☒ | The exact sample size (*n*) for each experimental group/condition, given as a discrete number and unit of measurement |
| ☐ | ☒ | A statement on whether measurements were taken from distinct samples or whether the same sample was measured repeatedly |
| ☐ | ☒ | The statistical test(s) used AND whether they are one- or two-sided<br>*Only common tests should be described solely by name; describe more complex techniques in the Methods section.* |
| ☐ | ☒ | A description of all covariates tested |
| ☐ | ☒ | A description of any assumptions or corrections, such as tests of normality and adjustment for multiple comparisons |
| ☐ | ☒ | A full description of the statistical parameters including central tendency (e.g. means) or other basic estimates (e.g. regression coefficient) AND variation (e.g. standard deviation) or associated estimates of uncertainty (e.g. confidence intervals) |
| ☐ | ☒ | For null hypothesis testing, the test statistic (e.g. *F*, *t*, *r*) with confidence intervals, effect sizes, degrees of freedom and *P* value noted<br>*Give P values as exact values whenever suitable.* |
| ☒ | ☐ | For Bayesian analysis, information on the choice of priors and Markov chain Monte Carlo settings |
| ☐ | ☒ | For hierarchical and complex designs, identification of the appropriate level for tests and full reporting of outcomes |
| ☐ | ☒ | Estimates of effect sizes (e.g. Cohen's *d*, Pearson's *r*), indicating how they were calculated |

*Our web collection on statistics for biologists contains articles on many of the points above.*

## Software and code

Policy information about availability of computer code

| Data collection | Bio Rad CFX MAESTRO software 3.1, Attune NxT software v3.1.2, ZEISS 3.11, -xcalibur 4.0 |
|---|---|
| Data analysis | Image J 150i (western blot and microscopy), FlowJo v 10.1 (flow cytometry), R v4.5, RStudio 2025.05.1+513 (statistics and data science), Python is 3.13.5(data science and algorithms), Bio Rad CFX MAESTRO software 3.1(RT-qPCR), ZEISS 3.11 (microscopy). QuPath v0.6.0 (quantification). DESeq2 (v1.48.2). |

For manuscripts utilizing custom algorithms or software that are central to the research but not yet described in published literature, software must be made available to editors and reviewers. We strongly encourage code deposition in a community repository (e.g. GitHub). See the Nature Portfolio guidelines for submitting code & software for further information.

## Data

Policy information about availability of data

All manuscripts must include a data availability statement. This statement should provide the following information, where applicable:
- Accession codes, unique identifiers, or web links for publicly available datasets
- A description of any restrictions on data availability
- For clinical datasets or third party data, please ensure that the statement adheres to our policy

Raw SASPomics data and complete MS data sets were uploaded to the Center for Computational Mass Spectrometry, to the MassIVE repository at UCSD under the dataset identifier MSV000098058. The proteomics and metabolomics datasets generated and analyzed during this study have been deposited in the MassIVE

# Research involving human participants, their data, or biological material

Policy information about studies with human participants or human data. See also policy information about sex, gender (identity/presentation), and sexual orientation and race, ethnicity and racism.

| | |
|---|---|
| Reporting on sex and gender | NA |
| Reporting on race, ethnicity, or other socially relevant groupings | NA |
| Population characteristics | Peripheral blood mononuclear cells (PBMCs) were purified from the blood of anonymous healthy donors provided by UCLA's Virology core and research center. Per request we specified healthy males and females adults. The study only involved the use of human biological samples, obtained from the UCLA Virology and research center (Los Angeles, CA) and did not contain personal information about the identity, sex, or age of the donor and is therefore exempt from IRB approval. |
| Recruitment | Samples form human blood donors remained anonymous per UCLA's Virology Core and research center, Los Angeles, CA. |
| Ethics oversight | NA |

Note that full information on the approval of the study protocol must also be provided in the manuscript.

# Field-specific reporting

Please select the one below that is the best fit for your research. If you are not sure, read the appropriate sections before making your selection.

☒ Life sciences ☐ Behavioural & social sciences ☐ Ecological, evolutionary & environmental sciences

For a reference copy of the document with all sections, see nature.com/documents/nr-reporting-summary-flat.pdf

# Life sciences study design

All studies must disclose on these points even when the disclosure is negative.

| | |
|---|---|
| Sample size | The sample size and statistical analyses were determined prior to data collection. Sample sizes were chosen based on mouse availability and experimental feasibility. No statistical methods were used to pre-determine sample sizes for in vitro and in vivo studies. No data were excluded from the analyses. All in vitro experiments used at least 3 biological replicates to meet the minimum requirement for parametric statistical testing. In vitro experiments for transcriptomics, proteomics, metabolomics and SASPomics required at least 4-6 biological replicates were determined by sample availability and sequencing efficiency. The number of mice for each in vivo experiment was determined by the mouse availability. Power analysis was not performed to determine cohort size, instead the size was determined by animal availability and mouse number used in prior similar studies with expected similar effect size. In figure 3a, 8 aged C57BL/6J male mice (22-24 months) was collected based of animal availability and 8 young mice were used to match the aged mice cohort size. In figure 3b, 2 old (19 month) male livers were used based on liver tissue availability and 2 young (3 month) male livers were used to match the aged mice cohort. In figure 3c, representative images of one mouse per age, old male (24 months) and young male (3 months) was used based on tissue availability for this qualitative study. For in vivo experiments in figure 6(F-K) an N=12 C57BL/6J male mice (24 months) was determined by animal availability. Six aged mice were allocated per experimental condition (veh vs ABT). One vehicle control mouse died therefore the final vehicle control in N=5. N-6 young male mice was determined to match the aged mice cohort size. Figure 7 (A-E) in vivo experiments that involved HFHCD the CTEP APOE mutant mice, 25 male mice/strain (8-10 weeks old) were used. Figure 7(G-S) 20 CTEP APOE mutant male mice crossed to C57BL/6J mice were generated across 3 cohorts. An even number of mice were allocated per condition as a vehicle control or treated with ABT-263. Extended Data Figure 1(I-M), 6-10 male young (3 months) and old (26 months) CD38 KO mice used for in vivo gene expression experiments also determined by cohort availability. The experiments were not randomized. Investigators were not blinded to allocation during experiments and outcome assessment. Statistical analyses were performed using the latest Prism (v10.6.1) and R (v4.5.2). The specific statistical tests used for each experiment are described in the figure legends. All tests were two-sided unless otherwise stated, and a p-value < 0.05 was considered statistically significant. Data distribution was assumed to be normal, but this was not formally tested. Individual data points are shown where appropriate to illustrate the distribution and variability of the data. All experiments were repeated independently three times with similar results. Detailed protocols and analysis methods are provided to ensure reproducibility. |
| Data exclusions | No data exclusions were preformed. |
| Replication | All experiments presented in this manuscript was independently repeated 2-3 times and all attempts replicate findings made in this manuscript. |

| Randomization | The experiments were not randomized |
|---|---|
| Blinding | Blinding was not preformed because the experiments were not impacted by investigator bias. |

# Reporting for specific materials, systems and methods

We require information from authors about some types of materials, experimental systems and methods used in many studies. Here, indicate whether each material, system or method listed is relevant to your study. If you are not sure if a list item applies to your research, read the appropriate section before selecting a response.

## Materials & experimental systems

| n/a | Involved in the study |
|---|---|
| ☐ | ☒ Antibodies |
| ☐ | ☒ Eukaryotic cell lines |
| ☒ | ☐ Palaeontology and archaeology |
| ☐ | ☒ Animals and other organisms |
| ☒ | ☐ Clinical data |
| ☒ | ☐ Dual use research of concern |
| ☒ | ☐ Plants |

## Methods

| n/a | Involved in the study |
|---|---|
| ☒ | ☐ ChIP-seq |
| ☐ | ☒ Flow cytometry |
| ☒ | ☐ MRI-based neuroimaging |

## Antibodies

| Antibodies used | Target--Company--Catalog Number--Anti-(R=rabbit, r=rat)--Application
mouse beta-Tubulin--Cell signaling--2146S--R--western- dilution 1:1000
mouse Lamin B1--Cell signaling--13435T--R--western-dilution 1:1000
mouse P-Histone--H2AX--Cell signaling-- 9718T-- R--western-dilution 1:1000
mouse P-NF-kappaB p65--Cell signaling-- 3033T-- R--western-dilution 1:1000
mouse NF-kappaB p65--Cell signaling -- 8242T --R--western-dilution 1:1000
mouse p21 --Abcam --ab 188224-- R--western-dilution 1:1000
mouse TREM2-- Invitrogen --MA5-28223 --rat--western-dilution 1:1000
mouse CMPK2-- Abcam -- ab139720 -- R -- western-dilution 1:1000
mouse p16 -- Abcam -- 211542 - western-dilution 1:1000
mouse Tom20 -- proteintech -- 11802-1-AP- Immunofluorescence-dilution 1:1000
mouse dsDNA -- Abcam -- ab27156-1002 -- Immunofluorescence-dilution 1:1000
mosue F4/80 -- Abcam --  ab6640--rat -- Immunofluorescence-dilution 1:200
mouse p21 --Abcam --ab 188224-- R-- Immunofluorescence-dilution 1:200 |
|---|---|
| Validation | All antibodies for flow cytometry applications were validated with proper isotype controls using primary mouse or human macrophages. Western blot antibodies, specifically mouse p21, p16, Trem2, and Cmpk2, were further validated by detecting bands of the correct size and when possible using samples from knockout mice that lack the target protein. The p21 Abcam antibody listed above was specific to p21 protein by western and for immunofluorescence. P16 was also tested in a similar method, however no commercially available antibody passed internal validations using whole body p16 KO mice for immunofluorescence. Therefore p16 antibodies were used only for western blot analysis, validated on BMDMs from mice lacking p16 in myelolid cells (LysMCre x p16 fl/fl). There are no commercial validated antibodies for the mouse Cmpk2 antibody, therefore, CRISPR was used disrupt the Cmpk2 locus and the Abcam antibody was specific to the Cmpk2 protein (please see Figure 2m).  Our Trem2 antibody (Invitrogen MA5-28223) was tested against BMDMs from a Trem2 KO background and we found the Invitrogen antibody to be specific (please see figure 4f). |

## Eukaryotic cell lines

Policy information about cell lines and Sex and Gender in Research

| Cell line source(s) | L929 cells used to make conditioned media to grow BMDMs were obtained from ATCC |
|---|---|
| Authentication | Cells were authenticated by ATCC. L929 cells were further validated by the ability of the supernatant rom these cells to differentiate and grow mouse BMDMs. |
| Mycoplasma contamination | L929 stocks kept in the laboratory were analyzed for Mycoplasma contamination annually and tested negative. |

| Commonly misidentified lines (See ICLAC register) | NA |

# Animals and other research organisms

Policy information about studies involving animals; ARRIVE guidelines recommended for reporting animal research, and Sex and Gender in Research

| Laboratory animals | The following laboratory animals were used in vitro experiments. Briefly, 8-12 week male mice were sacrificed for bone marrow harvest then subjected to macrophage differentiation in vitro in the presence of mCSF. In figure 1-2; N=3 male C57BL/6J mice were used to perform experiments testing macrophage senescence. No power analysis was performed to pre-determine an N=3; this number was predetermined based on animal availability. Other B6 mice were used for In vivo testing for figure 3. In figure 3, N=16 male C57BL/6J mice was determined the cohort size based on animal availability. The experiment evaluated the expression of Cdkn1a, SA-Beta gal, and Immunofluorescence. Half the mouse cohort is young (2-4 months) the other half is old (22-24 months); N=8 mice per age group was used for statistical testing. Four heart samples (2 young, 2 old) were allocated for other experiments unrelated to this manuscript. In figure 4 C57BL/6J and Trem2 -/- mice were used to harvest bone-marrow-derived macrophages. N=6 C57BL/6J young (8-12 week) male mice were used for proteomic analysis (Figure 4a-c) based on animal availability. N=3 C57BL/6J (8-12 week) male mice was used as wild-type control for Trem2 -/- N=3 (8-12 week) male mice (figure 4f-j). N=3 male young (8-12 week) C57BL/6J was used to harvest bone-marrow-derived macrophages for lipidomics experiments and Ac-LDL in vitro experiments (figure 5a-c and figure 6a-e).

For in vivo studies the following mice were used. C57BL/6J and Cd38KO mice were from The Jackson Laboratory (Bar Harbor, ME cat#: 000664 & 003727). All transgenic mice expressing human CETP were obtained from The Jackson Laboratory, and mice carrying the human APOE*3-Leiden variants were kindly provided by Dr. L. Havekes. All mice were young male adults (2-4 months) or old males (22-24 months) and the genetic backgrounds include C57BL/6J, Cd38 KO, CETP-APOE mutant, BXD19, and 129X1. Mice were housed and maintained at UCLA animal facilities on a 12 hour dark/light cycle with ambient temperatures of 20-26C and the relative humidity of 30-70%. In figure 6f-k, N=23 C57BL/6J male mice (N=12 young and N=11 old) were predetermined sample size based on animal availability. N=6 mice were allocated for every group except for the old vehicle group where one mouse acquired the tumor. No power analysis was performed to determine the sample size. In figure 7a-d the generation and characterization of the F1 mice for this study were determined based on animal availability. N=25 three month CETP-APOE mutant, BXD19, and 129X1 mice were placed on a high fat high cholesterol diet to assess the impact of excess cholesterol on macrophage function and senescence in vivo. N=3-4 mice were harvested per time point and per genotype based on animal availability. In figures 7g-s, N=19 CETP-APOEx C57BL/6J male mice were used for senolytic testing. Sample size was not predetermined strictly due to animal availability. N=9 male mice were placed on a HFHCD and treated with Vehicle, N=10 male mice were given ABT-263. |
| Wild animals | No wild animals were used in this study. |
| Reporting on sex | All mice used for in vitro and in vivo studies were male. |
| Field-collected samples | NA |
| Ethics oversight | Mice were bred at UCLA in accordance with the Institutional Animal Care and Use Committee (IACUC). Animal studies were approved by the UCLA Animal Research Committee and IACUC. |

Note that full information on the approval of the study protocol must also be provided in the manuscript.

# Plants

| Seed stocks | NA |
| Novel plant genotypes | NA |
| Authentication | NA |

# Flow Cytometry

## Plots

Confirm that:

☒ The axis labels state the marker and fluorochrome used (e.g. CD4-FITC).

☒ The axis scales are clearly visible. Include numbers along axes only for bottom left plot of group (a 'group' is an analysis of identical markers).

☒ All plots are contour plots with outliers or pseudocolor plots.

☒ A numerical value for number of cells or percentage (with statistics) is provided.

## Methodology

| | |
|---|---|
| Sample preparation | All flow cytometry experiments were performed on in vitro macrophage cultures. Activated BMDMs or senescent BMDMs were lifted from non-TC treated plates after activation using cold PBS with 5mM EDTA for 5-10 min prior to blocking with FC block and staining with antibodies. Click-iT Edu 488 labeling (ThermoFisher) was used to measure macrophage proliferation as per manufacturer's protocol. Surface staining with other fluorochrome conjugated antibodies were performed for 30 minutes on ice at 1 ug for every mL each in PBS. Cells were then given an antibody safe fix solution, permeabilized, then subjected to intracellular staining on ice for 30 min in PBS. Cells were then moved for analysis on when the Attune NxT was performed. |
| Instrument | Attune NxT Acoustic Focusing Cytometer |
| Software | Attune NxT software v3.1.2 with quantification on FlowJo v10 |
| Cell population abundance | All flow cytometry experiments were done in vitro using BMDM macrophage cultures. Cell abundance was determined by the number of macrophages positive for FFCH and SSCA. |
| Gating strategy | Viable macrophage populations in vitro were determined using FSC/SSC at approximately FAC 400k/SSC 150-600K based on voltage 80 FSC and 310 SSC.<br><br>Single cells were determined looking at FSC-H and FSC-A of a height to area ratio of one to one 1:1. |

☐ Tick this box to confirm that a figure exemplifying the gating strategy is provided in the Supplementary Information.

