## [Peer Review File · Nature Aging]

P21+TREM2+-Senescent Macrophages Fuel Inflammaging and Metabolic Dysfunction-Associated Steatotic Liver Disease.

Corresponding Author: Dr Anthony Covarrubias

Version 0:

Reviewer comments:

Reviewer #1

(Remarks to the Author)

This manuscript presents a comprehensive and well-executed study that identifies p21⁺Trem2⁺ senescent macrophages as key contributors to chronic inflammation in aging and metabolic dysfunction-associated steatotic liver disease (MASLD). This manuscript addresses a key question in senescent cell biology: what is a senescent macrophage? Especially in light of prior findings by Hall et al. (Aging. 2017) showing that senescence-associated markers like β gal at pH6 are not exclusive to traditional senescent cells (SnCs)—they're highly expressed by a subset of F4/80⁺ macrophages that accumulate around SnCs in vivo. When human SnCs are implanted in mice (protected in alginate beads), these marker-positive macrophages are recruited, produce strong luminescence in p16-reporter mice, and are selectively eliminated by liposomal clodronate—unlike bona fide SnCs—revealing a previously overlooked macrophage-driven component of aging inflammation. The authors developed an in vitro model of DNA damage-induced macrophage senescence in both mice and human cells, employing a robust multi-omic approach to distinguish senescent macrophages from other activation states. Key findings include the demonstration that DNA damage and cholesterol accumulation drive a stable senescent phenotype characterized by elevated SASP, mitochondrial dysfunction, and interferon signaling. The study further validates the in vivo relevance of these cells by showing their accumulation in aged mouse livers—specifically in Kupffer cells—and enrichment in human cirrhotic liver samples. Importantly, the authors identify a distinct transcriptomic signature for senescent macrophages and demonstrate that these cells can be selectively depleted by the senolytic ABT-263 (Navitoclax), resulting in improved liver pathology in MASLD models. Overall, this work provides significant mechanistic insights and a compelling rationale for targeting senescent macrophages as a therapeutic strategy in age-related inflammatory diseases. However, additional experiments to clarify the role of p21 in senescent macrophages, further analysis of relevant endpoints, and addressing the following specific concerns are necessary before the manuscript is suitable for publication. Therefore, I recommend major revisions.

Major points

1. The manuscript presents a viewpoint that there are p16⁺ macrophages and p21⁺ macrophages and that they are potentially divergent in the implications for developing age-related pathology. It remains to be seen if these are discrete individual populations or rather do they exist upon some form of a continuum. Additional work to address whether any percentage of macrophages are double positive (p16⁺ p21⁺) would be helpful. For instance in Figure 3c, where macrophage (F4/80) and p21 expression are shown in young and old mouse white adipose tissue, it would be informative to know whether p16 also colocalizes similarly, or if there are p16⁺/p21⁺ double-positive cells. It would also be helpful if the authors could provide representative data to support their explanation.
2. Further clarity should be made around TREM2 and p21 in the different polarization states in the macrophages. For example in Figure 4e, TREM2 expression varies down in M1, up in M2, and is highly elevated in SenMacs, which also show a glycosylated TREM2 pattern. While the authors show that p21 is elevated across all three populations, it would be helpful to include flow cytometry plots showing double-positive cells for TREM2 and p21 in each population.
3. The authors state that long-term acLDL loading increases SA-β-Gal without inducing a foamy phenotype, suggesting a continuum from control to foamy to SenMacs, consistent with p21 protein but not transcript levels. It would strengthen the data if p16 expression were also shown.
4. In Figure 5f, the authors note that Atherosclerosis diet-induced MASH leads to varying fibrosis levels across strains, with 129 being resistant and BxD19 highly fibrotic—correlating with p21, TREM2, and SASP expression. It would be important to also assess whether p16 follows a similar pattern.
5. In Figure 5n, the authors report that ABT-263 treatment in B6 mutant mice reduces body weight, splenomegaly, inflammation, p21, M1 macrophage markers, and TREM2, but not p16. It would be helpful to also evaluate the effect on M2 macrophage markers (Il10, Arg).
6. In the 3MR SenMacs, is there any overlap between p16 and p21-positive cells when using staining (p21 RFP)?

7. Regarding functional assays for SenMacs, phagocytosis appears increased—what is the effect on efferocytosis? Additionally, what role does TREM2 play: is it pathogenic or a response to stress?
8. Does bariatric surgery reduce p21 levels in the livers of MASLD patients? Are there any publicly available datasets that could be mined to investigate this?
9. In Figure 6b, damaged macrophages from human PBMCs show increased p21 but not p16 with reduced proliferation. Could the authors also report TREM2, IFN α/β , and GDF15/MMP9 levels is possible? It would be interesting to see if the findings in mice are conserved in the primary human SenMacs.

Minor points that need to be addressed

1. It would be helpful if the authors could elaborate on the identity and characteristics of the M0 state shown in Fig. 1j–l.
2. In Figure 3b, the image of SA- β -Gal staining in the livers of old mice could be improved. Crystallization of the staining is apparent, and the image appears relatively darker. Brightening the image would enhance clarity. If a higher-quality or more representative image is available, it is recommended to replace the current one.
3. In Figures 5r and 5s, ABT-263 treatment improves MASLD scores and reduces lipid droplet size but does not affect fibrosis. Could the authors clarify the difference between the two scoring systems used? Additionally, the detailed methodology for activity scoring seems does not appear to be present in the Methods section and should be included for clarity.
4. Could the authors provide more details on the methods used for bacterial phagocytosis assays?
5. The authors state on line 341 that “Consistent with this, we found that p21⁺TREM2⁺ senescent macrophages exhibited greater lipid droplet accumulation compared to control macrophages (Fig. 4f-g).” However these are SenMacs in those panels and were not specifically isolated for p21⁺TREM2⁺ status. So the wording may need to be adjusted.

Reviewer #2

(Remarks to the Author)

I co-reviewed this manuscript with one of the reviewers who provided the listed reports. This is part of the Nature Aging initiative to facilitate training in peer review and to provide appropriate recognition for Early Career Researchers who co-review manuscripts.

Reviewer #3

(Remarks to the Author)

In the manuscript titled “p21+ TREM⁺-senescent macrophages fuel inflammation and disease in aging and metabolic dysfunction-associated steatotic liver disease”, Salladay et al. first identified p21+ TREM⁺-senescent macrophages as a key source of liver inflammation in natural aging and MASLD development in mice. The used transcriptomic profiling to identify a unique transcriptomic signature to characterize macrophage senescence both in vitro and in vivo for both mice and humans. Especially, Senescent macrophages are characterized by a SASP and a type 1 IFN response to mtDNA. Interesting, they found p21 as a more reliable marker for macrophage senescence compared to Cdkn2a (p16) and cholesterol esters and ceramides could trigger macrophages senescence in aging and MASLD liver. Lastly, they confirmed the senolytic agent ABT-263 (Navitoclax) selectively eliminated p21⁺ senescent macrophages both in vitro and in a MASLD/MASH model, causing reduced hepatic steatosis and systemic inflammation. In my opinion, Several aspects of the experiments can be improved as listed below to further support and bring relevance to their findings.

Major comments:

1. Indeed, a major limitation of this approach lies in our incomplete understanding of which cell types undergo senescence during liver aging and MASLD/MASH. In this study, using multidimensional experiments, particularly the established cellular models, have verified that senescent macrophages are triggered by cholesterol esters and ceramides and demonstrated the senescent macrophages especially p21+ TREM2⁺ macrophages contributed to the development and progress in liver aging and MASLD/MASH. However, several previous studies have revealed senescent Liver sinusoidal endothelial cells (LSECs) and senescent hepatocytes also play an important roles in pathology of MASH both in mice and human. In liver sections, please co-stained p21 or p16 with alpha-SMA (activated hepatic stellate cell), F4/80 (macrophages), NHF4 (hepatocyte), and CD31 (LSECs), respectively. Is there evidence of macrophages phagocytosing senescent cells in vivo, and what methods can be used to detect this phenomenon?
2. As for figure 3c, the authors should also provide the con-staining p16 and F4/80 in liver sections. In addition to p16 and p21 expression in liver macrophages, other hepatic cells also express senescence markers, as supported by at least some literature. How can we confirm that p21 primarily originates from liver macrophages? Beyond single-cell RNA sequencing, what additional evidence is required? Why is it not possible—or how can we rule out—that macrophages are merely phagocytosing other senescent cells and presenting their markers? Regarding the promotion of senescence, how should we interpret the intercellular relationships in vivo?
3. As for line 119, given that MASLD is primary driven by cholesterol accumulation----. in fact, altered lipid metabolism (mainly triglycerides) continues to stand out as a major factor contributing to the disease. Even that cholesterol has also emerged as a crucial player in MASH pathogenesis, and cholesterol accumulation within hepatocyte lipid droplets rather than macrophages (PMID: 40310463). Here, in my opinion, the authors found that cholesterol and ceramide lipids were enriched in p21⁺ senescent macrophages; and, excess cholesterol alone was sufficient to drive naïve macrophages into a senescent state. These critical findings subsequently directed research focus toward elucidating how cholesterol, both in vivo and in vitro, influences macrophage senescence and thereby drives alterations in hepatic lipid metabolism and liver aging. Therefore, the sentence should be modified to improve its accuracy and logical consistency.
4. As for figure 5, authors found that the senolytic agent ABT-263 (Navitoclax) selectively eliminated p21⁺ senescent

macrophages both in vitro and in a HFHCD-induced MASH model, causing reduced hepatic steatosis and systemic inflammation. While quantitative analysis of oil red O staining revealed a statistically significant difference in hepatic lipid droplet area between treatment and control groups, the effect size was modest. Would it be possible to incorporate complementary staining methodologies? I suggest that Liver sections were stained with BODIPY and Filipin. In addition, total cholesterol and triglycerides in livers should assess. Despite the alleviation of hepatic steatosis and inflammation by ABT-263, why was there no reduction in liver fibrosis? The protective effects of ABT-263 should be further validated across diverse diet-induced MASH models. Specifically, the authors could employ the CDAHFD-induced MASH mouse model to rigorously assess its therapeutic efficacy against liver inflammation and fibrosis. Whether the senescent macrophages promote activation hepatic stellate cells?

Minor

1. Please supply the full details of references 8, 42, 43, 50, 53, 54.
2. As for legend in figure 1b, immunohistochemistry should be immunoblotting? The same error also found in Figure 2m and in Figure 4o. please check legends of all figures again.
3. As figure 1c, left, exhibited the data from Sen (Doxo) group.
4. Please analysis the density of blotting (for example Fig. 1b, 2c, 4o, 6c) . In general, western blots are shown with only one animal. Immunoblot assays from animal tissues can be highly variable.
5. If macrophages undergo senescence rather than apoptosis, and senescent macrophages exhibit increased Bcl-2 expression, are there corresponding changes in Bax and Bad levels? Furthermore, are these Bcl-2-high macrophages also TREM2+P21+ cells?

Reviewer #4

(Remarks to the Author)

In this study the authors report that p21+Trem2+ senescent macrophages are critical as source of chronic inflammation in aging and in MASLD. Employing in vitro studies and mouse studies as well as multi-omics approaches the authors show that senescent macrophages accumulate in aging livers and are also enriched in human cirrhotic liver tissue. Moreover using transcriptomic profiling it was also shown that macrophages senescence is related with a unique transcriptomic signature. Lastly they also showed, that a targeting these macrophages may reduce signs of MASLD. The authors use a lot of cutting-edge approaches to determine changes in the macrophage population in aging liver. While they present a lot of data there are several issues with the preparation and the study design.

In the introduction a lot of the results are presented. Therefore, the results section is sort of repetitive. It would be beneficial to use the introduction to introduce the topic rather than reporting the results twice.

While the experiments shown in Figure 1 are very interesting and have high technical merits, an naïve old control is missing. Doxorubicin is a classical chemical to mimic senescence but actually it targets arginase activity which is indeed altered in aged cells but is not the only change and is down stream of other alterations. The same for all other approaches. This sort of is a problem in other experiments, too (Figure 2). Still having a good model of a senescent macrophage is or would be really helpful.

The finding that senescent macrophages are accumulating in aged tissues is not very surprising as both the expression of p21 and the number of macrophages have been shown to before to be higher in liver tissue but also other tissues of old mice (and humans).

It would be beneficial to also extent the experiments targeting senescent macrophages in MASLD to aging mice. Indeed, aging is a risk factor for MASLD.

Please include a mouse modes of MASLD without a high fat diet as proof of concept.

Please include data from humans with earlier stages of MASLD. Cirrhosis an end stage disease. Also, it would be beneficial to compare these data to data of old aged humans.

Line 65: Sentence unclear. Macrophages accumulated in Kupffer cells?

Version 1:

Reviewer comments:

Reviewer #1

(Remarks to the Author)

In the revised manuscript, the authors have fully addressed all of the raised concerns and questions. The manuscript has been sufficiently approved to warrant publication and thus I recommend acceptance. Lastly, could the authors confirm the catalog number for the p16 antibody in the reporting summary, as it does not appear in the Abcam antibody database.

Reviewer #2

(Remarks to the Author)

I co-reviewed this manuscript with one of the reviewers who provided the listed reports. This is part of the Nature Aging initiative to facilitate training in peer review and to provide appropriate recognition for Early Career Researchers who co-review manuscripts.

Reviewer #3

(Remarks to the Author)

Thank you for submitting the revised manuscript and your detailed point-by-point response to my comments. I have carefully reviewed the authors' responses and the corresponding changes made to the manuscript. I am pleased to confirm that the authors have addressed all my concerns thoroughly and thoughtfully. In particular, I appreciate the additional experiments and data analyses they have conducted. These new results directly and convincingly support their conclusions, effectively resolving the uncertainties I had raised. The revisions have significantly enhanced the manuscript's clarity, rigor, and overall quality. The study's scientific value is now more clearly demonstrated and well-supported. Therefore, I am satisfied with the revisions and recommend that the manuscript, in its current form, is suitable for publication.

Reviewer #4

(Remarks to the Author)

The authors addressed most of my concerns. I only have some minor comments:

Please check the following sentence.

Line 183: Next, CD38 is an ectoenzyme with NADase enzymatic activity and is the primary consumer of NAD⁺ in LPS-activated macrophages³⁶.

Response to reviews NATAGING-A09693

Salladay-Perez et al., 2025

We thank the editors and reviewers for their helpful comments and critiques, which have allowed us to revise and strengthen the data and conclusions in our manuscript. Please see below our responses to each individual point, including new data in response to your feedback.

Response to Editor's comments:

Thank you for submitting your Article, "p21+TREM2+Senescent Macrophages Fuel Inflammation and Disease in Aging and Metabolic Dysfunction-Associated Steatotic Liver Disease". It has now been seen by 4 referees (including one co-reviewer), whose comments are below. As you will see, the reviewers find your work to be of potential interest but they have raised substantial concerns that must be addressed. In light of these comments, we would be interested in considering a revised version that addresses these serious concerns and includes new data and analyses.

The reviewers make several useful comments and valuable suggestions for new experiments and analyses. When revising the study, we'd advise paying particularly close attention to the requests for further characterization of macrophage senescence (addressing for example the comments on p16, polarization, function and ruling out an effect of phagocytosis). Between R3 and R4, several additional MASLD models are suggested - we would not require you to add each new model and while data from aged mice would perhaps be the most interesting from the journal's perspective, we'd encourage you to simply select the most appropriate orthogonal approach to ensure the robustness of your conclusions. Please let me know if you'd like to discuss any of the reviewers' comments or your revisions in detail.

We therefore invite you to revise your manuscript taking into account all reviewer and editor comments. Please highlight any significant changes to the text with tracked changes to facilitate review of the revised manuscript (please provide the text as a Word file, not a pdf).

Response to referee #1

This manuscript presents a comprehensive and well-executed study that identifies p21⁺Trem2⁺ senescent macrophages as key contributors to chronic inflammation in aging and metabolic dysfunction-associated steatotic liver disease (MASLD). This manuscript addresses a key question in senescent cell biology: what is a senescent macrophage? Especially in light of prior findings by Hall et al. (Aging. 2017) showing that senescence-associated markers like β gal at pH6 are not exclusive to traditional senescent cells (SnCs)—they're highly expressed by a subset of F4/80⁺ macrophages that accumulate around SnCs in vivo. When human SnCs are implanted in mice (protected in alginate beads), these marker-positive macrophages are recruited, produce strong luminescence in p16-reporter mice, and are selectively eliminated by liposomal clodronate—unlike bona fide SnCs—revealing a previously overlooked macrophage-driven component of aging inflammation. The authors developed an in vitro model of DNA damage-induced macrophage senescence in both mice and human cells, employing a robust multi-omic approach to distinguish senescent macrophages from other activation states. Key findings include the demonstration that DNA damage and cholesterol accumulation drive a stable senescent phenotype characterized by elevated SASP, mitochondrial dysfunction, and interferon signaling. The study further validates the in vivo relevance of these cells by showing their accumulation in aged mouse livers—specifically in Kupffer cells—and enrichment in human

cirrhotic liver samples. Importantly, the authors identify a distinct transcriptomic signature for senescent macrophages and demonstrate that these cells can be selectively depleted by the senolytic ABT-263 (Navitoclax), resulting in improved liver pathology in MASLD models. Overall, this work provides significant mechanistic insights and a compelling rationale for targeting senescent macrophages as a therapeutic strategy in age-related inflammatory diseases. However, additional experiments to clarify the role of p21 in senescent macrophages, further analysis of relevant endpoints, and addressing the following specific concerns are necessary before the manuscript is suitable for publication. Therefore, I recommend major revisions.

Major points

1. The manuscript presents a viewpoint that there are p16+ macrophages and p21+ macrophages and that they are potentially divergent in the implications for developing age-related pathology. It remains to be seen if these are discrete individual populations or rather do they exist upon some form of a continuum. Additional work to address whether any percentage of macrophages are double positive (p16+ p21+) would be helpful. For instance in Figure 3c, where macrophage (F4/80) and p21 expression are shown in young and old mouse white adipose tissue, it would be informative to know whether p16 also colocalizes similarly, or if there are p16+/p21+ double-positive cells. It would also be helpful if the authors could provide representative data to support their explanation.

We thank the reviewer for this comment. The idea that senescent macrophages are composed of discrete p16+ or p21+ populations in tissues is an ongoing question in the lab, and we are currently preparing a follow-up manuscript that provides a mechanistic explanation for why p16+ and p21+ senescent macrophages are distinct. In response to the additional experiments, we quantified the proportion of double-positive (p16+/p21+) Kupffer cells in the liver. Our analysis revealed that the double-positive population represents only a small fraction of the total macrophage population in aged mice. The updated quantification has been added to the revised manuscript (Figure 3h). We are also working to identify a reliable antibody for p16 immunofluorescence staining in mouse tissue sections. However, thus far, none of the commercially available antibodies we have tested for IF have shown specific on-target binding in our hands. Therefore, we rely on transcriptional analysis, the p16 3MR mouse model, and an internally validated mouse monoclonal p16 antibody for western blot analysis to assess p16 expression. In contrast, we have validated a p21 antibody for immunofluorescence, and representative images of p21 staining in liver tissue are now included in Figure 3c and Figure 6g. Together, these data indicate that the majority of macrophages in aging tissues we analyzed represent distinct populations, either p16+ or p21+, whereas a very small subset of macrophages co-expresses p16 and p21. Our data is consistent with recent publications describing distinct p21+ or p16+ senescent cells (referred to as senotypes) in both human and mouse tissues, characterized by differential and non-overlapping SASP phenotypes (PMID: 41162753; PMID: 41270738). These citations are also added in the main text line 168 and line 585.

2. Further clarity should be made around TREM2 and p21 in the different polarization states in the macrophages. For example in Figure 4e, TREM2 expression varies down in M1, up in M2, and is highly elevated in SenMacs, which also show a glycosylated TREM2 pattern. While the authors show that p21 is elevated across all three populations, it would be helpful to include flow cytometry plots showing double-positive cells for TREM2 and p21 in each population.

This is a thoughtful suggestion. To address this, we performed flow cytometry experiments to examine TREM2 and p21 co-expression across M1, M2, and senescent wild-type macrophages. We observed an unexpected upregulation of TREM2 signal under M1 polarization that was inconsistent with our transcriptomic, proteomic, and western blot data. Suspecting the antibody is binding non-specifically. We performed the experiment again

and tested the widely used conjugated mouse TREM2-APC antibody using *Trem2* *-/-* BMDMs. This experiment showed that the antibody displayed non-specific binding even in the *Trem2* *-/-* background, confirming an unreliable non-specific flow cytometry signal. Given these findings, we decided to exclude the flow cytometry data from the revised manuscript and instead include an updated western blot (new Figure 4f) that incorporates *Trem2* *-/-* controls. This figure confirms the upregulation of TREM2 and p21 expression, in parallel to the downregulation of p16, in senescent macrophages. Importantly, no TREM2 signal was detected in the KO background, validating antibody specificity in the western blot assay. Interestingly, p21 transcript levels (Figure 4j) and protein expression (Figure 4f) were reduced in *Trem2* *-/-* senescent macrophages, suggesting a functional link between TREM2 and the senescent cell state. We further characterized the *Trem2* *-/-* senescent macrophages in Figure 4g–j and Supplementary Figure 6. Notably, M2 macrophages upregulated both p21 and p16, along with TREM2, and both markers were reduced in the *Trem2* *-/-* background. Since IL-4 stimulation has been reported to protect macrophages from chronic DNA damage (PMID: 38262419) and promote macrophage proliferation (PMID: 24101381), these changes in p21 and p16 are likely not senescence-related but instead reflect the essential roles of these proteins in basic macrophage biology, and are consistent with other studies (PMID: 28768895).

3. The authors state that long-term acLDL loading increases SA- β -Gal without inducing a foamy phenotype, suggesting a continuum from control to foamy to SenMacs, consistent with p21 protein but not transcript levels. It would strengthen the data if p16 expressions were also shown.

Thank you for highlighting this important point. To clarify, both short-term and long-term Ac-LDL treatments result in a foamy macrophage phenotype, but SA- β -Gal and other senescent features do not appear until day 10. To prevent confusion, we removed this terminology and instead describe only the day of Ac-LDL exposure for these experiments (Figure 5e). We also assessed p16 protein expression through Western blot and RT-qPCR in macrophages following Ac-LDL treatment. The updated data are shown in Figures 5f and 5i. Unlike p21, which is consistently upregulated in this model, these results indicate that p16 remains unchanged or is slightly downregulated in macrophages chronically treated with Ac-LDL. These findings suggest that p16 is not a reliable marker of senescence in cholesterol-induced macrophages, similar to the DNA damage and aging phenotypes observed in Figures 1i and 3h. Overall, these data strengthen the conclusion that chronic cholesterol ester loading promotes a senescent-like macrophage state that is independent of increased p16 expression.

4. In Figure 5f, the authors note that Atherosclerosis diet-induced MASH leads to varying fibrosis levels across strains, with 129 being resistant and BxD19 highly fibrotic—correlating with p21, TREM2, and SASP expression. It would be important to also assess whether p16 follows a similar pattern.

As suggested, we assessed p16 (*Cdkn2a*) expression in the RNA-sequencing dataset used in Figure 7d; however, we were unable to detect p16 mRNA expression. The sequencing was performed using single-end reads with average read depth, which limits the detection and accurate quantification of lowly expressed transcripts such as *Cdkn2a*. To investigate whether different strains impact p16 expression in senescent macrophages, we isolated BMDMs from the 129/SvJ, C57Bl/6J, and BxD19 strains and induced senescence using our Sen(IR) approach. The resulting data, shown in Supplementary Figure 10a, revealed that *Cdkn2a* expression varies among the strains but was either unchanged or significantly downregulated in senescent conditions. These results suggest a strong genetic influence on *Cdkn2a* expression but reinforce our conclusion that p16 is an unreliable marker for detecting senescent macrophages across multiple mouse strains, at least *in vitro*. More reliable tools to detect p16 expression in mouse tissues are needed to better

address this question in aging and disease settings, and we are currently exploring other options for future studies.

5. In Figure 5n, the authors report that ABT-263 treatment in B6 mutant mice reduces body weight, splenomegaly, inflammation, p21, M1 macrophage markers, and TREM2, but not p16. It would be helpful to also evaluate the effect on M2 macrophage markers (Il10, Arg).

Understanding the effect of ABT-263 on M1 and M2 signatures in the liver is crucial for assessing the drug's impact on other macrophage subsets or genes associated with them. To achieve this, we analyzed our RNAseq dataset shown in figure 7o. Although IL-10 was not detected in this dataset, we selected a broad set of well-known M2 and M1 genes, including Arg1. The updated data is shown in supplemental figure 10c. Besides M_{Sen} genes (Figure 7o), we found that both M2 and M1 signatures were downregulated following ABT-263 treatment, suggesting a general decrease in macrophage activation signatures in liver disease. However, interpreting these results is complex because M1 and M2 genes are also expressed by senescent macrophages (Figure 1i and 3k).

6. In the 3MR SenMacs, is there any overlap between p16 and p21-positive cells when using staining (p21 RFP)?

We appreciate the reviewer's thoughtful question regarding potential overlap between p16- and p21-positive cells in the 3MR SenMacs. Unfortunately, we no longer have access to the 3MR mouse line which prevents us from conducting additional staining or colocalization analyses. We agree that determining the degree of overlap between these senescence markers would provide valuable insight, and we will aim to incorporate in future studies using alternative p16 and p21 senescence-reporter systems.

7. Regarding functional assays for SenMacs, phagocytosis appears increased—what is the effect on efferocytosis? Additionally, what role does TREM2 play: is it pathogenic or a response to stress?

We thank the reviewer for raising this important point, and we agree that the additional experiments substantially strengthen the manuscript. To address this comment, we quantified efferocytosis using EdU-labeled apoptotic bodies. We found that despite their increased general phagocytic activity toward *E. coli*, Sen(IR) macrophages exhibit a reduced capacity for efferocytosis compared with control macrophages (Supp. Fig. 3j). Although the elevated TREM2 expression in SenMacs might suggest enhanced efferocytosis, we observed that WT TREM2⁺ senescent macrophages instead display impaired efferocytosis function. These findings align with prior reports showing that TREM2 shedding diminishes TREM2-dependent efferocytosis in chronically activated, pro-inflammatory macrophages found in livers from steatotic-NASH mice, cells that, as we demonstrate in this study, are likely senescent macrophages (PMID: 36521495).

To investigate the functional role of upregulated TREM2, we analyzed Trem2^{-/-} macrophages (Supplemental Fig. 6 and Fig. 4j). Loss of TREM2 reduced p21 protein expression (Fig. 4f) but did not alter SA-β-gal activity or cell-cycle arrest (Sup. Fig. 6a–b), indicating that TREM2 is not required for macrophages to enter the senescent state. Instead, TREM2 appears to reinforce p21 signaling and downstream responses, including SASP expression. Supporting this interpretation, RNA-seq of Trem2^{-/-} senescent macrophages revealed the strongest transcriptional effects in lysosomal and lysosomal-membrane genes, as well as cholesterol biosynthesis pathways (Fig. 4h). Moreover, TREM2 deletion significantly downregulated the M_{Sen} signature, affecting ~75% of its component genes (Fig. 4i–j). Although it remains challenging to determine whether TREM2 is pathogenic or instead induced as a compensatory stress response *in vivo* during aging or disease,

our *in vitro* data indicate that TREM2 primarily supports senescent-cell lipid metabolism and SASP production. These findings identify TREM2 as a compelling target for future studies aimed at reducing senescent macrophage function and inflammation.

8. Does bariatric surgery reduce p21 levels in the livers of MASLD patients? Are there any publicly available datasets that could be mined to investigate this?

We agree that it would be interesting to examine whether bariatric surgery reduces hepatic p21 expression in MASLD patients. However, despite an extensive search and consultation with the UCLA Liver Research Center, we were unable to identify any publicly available datasets containing liver biopsy transcriptomic or protein-level data from both bariatric and non-bariatric patients. We were informed by clinicians at UCLA that obtaining such tissue samples is inherently challenging due to the risks associated with liver punch biopsies in otherwise healthy individuals, which likely explains the lack of available data.

9. In Figure 6b, damaged macrophages from human PBMCs show increased p21 but not p16 with reduced proliferation. Could the authors also report TREM2, IFN α / β , and GDF15/MMP9 levels is possible? It would be interesting to see if the findings in mice are conserved in the primary human SenMacs.

There was variability in our initial experiments, so to strengthen this analysis, we increased the number of human PBMC donors from 4 to 9 and performed a paired within-donor analysis. We quantified TREM2, GDF15, and CD38 expression and additionally examined IFN α , IFN β , MMP9, and MMP12 to assess SASP features. Consistent with senescence induction, both irradiation and doxorubicin treatments upregulated CDKN1A (p21), GDF15, and CD38. Notably, TREM2 was robustly induced by irradiation but not by doxorubicin (Fig. 8b). Because doxorubicin preferentially targets highly proliferative cells, and human PBMC-derived macrophages are less proliferative than murine BMDMs, this difference may reflect distinct responses to the extent or type of DNA damage generated by each treatment. Analysis of IFN α , IFN β , MMP9, and MMP12 revealed donor- and treatment-specific differences, highlighting the variability of the human SASP (Sup. Fig. 11). These results, similar to our experiments using multiple strains of mice, suggest that strong genetic determinants influence senescent-cell phenotypes in macrophages derived from different human donors. Nonetheless, as captured in our RNA-seq and qPCR data, several senescence-associated genes are more consistently expressed across donors, including p21, TREM2, GDF15, and CD38 which show conserved induction in mouse and human senescent macrophages.

Minor points

1. It would be helpful if the authors could elaborate on the identity and characteristics of the M0 state shown in Fig. 1j-l.

An M0 macrophage is a naïve or unactivated macrophage derived using a standard 7-day BMDM differentiation protocol. M0 macrophages have not yet been polarized into a pro-inflammatory (M1) or anti-inflammatory (M2) state. In contrast, our passage (control) macrophages are M0 macrophages cultured for an additional 10 days alongside senescent macrophages. We clarify this on Line 173-4 and line 1226-9.

2. In Figure 3b, the image of SA- β -Gal staining in the livers of old mice could be improved. Crystallization of the staining is apparent, and the image appears relatively darker. Brightening the image would enhance clarity. If a higher-quality or more representative image is available, it is recommended to replace the current one.

We thank the reviewer for this helpful suggestion. While we tried to optimize our SA- β -Gal images in Figure 3b, the images presented are the most representative of the observed phenotype. We acknowledge that the SA- β -

Gal staining has some crystal formation; however, both young and old liver tissues were treated using the same conditions. Interestingly, we only see the SA- β -Gal activity and crystal formation in the old tissues. We are currently exploring ways to improve our SA- β -Gal tissue staining techniques and to develop alternative approaches, though this remains technically challenging.

3. In Figures 5r and 5s, ABT-263 treatment improves MASLD scores and reduces lipid droplet size but does not affect fibrosis. Could the authors clarify the difference between the two scoring systems used? Additionally, the detailed methodology for activity scoring seems does not appear to be present in the Methods section and should be included for clarity.

The scoring systems are derived from the MASH Clinical Research Network (PMID: 15915461). The fibrosis score is evaluated separately from the NAS or MASLD score. Total NAS score represents the sum of scores for steatosis, lobular inflammation, and ballooning, whereas the fibrosis score measures the amount of liver scarring ranging from F0-F4. We added the citation to the methods section for accessibility and clarity.

4. Could the authors provide more details on the methods used for bacterial phagocytosis assays?

We provided more details of the methods used for the bacterial phagocytosis assay as well as the efferocytosis assay of apoptotic bodies. You can find this new section in the methods pg 17 lines 762-773.

5. The authors state on line 341 that “Consistent with this, we found that p21⁺TREM2⁺ senescent macrophages exhibited greater lipid droplet accumulation compared to control macrophages (Fig. 4f-g).” However these are SenMacs in those panels and were not specifically isolated for p21⁺TREM2⁺ status. So the wording may need to be adjusted.

Thank you for raising this point. The previous sentence implies we isolated p21+trem2+ senescent macrophages when, in this case, it was bulk, so a mix of low- and high-expressing populations. You can find the updated sentence on pg. 8, lines 327-329: “Consistent with this, we found that senescent macrophages exhibited greater lipid droplet accumulation compared to control macrophages.”

Reviewer #2 (Remarks to the Author):

I co-reviewed this manuscript with one of the reviewers who provided the listed reports. This is part of the Nature Aging initiative to facilitate training in peer review and to provide appropriate recognition for Early Career Researchers who co-review manuscripts.

We thank Reviewer #2 for reviewing our manuscript and offering thoughtful suggestions to improve the clarity and quality of our work. We also value the involvement of Early Career Researchers in the peer-review process, which improves training while increasing transparency and fostering broader scientific engagement, ultimately benefiting our work and the field at large.

Response to referee #3

In the manuscript titled “p21+ TREM+-senescent macrophages fuel inflammation and disease in aging and metabolic dysfunction-associated steatotic liver disease”, Salladay et al. first identified p21+ TREM+-senescent macrophages as a key source of liver inflammation in natural aging and MASLD development in

mice. The used transcriptomic profiling to identify a unique transcriptomic signature to characterize macrophage senescence both in vitro and in vivo for both mice and humans. Especially, Senescent macrophages are characterized by a SASP and a type 1 IFN response to mtDNA. Interesting, they found p21 as a more reliable marker for macrophage senescence compared to Cdkn2a (p16) and cholesterol esters and ceramides could trigger macrophages senescence in aging and MASLD liver. Lastly, they confirmed the senolytic agent ABT-263 (Navitoclax) selectively eliminated p21⁺ senescent macrophages both in vitro and in a MASLD/MASH model, causing reduced hepatic steatosis and systemic inflammation. In my opinion, several aspects of the experiments can be improved as listed below to further support and bring relevance to their findings.

Major comments:

1. Indeed, a major limitation of this approach lies in our incomplete understanding of which cell types undergo senescence during liver aging and MASLD/MASH. In this study, using multidimensional experiments, particularly the established cellular models, have verified that senescent macrophages are triggered by cholesterol esters and ceramides and demonstrated the senescent macrophages especially p21⁺ TREM2⁺ macrophages contributed to the development and progress in liver aging and MASLD/MASH. However, several previous studies have revealed senescent Liver sinusoidal endothelial cells (LSECs) and senescent hepatocytes also play an important roles in pathology of MASH both in mice and human. In liver sections, please co-stained p21 or p16 with alpha-SMA (activated hepatic stellate cell), F4/80 (macrophages), NHF4 (hepatocyte), and CD31 (LSECs), respectively. Is there evidence of macrophages phagocytosing senescent cells in vivo, and what methods can be used to detect this phenomenon?

We appreciate the reviewer's comment on the importance of identifying which liver cells become senescent and the potential interactions between non-senescent macrophages and senescent cells during aging and MASLD. Unfortunately, we are unable to identify a reliable, commercially available p16 antibody suitable for immunofluorescence co-staining in mouse liver, which limits our ability to assess p16 expression in tissue sections. Identifying reliable p16 antibodies for mouse cells has been a major challenge in the field. We were fortunate that our colleague, Dr. Darren Baker, shared a p16 antibody that detects mouse p16 in western blots, which we validated using p16 KO cells. We have now used this antibody in multiple figures of the revised manuscript. Unfortunately, this antibody does not work for IF applications in either the Baker Lab or our lab. Addressing this question thoroughly will require using an alternative approach such as spatial RNA-Seq, which we are currently planning for a future study measuring senescence markers in aging and MASLD livers.

We thank the reviewer for raising the possibility that resident macrophages might internalize senescent cells *in vivo* and thereby express senescence-associated transcripts without themselves being senescent. However, our data in Figure 3, showing that macrophages are the primary cell type expressing mRNA transcripts of p21 (Cdkn1a) and not p16 (Cdkn2a), were derived from the publicly available dataset Tabula Muris Senis (PMID: 32669714). As discussed in the Tabula Muris Senis manuscript, the consortium employed strict quality control measures to include only single cells and avoid doublets, cell aggregates, and debris. Additionally, we implemented quality-control steps in our single-cell RNA-seq pipelines to exclude cells with ambient or contaminant RNA. While this does not preclude the possibility that macrophages might have ingested a highly p21⁺ expressing non-macrophage cell, the fact that the only highly p21⁺ expressing cells observed in the aging liver were macrophages makes this unlikely.

Furthermore, previous studies have reported that senescent cells often express CD47 and CD274 (PD-L1) to evade immune clearance, suggesting that phagocytic uptake by macrophages *in vivo* is unlikely (PMID: 36459066, PMID: 36323784, PMID: 39103548). Additionally, CD274 (PD-L1) is part of the 67-gene signature

that represents genes upregulated in Sen(IR) macrophages *in vitro* and aging Kupffer cells *in vivo* (Figure 3 j-k). This indicates that aging senescent macrophages avoid cell death via macrophage efferocytosis. We further tested whether senescent macrophages possess efferocytosis capabilities. In new data shown in Supp Figure 3j, we observed that senescent macrophages show a reduced capacity for dead cell clearance *in vitro*, consistent with other research indicating that aged macrophages lose basic effector functions (PMID: 38578825) and that macrophages in steatotic livers exhibit decreased efferocytosis (PMID: 36521495).

2. As for figure 3c, the authors should also provide the con-staining p16 and F4/80 in liver sections. In addition to p16 and p21 expression in liver macrophages, other hepatic cells also express senescence markers, as supported by at least some literature. How can we confirm that p21 primarily originates from liver macrophages? Beyond single-cell RNA sequencing, what additional evidence is required? Why is it not possible—or how can we rule out—that macrophages are merely phagocytosing other senescent cells and presenting their markers? Regarding the promotion of senescence, how should we interpret the intercellular relationships *in vivo*?

As noted in our response to the previous comment, we were unable to identify a commercially available p16 antibody suitable for immunofluorescence co-staining in mouse liver, which currently limits our ability to perform these analyses. However, we successfully performed co-staining for p21 and F4/80 in aged liver sections and found that approximately 50% of hepatic macrophages are p21⁺ on average (see figure 6h). Notably, these p21⁺ macrophages account for nearly 80% of p21⁺ cells in the aged liver (Supplementary Figure 8e), supporting the conclusion that macrophages are the predominant senescent cell population in this context.

Regarding the concern that macrophages may be phagocytosing senescent cells, please see our comments discussed above.

3. As for line 119, given that MASLD is primarily driven by cholesterol accumulation----. in fact, altered lipid metabolism (mainly triglycerides) continues to stand out as a major factor contributing to the disease. Even that cholesterol has also emerged as a crucial player in MASH pathogenesis, and cholesterol accumulation within hepatocyte lipid droplets rather than macrophages (PMID: 40310463). Here, in my opinion, the authors found that cholesterol and ceramide lipids were enriched in p21⁺ senescent macrophages; and, excess cholesterol alone was sufficient to drive naïve macrophages into a senescent state. These critical findings subsequently directed research focus toward elucidating how cholesterol, both *in vivo* and *in vitro*, influences macrophage senescence and thereby drives alterations in hepatic lipid metabolism and liver aging. Therefore, the sentence should be modified to improve its accuracy and logical consistency.

We agree that the prior text was not accurate, and that triglyceride (TG) and cholesterol ester (CE) metabolism are both key drivers of MASLD pathogenesis. Moreover, in response to prior feedback that lines 103–127 of the original manuscript were overly results-focused for the introduction, we removed this text and added more background, emphasizing the paper's primary goal: to determine whether macrophages can undergo senescence *in vitro* and *in vivo*, and whether they are a significant source of inflammation during aging.

4. As for figure 5, authors found that the senolytic agent ABT-263 (Navitoclax) selectively eliminated p21⁺ senescent macrophages both *in vitro* and in a HFHCD-induced MASH model, causing reduced hepatic steatosis and systemic inflammation. While quantitative analysis of oil red O staining revealed a statistically significant difference in hepatic lipid droplet area between treatment and control groups, the effect size was modest. Would it be possible to incorporate complementary staining methodologies? I suggest that Liver

sections were stained with BODIPY and Filipin. In addition, total cholesterol and triglycerides in livers should be assessed. Despite the alleviation of hepatic steatosis and inflammation by ABT-263, why was there no reduction in liver fibrosis? The protective effects of ABT-263 should be further validated across diverse diet-induced MASH models. Specifically, the authors could employ the CDAHFD-induced MASH mouse model to rigorously assess its therapeutic efficacy against liver inflammation and fibrosis. Whether the senescent macrophages promote activation of hepatic stellate cells?

We have updated the previous Figure 5t (now Figure 7s) to display statistics for all individual fields of view rather than averages per mouse, providing a more accurate representation of the variance of lipid droplet diameter and area. Although the reduction in lipid droplet size in the previous figure 5t appears modest, a small decrease in droplet size measured in diameter can result in a substantial decrease in total lipid volume, given the cubic relationship between diameter and volume. It is also important to note that the HFHC diet used in this study primarily represents MASLD rather than advanced MASH, which likely explains the absence of significant effects on fibrosis. The persistent presence of dietary lipids in this model maintains lipid droplets to a basal level, even following ABT-263 treatment. Our intention with these experiments was to assess the acute impact of senescent macrophage clearance on lipid accumulation and inflammation in cholesterol-driven fatty liver disease. We mention this model is a cholesterol-driven fatty liver disease because the CETP* Apoe mutant mice do not develop MASLD or MASH on a standard HFD. It's only when 1% cholesterol is added to the diet that they develop a fatty liver disease pathology (PMID: 29907965) therefore the term cholesterol-driven disease is used. To further address the reviewer's suggestions, we performed lipidomic profiling on liver tissues from this experiment (see Supp Figure 9e). These analyses revealed pronounced downregulation of cholesterol esters and a subset of triglycerides following ABT-263 treatment, supporting the conclusion that elimination of p21⁺ senescent macrophages remodels hepatic lipid metabolism. Additionally, we analyzed public datasets for orthogonal models of liver disease, including CDAA-HFD, CCl₄, and the Amylin diet models, and found a consistent increase in macrophage senescence (MSen) signatures in Kupffer cells across these datasets (see Supp Figure 5b-f). These findings reinforce the generalizability of our observations and support a broader role for senescent macrophages in liver lipid dysregulation and inflammation. We agree that future work should evaluate the efficacy of ABT-263 in more fibrotic MASH models and further explore whether senescent macrophages contribute to hepatic stellate cell activation. These studies are ongoing in our laboratory.

Minor

1. Please supply the full details of references 8, 42, 43, 50, 53, 54.

Thank you for the minor correction. We adjusted all citations that required proper formatting. Please see below.

8. Aravinthan, A. et al. Hepatocyte senescence predicts progression in non-alcohol-related fatty liver disease. *Journal of Hepatology* 58, 549–556 (2013).

42-> 44 Guan, X. et al. Microglial CMPK2 promotes neuroinflammation and brain injury after ischemic stroke. *Cell Reports Medicine* 5, 101522 (2024).

43 -> 45 The Tabula Muris Consortium, A single-cell transcriptomic atlas characterizes ageing tissues in the mouse. *Nature* 583, 590–595 (2020).

49 -> 48 Su, Q & K, Sun Y, et al. Single-cell RNA transcriptome landscape of hepatocytes and non-parenchymal cells in healthy and NAFLD mouse liver. *iScience* 24, 103233 (2021).

50-> 52 X. Xuelian, et al. Landscape of Intercellular Crosstalk in Healthy and NASH Liver Revealed by Single-Cell Secretome Gene Analysis. *Molecular Cell* 75, 644–660.e5 (2019).

53-> 60 Tall, A & WesterterpInflammasomes, W. Inflammasomes, neutrophil extracellular traps, and cholesterol. *Journal of Lipid Research* 60, 721–727 (2019).

54 -> 63 Hsieh, W. et al. Profiling of mouse macrophage lipidome using direct infusion shotgun mass spectrometry. STAR Protocols 2, 100235 (2021).

2.As for legend in figure 1b, immunohistochemistry should be immunoblotting? The same error also found in Figure 2m and in Figure 4o. please check legends of all figures again.

Thank you, we changed accordingly.

Figure 1 (b). SDS-PAGE gels and immunoblotting (western blot) with imageJ quantification relative to loading control.

Figure 2. (j) SDS-PAGE gels immunoblotting (western blot) of macrophages in response to days post IR. ImageJ quantification relative to loading controls are shown to the right.

Figure 4 (f) SDS-PAGE gels and immunoblotting (western blot) for TREM2 (both glycosylated and total), for p21, and p16. ImageJ quantification for band intensity are shown to the right and represent a relative intensity over the loading control.

3.As figure 1c, left, exhibited the data from Sen (Doxo) group.

Thank you for this suggestion. We added the Sen(Doxo) quantification (please see new figure 1c).

4.Please analysis the density of blotting (for example Fig. 1b, 2c, 4o, 6c) . In general, western blots are shown with only one animal. Immunoblot assays from animal tissues can be highly variable.

These data are derived from *in vitro*-grown BMDMs, which in our experience have a high degree of reproducibility across mouse donors. To ensure reproducibility, we derived BMDMs from young mice 8-12 weeks of age and using the same strain background C57BL/6J (including any mutant mice used in the study, besides experiments using BXD19 or 129 strains). These experiments are also representative of multiple experiments across many samples derived from the bone marrow of different mice donors. We do agree that direct WB analysis of tissues from mice is highly variable and should require multiple tissue samples from large cohorts of mice and statistical quantification. Please see prior studies where we have presented and quantified western blots from tissue samples derived from multiple old and young mice (PMID: 33199924).

As suggested, please see updated quantification for western blots in: Figure 1b, Figure 2J, Figure 4f, Figure 5f, & Figure 8c.

5.If macrophages undergo senescence rather than apoptosis, and senescent macrophages exhibit increased Bcl-2 expression, are there corresponding changes in Bax and Bad levels? Furthermore, are these Bcl-2-high macrophages also TREM2+P21+ cells?

Yes, the *in vitro* sen(IR) and sen(Ac-LDL) models express higher *Bcl2* mRNA transcripts relative to control conditions (see Figure 6a). We also investigated a panel of anti- and pro-apoptotic genes under these same conditions (see Figure 6a). Interestingly, *Bak1* (BAK) and *Bax* (BAX) mRNA levels are slightly greater in *in vitro* sen(IR) and sen(Ac-LDL) macrophages, but don't meet a significant p-value threshold in all conditions. This suggests that BCL-2 is preventing the activation of apoptosis, possibly via mitochondrial BAK/BAX formation, a hypothesis we tested with ABT-263 (see figure 6b).

Response to referee #4

In this study the authors report that p21+Trem2+ senescent macrophages are critical as source of chronic inflammation in aging and in MASLD. Employing *in vitro* studies and mouse studies as well as multi-omics approaches the authors show that senescent macrophages accumulate in aging livers and are also enriched in human cirrhotic liver tissue. Moreover using transcriptomic profiling it was also shown that macrophages senescence is related with a unique transcriptomic signature. Lastly they also showed, that a targeting these macrophages may reduce signs of MASLD. The authors use a lot of cutting-edge approaches to determine changes in the macrophage population in aging liver. While they present a lot of data there are several issues with the preparation and the study design.

Major comments

1. In the introduction a lot of the results are presented. Therefore, the results section is sort of repetitive. It would be beneficial to use the introduction to introduce the topic rather than reporting the results twice.

We thank the referee for their input on organizing the introduction. As discussed above, the original manuscript intro was overly results-focused. Therefore, we removed this text and added more background, emphasizing the paper's primary goal: to determine whether macrophages can undergo senescence *in vitro* and *in vivo*, and whether they are a significant source of inflammation during aging.

2. While the experiments shown in Figure 1 are very interesting and have high technical merits, an naïve old control is missing. Doxorubicin is a classical chemical to mimic senescence but actually it targets arginase activity which is indeed altered in aged cells but is not the only change and is downstream of other alterations. The same for all other approaches. This sort of is a problem in other experiments, too (Figure 2). Still having a good model of a senescent macrophage is or would be really helpful.

The primary goal in Figure 1 was to use BMDMs derived from young healthy mice and introduce DNA damage to determine whether macrophages can undergo senescence, and if so, what this phenotype looks like. We intentionally chose BMDMs from young mice because cells from older mice can acquire cellular damage and activate cell stress pathways (e.g., retrotransposon activation, genomic instability, telomere shortening, metabolic stress, etc.) that might interfere with or influence the inflammatory pathways and/or senescence traits. Once we confirmed that our irradiated and doxo-treated macrophages exhibited stable senescence, we validated in later Figures 3, 6, and 7 whether these phenotypes also reflect *in vivo* senescence in macrophages from aging and MASLD mice. In doing this, we developed a macrophage senescence (MSen) gene signature based on senescent macrophages *in vitro* and aged Kupffer cells in the liver (Fig. 3j). Notably, ~25% of upregulated DEGs in aged KCs overlapped with the MSen signature, indicating that liver-resident macrophages in aged mice share many genes associated with our *in vitro* model.

As suggested, we included new analysis to determine if the intrinsic age of the bone marrow affects the *in vitro* senescent macrophage phenotype. To do so, we aged mice to 23 months, generated bone marrow-derived macrophages (BMDMs), and induced senescence. In the newly added data (Supplementary Fig. 2), we observed no significant differences in hallmark senescence phenotypes between senescent BMDMs from young versus aged mice. These findings suggest that old BMDMs do not inherently possess any major aging traits that influence or worsen *in vitro* DNA damage-induced macrophage senescence phenotypes.

As mentioned, DNA damage may influence Arginase expression in senescent macrophages. However, we found that irradiation or doxorubicin treatment modestly increased arginase activity on its own, as *Arg1* mRNA

expression and enzymatic activity remained low and were significantly increased only in response to IL-4 (Supplementary Fig. 3k–l).

3. The finding that senescent macrophages are accumulating in aged tissues is not very surprising as both the expression of p21 and the number of macrophages have been shown to be higher in liver tissue but also other tissues of old mice (and humans). It would be beneficial to also extend the experiments targeting senescent macrophages in MASLD to aging mice. Indeed, aging is a risk factor for MASLD. Please include a mouse model of MASLD without a high fat diet as proof of concept.

We agree that macrophage accumulation and elevated p21 expression have been previously reported in aged tissues, including in our earlier manuscript (PMID: 33199924). However, the origins of p21 expression in these tissues remain unclear. Our study's goal was not to replicate this observation but to determine whether senescent macrophages are a major source of p21+ senescent cells and whether we can distinguish between age-associated inflammatory macrophages (M1-like) and truly senescent macrophages that also display M1-like and M2 features.

To address this, we first compared our *in vitro* model of senescent macrophages to classical M1 and M2 macrophages using RNA-seq (Fig. 1j–k). We found that senescent macrophages are distinct from M1 macrophages and as divergent from M1s as M2s are, demonstrating that macrophage senescence represents a unique activation state. Furthermore, senescent macrophages are M1-like because they express a SASP and exhibit a strong type I interferon response to cytosolic mtDNA (Fig. 3d, h, k–r). Importantly, only senescent macrophages, specifically Sen(IR) and Sen(Ac-LDL), undergo apoptosis in response to the senolytic ABT-263, whereas M1 and M2 macrophages do not (Fig. 6e). This highlights that the senolytic selectively targets senescence rather than classically activated macrophages. In response to the reviewer's suggestion, we have now included data from an aging mouse model to evaluate whether ABT-263 can reduce senescent cell burden in aged liver tissue and MASLD in aging mice. Consistent with our hypothesis, ABT-263 treatment resulted in approximately a 50% reduction in p21+ macrophages in the aged liver (Fig. 6f–h), a decrease in inflammatory markers related to Kupffer cells (Fig. 6i–j), and a reduction in MASLD in aging mice (Fig. 6k). These findings support the idea that aging and metabolic disease promote the accumulation of senescent macrophages in the liver and that clearing these cells may help mitigate age-related risk for metabolic disease and potentially other aging-related diseases.

4. Please include data from humans with earlier stages of MASLD. Cirrhosis an end stage disease. Also, it would be beneficial to compare these data to data of old aged humans.

To our knowledge, publicly available datasets comparing young and aged human liver macrophage populations do not currently exist. We consulted with clinicians at the UCLA Liver Research Center (which we are members of), who confirmed that such datasets are unavailable and that obtaining well-characterized young human liver samples is particularly difficult due to limited clinical indications for liver biopsy or resection in healthy young individuals (also, they pose a significant health risk). We agree that establishing such datasets would be highly valuable to the field and represent an important future direction. Therefore, in a future study, we plan to obtain liver samples from deceased donors, aged donors, and human liver samples from both early- and late-stage MASLD patients for spatial transcriptomic analyses. However, this project requires IRB approval, and we are in the early planning stages with assistance from collaborators at the UCLA Liver Research Center.

Minor comments

Line 65: Sentence unclear. Macrophages accumulated in Kupffer cells?

Thank you for the correction. We updated the abstract and removed this sentence to avoid confusion.

Response to reviews NATAGING-A09693

Salladay-Perez et al., 2026

We sincerely thank the editors and reviewers for the time and effort devoted to evaluating our revised manuscript. We appreciate the constructive feedback and are pleased that the additional revisions and experiments have addressed the reviewers' major concerns. Below, we provide detailed responses to the editorial and formatting suggestions, as well as to some specific points highlighted by the reviewers.

Response to Editor's comments:

Dear Anthony,

Thank you for submitting your revised manuscript "P21+TREM2+-Senescent Macrophages Fuel Inflammaging and Metabolic Dysfunction-Associated Steatotic Liver Disease." (NATAGING-A09693A). It has now been seen by the original referees and their comments are below. The reviewers find that the paper has improved in revision, and therefore we'll be happy in principle to publish it in *Nature Aging*, pending minor revisions to satisfy the referees' final requests and to comply with our editorial and formatting guidelines.

We thank the editors and the *Nature Aging* staff for their thoughtful evaluation of our revised manuscript and for the detailed guidance provided to facilitate its final preparation for publication. We have carefully followed the Author Checklist and implemented all requested revisions to ensure compliance with the journal's editorial and formatting standards. These revisions have streamlined and strengthened the manuscript, including reducing the main text to fewer than 6000 words and aligning the content with *Nature Aging's* policies and practices.

Response to referee #1

In the revised manuscript, the authors have fully addressed all of the raised concerns and questions. The manuscript has been sufficiently approved to warrant publication and thus I recommend acceptance. Lastly, could the authors confirm the catalog number for the p16 antibody in the reporting summary, as it does not appear in the Abcam antibody database.

We thank reviewer #1 for their helpful comments and suggestions, which greatly improved the quality of the manuscript, and for highlighting this error in the reporting of the p16 antibody. The correct p16 antibody cat# is Ab211542. We have now updated the correct antibody information in our reporting summary.

I co-reviewed this manuscript with one of the reviewers who provided the listed reports. This is part of the *Nature Aging* initiative to facilitate training in peer review and to provide appropriate recognition for Early Career Researchers who co-review manuscripts.

We thank Reviewer #2 for reviewing our manuscript and offering thoughtful suggestions to improve the clarity and quality of our work.

Response to referee #3

Thank you for submitting the revised manuscript and your detailed point-by-point response to my comments. I have carefully reviewed the authors' responses and the corresponding changes made to the manuscript. I am pleased to confirm that the authors have addressed all my concerns thoroughly and thoughtfully. In particular, I appreciate the additional experiments and data analyses they have conducted. These new results directly and convincingly support their conclusions, effectively resolving the uncertainties I had raised. The revisions have significantly enhanced the manuscript's clarity, rigor, and overall quality. The study's scientific value is now more clearly demonstrated and well-supported. Therefore, I am satisfied with the revisions and recommend that the manuscript, in its current form, is suitable for publication.

We thank Reviewer #3 for their thorough evaluation of our manuscript and for the insightful comments and suggested experiments. We are pleased that our revisions have addressed their major concerns, which we agree have significantly strengthened the clarity, rigor, quality, and impact of the manuscript.

Response to referee #4

The authors addressed most of my concerns. I only have some minor comments:
Please check the following sentence. Line 183: Next, CD38 is an ectoenzyme with NADase enzymatic activity and is the primary consumer of NAD⁺ in LPS-activated macrophages³⁶.

We thank reviewer #4 for their helpful suggestions and are pleased that we have satisfied their major concerns. We also acknowledge their minor concern about the line above. Although we believe this sentence is supported by the data in the cited manuscript, we acknowledge that there may be a difference of opinion and interpretation of the data. To address the concern, we revised the sentence to “Next, CD38 is an NADase enzyme that is highly expressed by M1 macrophages,” which is also supported by the same citation (PMID: 33199924) and data, and is generally an accepted phenotype observed by multiple independent groups.